# MAPL regulates gasdermin-mediated release of mtDNA from lysosomes to drive pyroptotic cell death

Mai Nguyen [1,2,6], Jack J. Collier [1,2,6], Olesia Ignatenko [1,2], Genevieve Morin [3], Vanessa Goyon[1], Alexandre Janer[1], Camila Tiefensee Ribeiro[1,2], Austen J. Milnerwood [1], Sidong Huang[3,4], Michel Desjardins [2,5] & Heidi M. McBride [1,2] ✉

Mitochondrial control of cell death is of central importance to disease mechanisms from cancer to neurodegeneration. Mitochondrial anchored protein ligase (MAPL) is an outer mitochondrial membrane small ubiquitin-like modifier ligase that is a key determinant of cell survival, yet how MAPL controls the fate of this process remains unclear. Combining genome-wide functional genetic screening and cell biological approaches, we found that MAPL induces pyroptosis through an inflammatory pathway involving mitochondria and lysosomes. MAPL overexpression promotes mitochondrial DNA trafficking in mitochondrial-derived vesicles to lysosomes, which are permeabilized in a process requiring gasdermin pores. This triggers the release of mtDNA into the cytosol, activating the DNA sensor cGAS, required for cell death. Additionally, multiple Parkinson's disease-related genes, including *VPS35* and *LRRK2*, also regulate MAPL-induced pyroptosis. Notably, depletion of MAPL, *LRRK2* or *VPS35* inhibited inflammatory cell death in primary macrophages, placing MAPL and the mitochondria–lysosome pathway at the nexus of immune signalling and cell death.

Cell death is a fundamental event in multicellular organisms. It occurs in response to developmental cues to sculpt tissues and can be induced by a multitude of external signals including inflammation and metabolic stress[1]. It has long been established that mitochondria play central regulatory roles in cell death pathways, yet much remains to be learnt about the signalling events that couple external triggers to the activation of mitochondrial machineries that facilitate death. Many of these signals induce alterations in mitochondrial architecture, including cristae remodelling, stabilization of endoplasmic reticulum contact sites and fragmentation. During apoptosis, the mitochondrial anchored protein ligase (MAPL; gene name *MUL1*), the outer mitochondrial membrane

and peroxisomal small ubiquitin-like modifier (SUMO) E3 ligase, conjugates SUMO1 to the fission GTPase DRP1. This stabilizes the oligomer and maintains mitochondrial constriction at contact sites with the endoplasmic reticulum[2]. The stabilized constriction facilitates calcium flux and cristae remodelling required for efficient assembly of BAX/BAK pores and cytochrome c release, which ultimately leads to apoptotic cell death. Consistently, MAPL depletion protects against pro-apoptotic stimuli, whereas overexpression drives cell death[3,4]. Evidence now supports the role of MAPL in promoting cell death in multiple contexts in vivo; mice lacking MAPL develop spontaneous hepatocellular carcinoma[4], whereas selective loss of MAPL in parvalbumin

[1]Department of Neurology and Neuroscience, Montréal Neurological Institute, McGill University, Montréal, Quebec, Canada. [2]Aligning Science Across Parkinson's (ASAP) Collaborative Research Network, Chevy Chase, MD, USA. [3]Rosalind & Morris Goodman Cancer Institute, McGill University, Montreal, Quebec, Canada. [4]Department of Biochemistry, McGill University, Montréal, Quebec, Canada. [5]Département de Pathologie et Biologie Cellulaire, Université de Montréal, Montréal, Quebec, Canada. [6]These authors contributed equally: Mai Nguyen, Jack J. Collier. ✉e-mail: heidi.mcbride@mcgill.ca

interneurons protects neonatal pups against anaesthesia-induced cell death and memory loss[5]. In addition, a recent cardiomyocyte-specific MAPL[−/−] mouse model demonstrated protection against septic cardiomyopathy showing reduced cell death and inflammation[6].

Studies identifying SUMO and ubiquitin substrates of MAPL highlight additional physiological and cellular functions of this evolutionarily conserved protein[7]. MAPL has been linked to mitophagy[8,9], cell growth and stress sensing[10–15], bile acid metabolism in liver[4] and innate immune signalling[16–19]. Taken together with recently developed MAPL[−/−] mouse models, these studies suggest that MAPL couples several cellular signalling cascades to distinct mitochondrial responses.

Given that exogenous MAPL expression induces highly penetrant cell death, we developed a genome-wide CRISPR knockout screen to systematically identify proteins that act along this pathway. The data place MAPL as a key protein ligase that drives inflammasome activation and the generation of mitochondrial-derived vesicles (MDVs) containing mitochondrial DNA (mtDNA). The essential contribution of lysosomes as the site of mtDNA release into the cytosol offers a new mechanism of cGAS–STING activation regulated by gasdermin pores and a series of Parkinson's disease-related proteins. Collectively, our data provide new insights into the integrated role of mitochondrial signalling in pyroptotic cell death.

## Results

### MAPL induces BAX/BAK-independent cell death

To simulate MAPL activation, we transduced cells with adenovirus to express Flag-tagged MAPL (MAPL–Flag) alongside two controls: a MAPL deletion mutant lacking the RING domain (ΔRING–Flag) required for MAPL's SUMOylation activity, or an (empty) virus encoding reverse tetracycline transactivator (rtTA)[20] (Fig. 1a). As observed previously, both MAPL–Flag and ΔRING–Flag are efficiently targeted to mitochondria[21] (Extended Data Fig. 1a). Exogenous MAPL expression induced cell death in multiple cell lines, which was dependent on the RING domain (Fig. 1b), revealing that MAPL-induced cell death depends on SUMOylation (Fig. 1b). To test whether MAPL overexpression killed cells through a canonical BAX-dependent apoptotic pathway, we transduced rtTA or MAPL–Flag adenovirus into wild-type (WT) baby mouse kidney (BMK) cells or BMK cells lacking both BAX and BAK (BAX/BAK DKO)[22]. As a positive control, we used adenovirus to express truncated BID (tBID), which directly activates BAX to drive apoptosis[23]. While both tBID or MAPL expression efficiently induced cell death and caspase-3/7 cleavage in WT cells, only tBID-induced cell death required BAX/BAK (Fig. 1c)[24]. This indicated that MAPL induced cell death and caspase-3/7 cleavage through a BAX/BAK-independent pathway.

### Genome-wide CRISPR survival screen reveals that MAPL induces pyroptosis

To characterize this cell death pathway, we performed a genome-wide CRISPR knockout screen to identify genes required for MAPL-induced cell death (Fig. 1d). The top-ranked candidate gene was coxsackievirus and adenovirus receptor (*CXADR*), essential for the initial transduction with Ad–rtTA or Ad–MAPL, providing internal validation of the screen (Fig. 1e and Supplementary Table 1). Gene set enrichment analysis identified top hits linked to inflammatory responses, including interleukin (IL)-1 signalling, MyD88-related signalling and Toll-like receptor (TLR) cascades (Fig. 1f). Other immune-related genes whose knockout was protective against MAPL-induced cell death included *NFKB2*, *IL18RAP* (adaptor for IL-18-induced activation of JAK–STAT)[25] and *IFNAR2* (encoding the interferon (IFN) α and β receptor[26]). Additional protective gene deletions linked to TLR signalling included *MST4* (encoding STK26)[27] and *PELI3* and *IRAK4*, encoding a ubiquitin ligase and kinase, respectively, that regulate TLR and IL-1 signalling[28,29] (Fig. 1g).

We confirmed that MAPL-induced cell death is inflammatory in nature in U2OS cells given the RING-dependent expression and release of the cytokine IL-6 (Fig. 1h,i). Similarly, MAPL expression in U2OS cells rapidly upregulated NLRP3 messenger RNA and protein (Fig. 1h,j), whereas low levels of NLRP3 oligomerization were detected by blue native gel analysis (BN–PAGE) (Extended Data Fig. 1b). We confirmed these observations in a second cell type, human fibroblasts, where MAPL induced pro-inflammatory gene upregulation of *IL6, NLRP3* and *IL1B* in a RING-dependent manner (Extended Data Fig. 1c). Together, these data demonstrate that MAPL activity can drive immune signalling directly from the mitochondrial outer membrane in multiple cell types, which is remarkable given that these cell lines are historically reported to have limited immune signalling capacity[30,31].

The screen identified additional regulators of pyroptotic cell death, including the inflammasome sensor (NLRP3), adaptor (ASC1) and the effector (caspase-1)[32], and key substrates of caspase-1 and caspase-3 activation, gasdermin (GSDM) D and E, respectively (Fig. 1g). Cleaved N-terminal GSDMD fragments form pores in the plasma membrane leading to cell rupture, the hallmark feature of pyroptosis[33]. Immunoblotting analyses confirmed that expression of MAPL upregulated NLRP3 protein levels and cleavage of both GSDMD and GSDME (Fig. 1j). We found MAPL-induced GSDM cleavage is caspase-dependent, as it is nearly absent in the presence of the pan-caspase inhibitor ZVAD (Extended Data Fig. 1e). We then examined plasma membrane permeabilization, indicated by cellular uptake of a cell impermeant DNA stain, SYTOX Green that can only access nuclear DNA when the plasma membrane is ruptured[33]. Ectopic expression of MAPL resulted in SYTOX Green uptake into cells, whereas tBID induced apoptotic morphology with no breach of the plasma membrane, demonstrating a stark distinction between apoptotic (tBID) and pyroptotic (MAPL) death pathways (Fig. 1k). siRNA depletion of either GSDMD or GSDME protected against MAPL-induced cell membrane rupture and death (Fig. 1l and Extended Data Fig. 1d). These data indicate co-dependency of GSDMD and GSDME within this pathway, where they must act at distinct steps along the pathway.

**Fig. 1 | Genome-wide CRISPR knockout screen reveals that MAPL induces pyroptosis. a**, Representative immunoblot showing MAPL–Flag or MAPL–ΔRING–Flag expression in U2OS[OCT–GFP] cells 24 h after viral transduction (*n* = 3 independent experiments). **b**, CellTitre Glo Assay measuring cell death in multiple human cell lines after expression of rtTA, MAPL, ΔRING or tBID for 48–96 h. *n* = 6 independent experiments for U2OS, *n* = 3 for human fibroblast and for HUH-7 *n* = 2 for HUH-7. For comparison of MAPL versus rtTA, *P* < 0.0001 for U2OS, *P* = 0.0053 for human fibroblast and for HUH-7. **c**, CellTitre Glo Assay measuring cell death in WT BMK cells and BAX/BAK DKO BMK after expression of rtTA, MAPL or tBID for 48 h (left). Data from four independent experiments are expressed as percentage of cell survival relative to rtTA-expressing WT, to which *P* values are shown relative to **c**. Immunoblot showing cleavage of caspase-3 and 7 after expression of rtTA, MAPL, ΔRING or tBID. *n* = 3 independent experiments (right). **d**, Overview of genome-wide CRISPR screening approach using HUH-7 cells. **e**, Plot of differential gene scores from the screen −log$_2$(ratio of normalized averages between rtTA and MAPL), mean value from four sgRNAs per gene. **f**, Overrepresentation analysis showing the top five Reactome pathway hits from genes with differential gene score >2. **g**, Selected hits from CRISPR screen involved in inflammation. **h**, qRT–PCR assessment of IL-6 and NLRP3 mRNA levels in U2OS[neo] (24 and 48 h after expression). Data (*n* = 3 independent experiments) are expressed as fold change over control cells treated with rtTA. **i**, ELISA assay measuring the release of IL-6 from U2OS[neo] following MAPL expression. *n* = 2 independent experiments. **j**, Western blot analysis of U2OS[neo] following 48 h rtTA or MAPL expression. *n* = 3 independent experiments. **k**, Representative fluorescent images of U2OS cells showing that MAPL, and not tBID, induced inflammatory cell death via SYTOX Green uptake (representative of *n* = 3 independent repeats). **l**, GSDMD or GSDME depletion significantly inhibit MAPL-induced SYTOX Green uptake (*n* = 849, 623, 1,110, 807, 1,341 and 1,853 cells, in order). Graphs show means of independent repeats ± s.e.m., with analysis by one-way analysis of variance (ANOVA) with Tukey's multiple comparison test.

## Cytosolic mtDNA drives MAPL-induced cell death

The second-most enriched sgRNA target upon MAPL expression within our CRISPR screen was the cytosolic sensor of double-stranded DNA, cGAS (cyclic GMP–AMP synthase, encoded by *MB21D1*), and its partner STING was also among the hits (Figs. 1g and 2a). Upon binding DNA, cGAS generates cGAMP, which binds and activates STING to drive type I IFN responses[34]. We validated the requirement for cGAS in MAPL-induced death using siRNA approaches in U2OS cells, with cGAS depletion limiting MAPL-induced cell death (Fig. 2b and Extended Data Fig. 2a). In addition, MAPL overexpression triggers a

RING domain-dependent type I IFN response through increased *IFNA4* and *IFNB1* expression in human fibroblasts. (Fig. 2c). To determine whether mtDNA was required to activate cGAS during MAPL-induced cell death, we expressed MAPL in parental 143b cells or their derivative line lacking mtDNA (Rho0, validated by PCR in Extended Data Fig. 2b)[35]. Rho0 cells were fully protected against MAPL-induced pyroptosis, as measured by SYTOX Green uptake (Fig. 2d), indicating that mtDNA is likely responsible for cGAS–STING activation. Cytosolic DNA has also been shown to directly activate the NLRP3 inflammasome, leading to GSDM cleavage[36]. While mtDNA was essential for cell death, MAPL

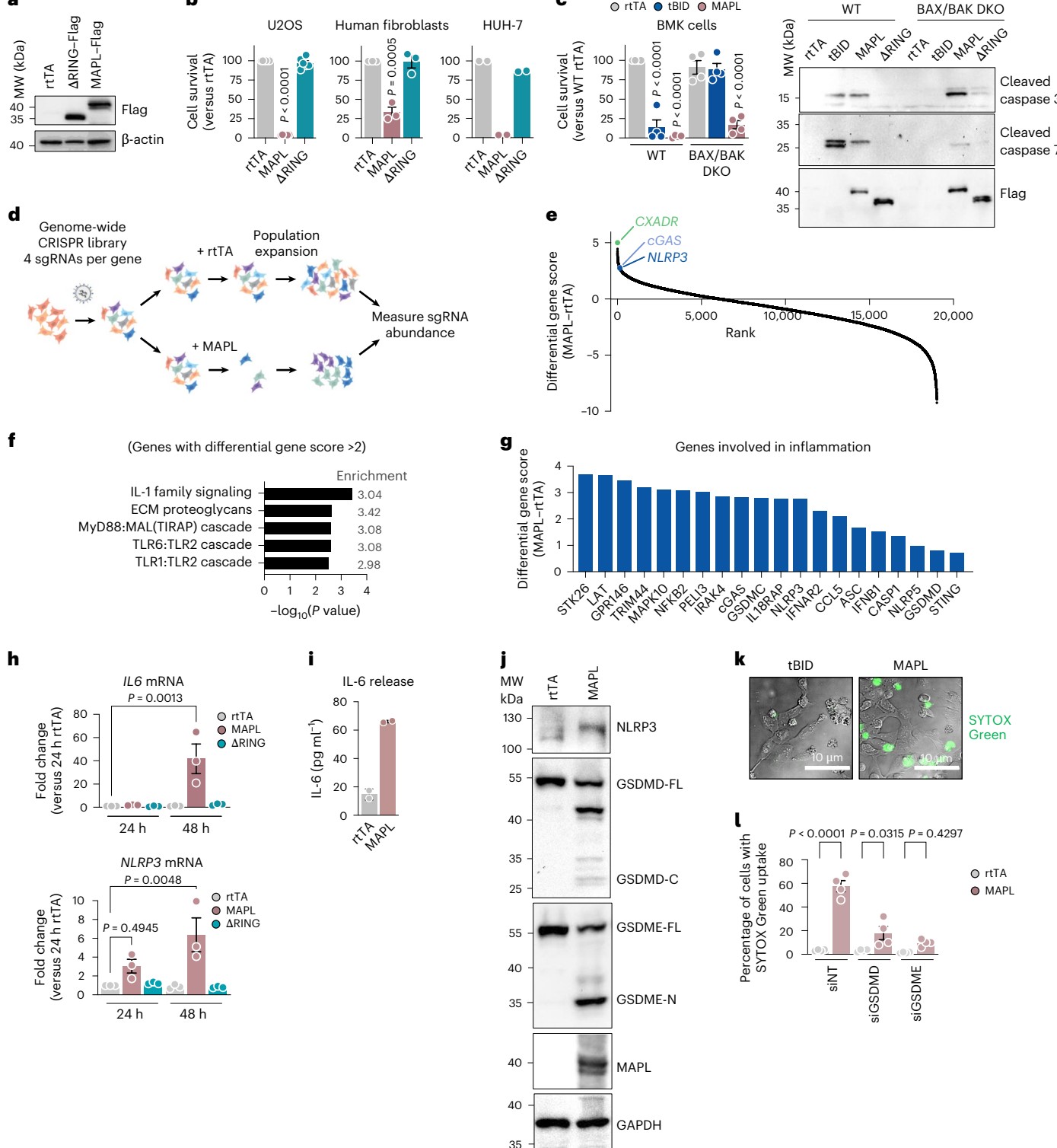

expression in Rho0 cells did not completely abolish the cleavage of GSDMD and E cleavage, indicating that MAPL overexpression can initiate the inflammasome/caspase machinery required to cleave GSDMs independently, or upstream, of mtDNA release into the cytosol (Fig. 2e).

Confocal microscopy showed a large increase in distinct DNA-positive foci in the cytosol of U2OS cells following MAPL expression, a process fully dependent on MAPL SUMOylation activity (Fig. 2f,g). These DNA foci were absent in Rho0 cells confirming their mitochondrial origin (Fig. 2h,i). Having shown that MAPL induced GSDM cleavage, we tested whether mtDNA was released from mitochondria through GSDM pores, previously shown to form in the mitochondrial membranes[37–40]. However, silencing GSDMC, GSDMD and GSDME together had no effect on the appearance of cytosolic mtDNA foci in MAPL-expressing cells (Fig. 2j). The blots also confirmed that MAPL-induced activation of caspase-3 would drive cleavage of GSDME[41,42], as silencing either GSDMD or E had no impact on caspase-3/7 cleavage (Fig. 2k). Of note, these data revealed that loss of GSDME blocked the MAPL-induced cleavage of GSDMD, whereas the reverse was not true, placing GSDME upstream of GSDMD along the pathway (Fig. 2k).

### MIROs facilitate mtDNA removal in mitochondrial-derived vesicles

We next sought to understand how mtDNA is released from mitochondria. Using time-lapse imaging with PicoGreen to label DNA, we observed the active and rapid release of mtDNA foci from mitochondria in MAPL-expressing cells with a linear trajectory, reminiscent of previously described MDV production[43,44] (Fig. 3a and Supplementary Video 1). Cargo-selected MDVs are formed by the lateral tubulation of mitochondrial membranes along microtubules via the Rho GTPases MIRO1 and MIRO2 before DRP1-mediated scission[44]. We observed that the knockdown of MIRO1/MIRO2 significantly reduced the number of mtDNA foci released into the cytosol after MAPL expression (Fig. 3b and Extended Data Fig. 3a). mtDNA exit into the cytosol has been shown to occur through voltage-dependent anion channel (VDAC) pores[45,46] or VDAC-induced inner membrane vesicles (VDIMs)[47]. However, silencing VDAC1 did not impact mtDNA foci in cytosol (Fig. 3c) and MAPL-induced cell death was unaffected (Extended Data Fig. 3b,c). These results indicate that mtDNA is not released through mitochondrial pores formed by either BAX/BAK, GSDMs or VDAC.

We next sought to identify protein components within mtDNA-positive foci to further determine whether these structures were membrane-bound. After testing a series of candidate MDV cargoes[48], we found that approximately 35% of cells contained cytosolic mtDNA foci that co-stained with mitochondrial complex I but not TOM40 (mtDNA⁺/complex I⁺/TOM40⁻) (Fig. 3d,e). We observed mtDNA-positive structures emerging from mitochondria at the tip of a thin tubule containing complex I but not TOM40, consistent with the MIRO-dependent tubulation preceding MDV formation (Fig. 3f). Electron microscopy (EM) analyses revealed that MAPL overexpression led to the formation of electron-dense double-membrane protrusions

from mitochondria, consistent with the production of vesicular profiles (Fig. 3g).

### mtDNA is trafficked to endolysosomal subcompartments via MDVs

MDVs are known to deliver mitochondrial cargo to endolysosomal compartments[49–51] and we found that approximately 12% of lysosomes contained DNA after MAPL expression (Fig. 4a–c). Although mitochondria can be delivered to lysosomes through autophagy pathways, we did not observe the recruitment of the key mitophagy protein Parkin–GFP[52] to mitochondria after rtTA or MAPL overexpression (Extended Data Fig. 4a). Instead, time-lapse imaging highlighted a process where MDVs are delivered to lysosomes already containing DNA (Fig. 4d, Supplementary Video 2 and Extended Data Fig. 4b). In a second type of dynamic event, we observed lysosomal arrival proximal to the mtDNA exit sites along a mitochondrial tubule, enabling direct and rapid transfer of content from mitochondria to lysosomes (Fig. 4e and Supplementary Video 3). These data identify a subset of lysosomes as the target organelle for mtDNA-containing MDVs.

### VPS35 and LRRK2 promote MAPL-induced cell death

The involvement of lysosomes in mtDNA trafficking during MAPL-induced cell death was supported by a series of lysosomal hits within the CRISPR screen (Fig. 5a,b). Hits included Rab5B, Rab8B, Rab1A, Rab3D, Rab42, Rab30 and Rab7B, vesicle tethering subunits of the HOPS (VPS33b) and CORVET (VPS41) complexes[53–55], as well as ESCRT-III machinery responsible for the generation of intraluminal vesicles within the multivesicular body (CHMP6 and CHMP4B)[56] (Fig. 5c). The extensive list of trafficking machineries suggested that endolysosomal identity and sorting are essential for MAPL-induced cell death. Notably, a series of Parkinson's disease (PD)-related genes linked to the endomembrane system, including *VPS35*, *LRRK2*, *GBA*, *TMEM175* and *VPS13C* were also identified (Fig. 5c) indicating that PD proteins may act along the MAPL pathway to regulate immune signalling and cell death[57].

Silencing VPS35 or LRRK2 in U2OS cells efficiently rescued cell death induced by MAPL, but not by tBID, demonstrating the importance of PD-related proteins in this pyroptotic pathway (Fig. 5d). We previously showed that the PD-related protein VPS35 interacts with MAPL to regulate the trafficking of MDVs to peroxisomes[58]. VPS35 is a component of the retromer complex[59] that acts to sort cargo within the endomembrane system. VPS35 has also been linked with mitochondrial to lysosome contacts and/or transport pathways[60–62]. Consistent with evidence that VPS35 acts in the generation of some classes of MDVs, silencing this protein decreased the number of cytosolic mtDNA foci in U2OS cells (Fig. 5e,f and Extended Data Fig. 5a). In contrast, LRRK2 silencing had no impact on the generation of cytosolic DNA foci (Fig. 5e,f and Extended Data Fig. 5b), suggesting it functions downstream of MDV release. These data are consistent with a recent study showing the requirement for VPS35 in the generation of mtDNA-positive MDVs induced by mtDNA replicative stress[63].

---

**Fig. 2 | MAPL-induced pyroptosis requires cGAS and mtDNA release.**
**a**, Analysis of the CRISPR screen showing enrichment of sgRNAs in Ad–rtTA (left) and Ad–MAPL (right) transduced conditions. **b**, CellTitre Glo Assay in U2OS after 3-day cGAS knockdown by siRNA then 48 h rtTA or MAPL expression. *n* = 4 independent experiments. **c**, qRT–PCR measurement of mRNA in human fibroblasts after 48 h rtTA, MAPL or ΔRING expression. *n* = 3 independent experiments. **d**, Measurement of inflammatory cell death by SYTOX Green uptake in 143b (WT) or Rho0 (143b lacking mtDNA) after 48 h of rtTA, MAPL or ΔRING expression. *n* = 3 independent experiments (*n* = 1,283, 882, 1,456, 948, 1,212 and 1,384 cells analysed, in order of presentation). **e**, Western blot analysis of 143b (WT) and Rho0 cells 48 h after MAPL expression, representative of three independent experiments. Arrowhead represents cleaved GSDME or GSDMD. **f,g**, Representative confocal microscope images of cytosolic DNA foci (green)

in U2OS^OCT–GFP cells (magenta) from *n* = 3 independent experiments (**f**) with analysis of number of cytosolic DNA foci from three independent experiments (**g**) (*n* = 178, 120 and 171 cells, in order). Circles highlight DNA⁺/OCT–GFP⁻ MDVs. **h,i**, Representative confocal microscope images of cytosolic DNA foci (circles) in 143b (WT) and Rho0 cells (**h**) and quantification from three independent experiments (*n* = 123, 75, 100 and 92 cells, in order) (**i**). **j**, Quantification of cytosolic DNA foci by immunofluorescence after 3-day GSDMC/D/E knockdown by siRNA (siGSDMs) followed by 48-h rtTA or MAPL expression. *n* = 3 independent experiments (*n* = 92, 103, 95 and 72 cells, in order). **k**, Representative immunoblot analysis of caspase-3/7 and GSDMD/E cleavage in U2OS^neo in response to MAPL expression (*n* = 3 independent experiments). Red asterisk denotes the cleaved GSDMD. Graphs show means of independent repeats ± s.e.m., with analysis by one-way ANOVA with Tukey's multiple comparison test (**b–d**, **g**, **i**, **j**).

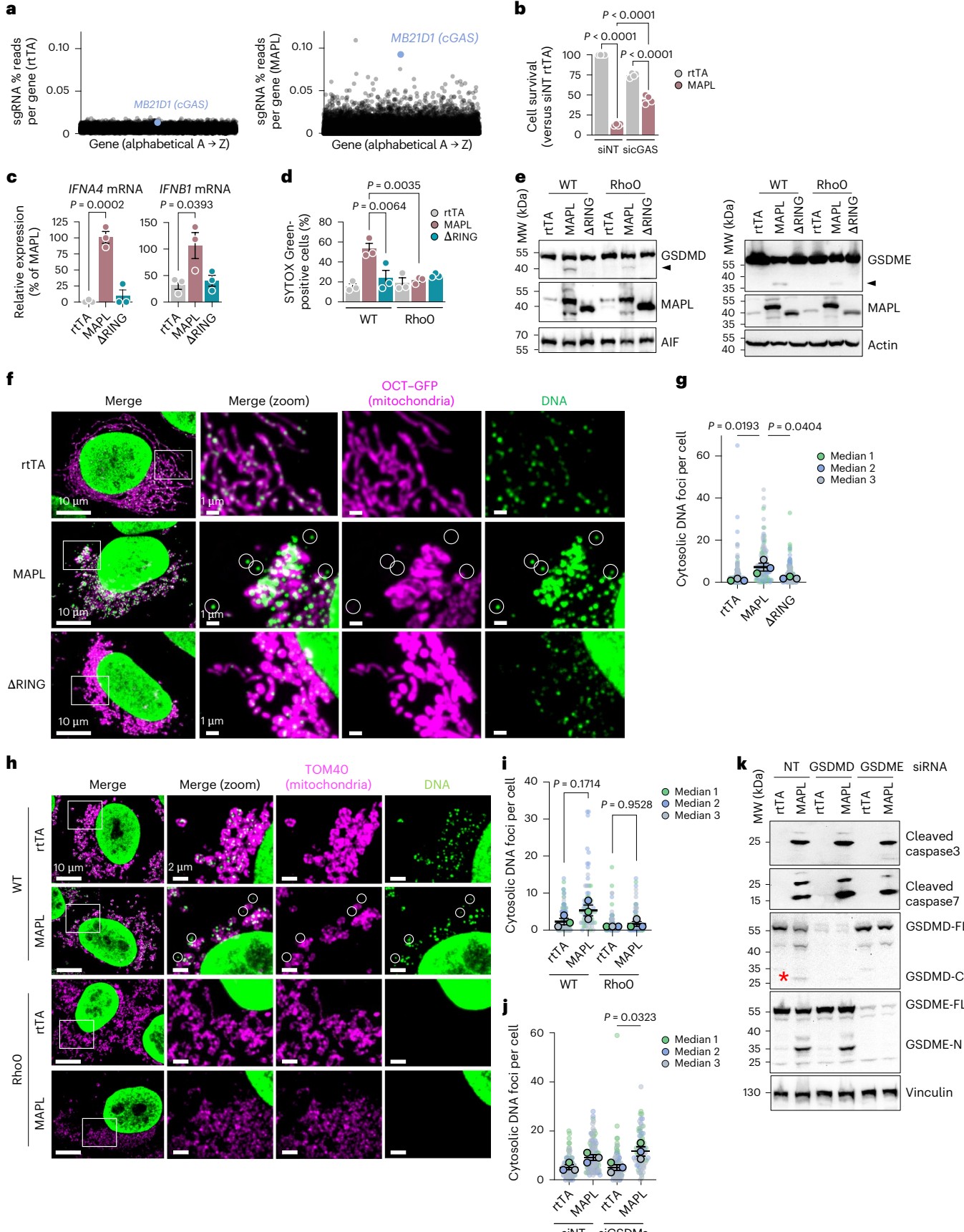

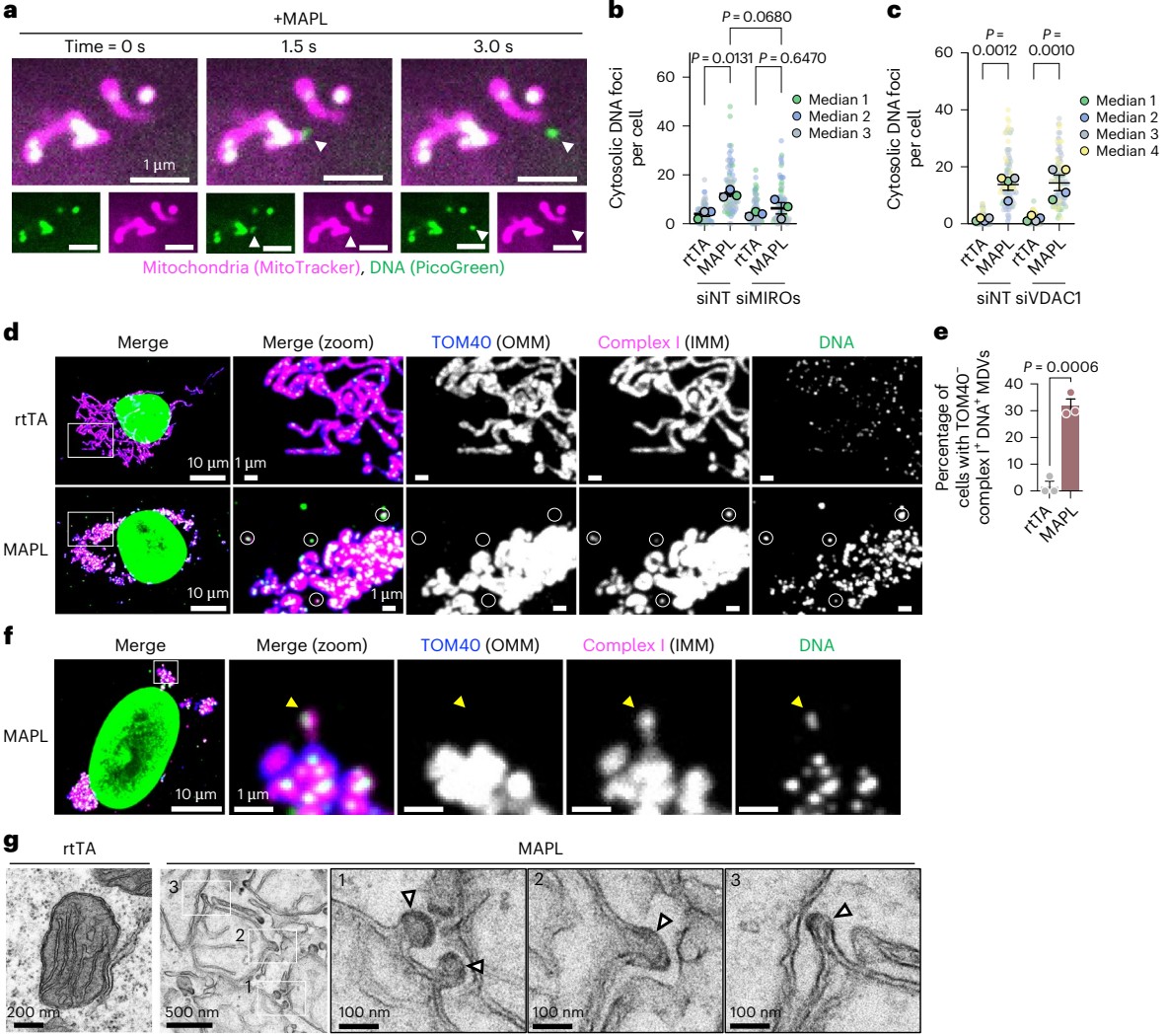

**Fig. 3 | MAPL promotes release of mtDNA in mitochondrial-derived vesicles.**
**a**, Time-lapse confocal microscopy of U2OS cells expressing MAPL for 24 h showing release of mtDNA into cytosol (arrowhead). **b**, Analysis of cytosolic DNA foci by confocal microscopy after 3-day MIRO1/2 depletion by siRNA (siMIROs) followed by expression of MAPL for 48 h. $n$ = 3 independent experiments ($n$ = 76, 70, 82 and 69 cells, in order). **c**, Analysis of cytosolic DNA foci by confocal microscopy after 5-day VDAC1 depletion by siRNA (VDAC1 or NT as control) followed by expression of MAPL for 48 h. $n$ = 3 independent experiments ($n$ = 87, 88, 78 and 87 cells, in order). Graphs show means of independent repeats ± s.e.m., with analysis by one-way ANOVA with Tukey's multiple comparison test

(**b**,**c**). **d**,**e**, Immunofluorescence images of human fibroblasts after 24 h rtTA or MAPL expression showing TOM40⁻/complex I⁺/DNA⁺ MDVs (circles) (**d**), with quantification of three independent repeats ($n$ = 59 and 104 cells, in order) (**e**), with comparison by two-tailed, unpaired $t$-test. **f**, TOM40⁻/complex I⁺/DNA⁺ tubulation involved in MDV formation (arrowhead). **g**, Representative electron micrographs showing electron-dense double-membrane protrusions (arrowheads) in U2OS cells after 24 h of MAPL expression ($n$ = 2 independent experiments). IMM, inner mitochondrial membrane; OMM, outer mitochondrial membrane.

LRRK2 has been shown to regulate lysosomal function[64–68]; however, its loss did not impact the delivery of mtDNA to lysosomes in MAPL-expressing cells (Fig. 5g,h). While there is a trend toward an ~10% increase in lysosomal size upon MAPL expression (Extended Data Fig. 5c), we observed a subset of very enlarged (1–3 μm) lysosomes in LRRK2-silenced cells upon MAPL expression (Fig. 5g,i). This is indicative of lysosomal stress, a phenotype seen in LRRK2⁻/⁻ tissues and cells under stress conditions[66,69–73]. These data confirm that loss of LRRK2 did not alter mtDNA delivery to lysosomes but revealed underlying changes in the lysosomal response to MAPL expression in the absence of LRRK2.

**mtDNA is released into the cytosol from damaged lysosomes**
With mtDNA within the lysosome lumen, the question of how it is released to the cytosol for the activation of cGAS remained unsolved.

We hypothesized that the mtDNA-containing lysosomes may become selectively ruptured, which we monitored by quantifying the recruitment of cytosolic β-galactoside-binding lectin protein 3 (galectin-3; GAL3)[74]. Indeed GAL3–RFP was recruited to a subset of LAMP1-stained lysosomes in MAPL-expressing cells (Fig. 6a,b). Lysosome breach was dependent on MAPL ligase activity, as this was not observed upon ectopic expression of MAPL-ΔRING (Fig. 6b). GAL3 can drive recruitment of autophagy adaptors to promote lysophagy and/or ESCRT machinery to repair lysosomal membrane wounds, processes that ultimately enhance cell survival[74,75]. Consistent with this U2OS cells transiently expressing GAL3–RFP displayed increased survival in a colony formation assay after MAPL overexpression (Fig. 6c). This provides evidence that lysosomal breach is a key step of the death pathway.

Notably, GAL3 often colocalized with PicoGreen signal, where approximately 50% of damaged lysosomes contained DNA after 48 h

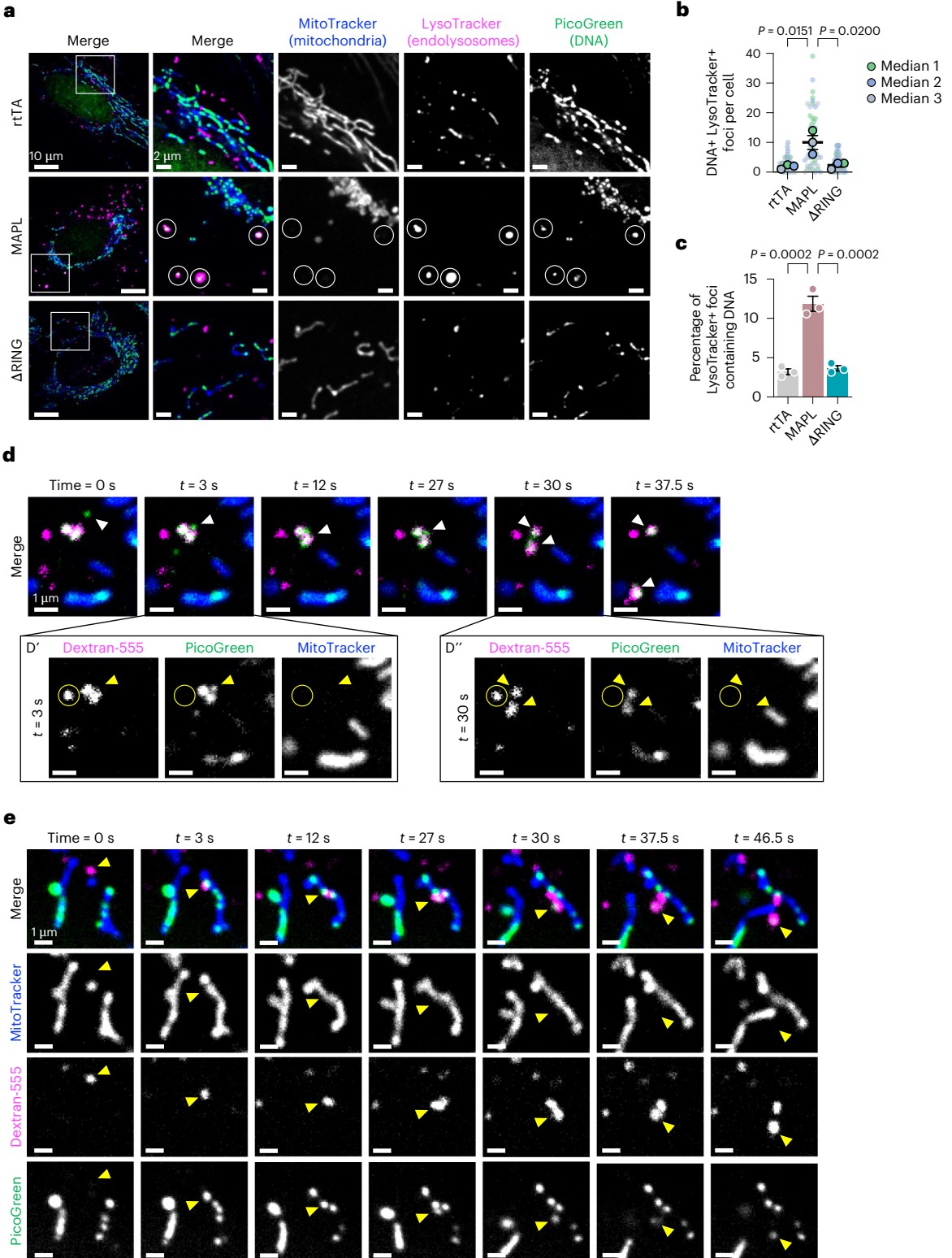

**Fig. 4 | mtDNA is delivered to endolysosomes. a–c,** Representative images of live-cell imaging showing the presence of DNA (green and circles) in endolysosomes marked by LysoTracker (magenta) in U2OS^neo cells following 24 h of MAPL expression (**a**), with quantification of three independent experiments (*n* = 53, 48 and 52 cells, comprising 4,352, 4,605 and 4,405 lysosomes, respectively, and in order) (**b,c**). Graphs show means of independent repeats ± s.e.m., with analysis by one-way ANOVA with Tukey's multiple comparison test. **d,** Time-lapse confocal microscopy showing PicoGreen-positive DNA trafficking to Dextran-labelled lysosome. At *t* = 3 s, arrowhead shows DNA contact the lysosome (**d′**), then at *t* = 30 s, arrowheads mark the completion of DNA engulfment by the lysosome (**d″**). Circle highlights a DNA-negative lysosome for comparison. **e,** 0–12 s shows the recruitment of a lysosome to the mitochondrial surface, 27–30 s shows the removal of mtDNA into the lysosome and 37.5–46.5 s shows that lysosomes move away from the mitochondrial surface with DNA (arrowhead).

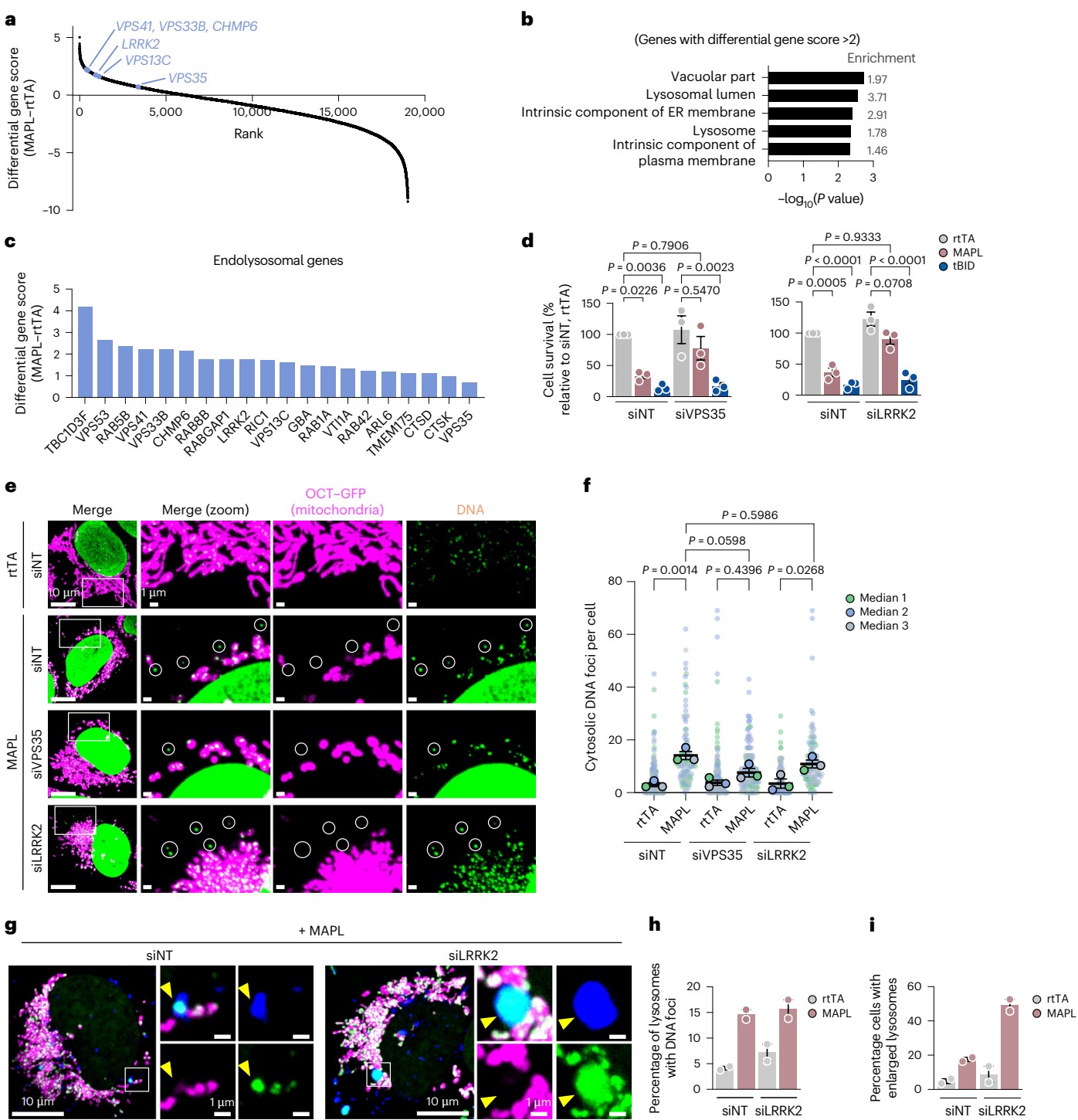

**Fig. 5 | VPS35 and LRRK2 promote MAPL-induced cell death. a**, Plot of differential gene scores from the CRISPR screen (overview in Fig. 1d) highlighting enrichment of genes involved in endolysosomal function and identity. **b**, Top five hits from Gene Ontology overrepresentation analysis of genes with differential gene score >2. **c**, Selected endolysosome-related hits from CRISPR screen. **d**, SYTOX Green uptake measurement following silencing of VPS35 (siVPS35) or LRRK2 (si-LRRK2) for 3 days in U2OS cells then 48 h expression of rtTA, MAPL or tBID. n = 3 independent experiments. **e,f**, VPS35 or LRRK2 knockdown in U2OS^OCT–GFP cells for 3 days by siRNA before MAPL expression for 48 h. Cytosolic DNA was analysed by immunofluorescence microscopy (**e**) with quantification of three independent experiments (**f**) (n = 146, 96, 144, 127, 144 and 100 cells, in order). **g–i**, Example of enlarged lysosomes after 24-h MAPL overexpression following 3-day LRRK2 depletion (siLRRK2) in U2OS cells (**g**), including quantification of percentage of DNA⁺ lysosomes (**h**) and percentage of cells with enlarged lysosomes (**i**). n = 2 independent experiments (n = 33, 62, 33 and 72 cells). Graphs show means of independent repeats ± s.e.m., with analysis by one-way ANOVA with Tukey's multiple comparison test (**d,f**).

of MAPL expression (Fig. 6d–f). To visualize direct DNA release from GAL3+ structures, we performed video analysis focusing on a larger GAL3+ lysosome and captured tubulation events from DNA-containing lysosomes and observed 'puffs' of PicoGreen stain diffusely exiting these organelles that would then dissipate (Fig. 6g and Supplementary Video 4). This provides further evidence that DNA can exit breached lysosomes. We then used qPCR to amplify mtDNA from highly purified cytosol and confirmed the presence of mitochondrial encoded genes MT-CO1 and, to a lesser extent, MT-ND6 and MT-ND1 (Fig. 6h and Extended Data Fig. 6). These data confirm that mtDNA is released into the cytosol from lysosomes that breach under conditions of MAPL activation.

## LRRK2 and GSDME regulate lysosome breach

We next sought to determine what caused lysosomal breach upon expression of MAPL. LRRK2 was shown to be recruited to damaged lysosomes induced by extrinsic stressors like LLOME, chloroquine or zymosan, where it plays a key role in mediating membrane repair[37,64,65,67]. While loss of LRRK2 did not affect mtDNA delivery to lysosomes (Fig. 5g,h), there was a significant reduction in GAL3 recruitment onto lysosomes in LRRK2-silenced, MAPL-expressing cells (Fig. 6i). This suggests that the loss of LRRK2 protected against cell death by reducing lysosomal breach.

We next tested whether the presence of mtDNA within this subset of lysosomes may be required to signal or mediate the breach in the membranes, yet there was no effect on GAL3 recruitment to lysosomes in Rho0 cells expressing Ad–MAPL (Fig. 6j). As Rho0 cells still exhibited cleavage of GSDMs, we tested whether they form pores in lysosomes upon MAPL expression. Indeed, silencing GSDMD and E together resulted in a near complete absence of lysosome breach, measured by GAL3 recruitment to lysosomes (Fig. 6k). Individual silencing of either GSDMD or GSDME showed a striking dependence on GSDME, where loss of GSDMD had no impact on GAL3 recruitment to lysosomes (Fig. 6l). Western blot confirmation of silencing efficiency again revealed the dependency on GSDME for GSDMD cleavage induced upon MAPL expression (Fig. 6l). To biochemically test whether the pore-forming N terminus of GSDME was detectable on lysosomes and may therefore cause their breach, we generated stable U2OS cells expressing a lysosomal membrane marker TMEM192-3xHA to immune-isolate lysosomes[76]. Immunoblotting revealed that the cleaved N-terminal fragment of endogenous GSDME was present in lysosome fractions isolated from MAPL-expressing cells (Fig. 6m). This indicates that full-length GSDME is present on lysosomal membranes, and that GSDME fragments are associated (and likely forming pores) with lysosomal membranes, following MAPL expression. This is consistent with GSDME pores enabling the release of mtDNA from lysosomes to the cytosol, leading to activation of cGAS–STING signalling.

## MAPL regulates inflammatory responses in primary macrophages

We next tested whether MAPL is necessary and sufficient for pyroptotic cell death and immune signalling in primary mouse bone-marrow-derived macrophages (BMDMs). Biochemical markers of pyroptosis were efficiently induced in BMDMs ectopically expressing MAPL, including NLRP3, ASC, p-STAT3 and nuclear factor (NF)-κB. Consistent with other cell types, MAPL triggered cleavage of GSDMD and GSDME (Fig. 7a), and cell death measured by SYTOX Green, within 36 h (Fig. 7b). Furthermore, we found that MAPL-induced cell death was blocked by inhibitors of NLRP3 (MCC950) and caspase-1 (YVAD) (Fig. 7b). This provides further evidence supporting the involvement of NLRP3/caspase-1-mediated pyroptotic cell death when MAPL is expressed.

To test whether the mtDNA pathway was also required for MAPL-induced death in BMDM we employed inhibitors of cGAS (G140) or STING (H151), which also prevented cell death (Fig. 7b). This again places both the NLRP3–Casp1 and cGAS–STING pathways as integral to MAPL-induced cell death, here in professional immune cells. While NLRP3 was required for MAPL-induced cell death, BN-PAGE analysis did not reveal oligomerization of the inflammasome complex, as seen after 4 hr treatment with lipopolysaccharide (LPS) and nigiricin (Extended Data Fig. 7a). However, NLRP3 and ASC protein pelleted in a detergent-insoluble fraction in MAPL-expressing cells (Extended Data Fig. 7b), consistent with a phase-separated, active inflammasome[77].

To test the requirement for MAPL in pyroptotic cell death we isolated primary BMDMs from WT and MAPL−/− mice[4,5,18] and treated them with LPS and IFNγ to mimic bacterial infection and induce pyroptosis. This treatment led to the generation of NLRP3 oligomers, along with canonical pyroptotic markers pSTAT3, NF-κB and gasdermin cleavage, all of which were reduced in presence of the NLRP3 inhibitor (Extended Data Fig. 7c,d). MAPL−/− BMDMs were resistant to pyroptosis after LPS and IFNγ treatment, measured by SYTOX Green uptake (Fig. 7c,d). LPS and IFNγ treatment also induced mtDNA+ MDV production after 6 h of treatment, a process fully dependent on MAPL (Fig. 7e,f). These data further implicate MAPL as an essential driver of mtDNA release during inflammation.

We then examined whether MAPL is required for LPS/nigericin-induced pathways where nigericin actively exchanges potassium for hydrogen across membranes, directly disrupting electrochemical potential and activating mitochondrial ATPase activity. This is a common model for pyroptotic death where it has been demonstrated that GSDMs form pores directly in the mitochondria[39,78], perhaps negating the requirement for mtDNA+ MDVs to deliver cargoes to the lysosome. As expected, loss of MAPL had no impact on the nigericin-induced assembly of NLRP3 oligomers, phosphorylated STAT3 or NF-κB (Extended Data Fig. 7e,f).

**Fig. 6 | Lysosome breach causes mtDNA escape into cytosol.**
**a,b**, Immunofluorescence images of U2OS^GAL3−RFP showing breached (GAL3−RFP-positive) LAMP1-positive lysosomes (arrowheads) after 48 h expression of MAPL, but not after rtTA or ΔRING expression (**a**), with quantification of three independent repeats (**b**) (n = 173, 124 and 155 cells, in order). **c**, U2OS cell survival measured by colony formation assay after transient pcDNA3.1 (empty vector) or GAL3−RFP expression before MAPL expression and recovery for three weeks. **d–f**, Live imaging of U2OS^GAL3−RFP cells showing GAL3−RFP-positive lysosomes containing DNA (PicoGreen) after 48 h expression of MAPL (arrowheads) (**d**), but not after ΔRING expression, with quantification at 24 h and/or 48 h post-expression of three independent experiments (**e,f**) (n = 61, 64, 84 and 65 cells, in order). **g**, Evidence of DNA exit from breached lysosomes (arrowheads) using time-lapse confocal imaging after 48 h of MAPL expression in U2OS^GAL3−RFP cells. **h**, qPCR of U2OS^neo cytosolic fractions after rtTA or MAPL expression for 24 h. Data (n = 3) are expressed as fold change over cells treated with rtTA. **i**, Quantification of GAL3−RFP+ lysosomes in U2OS^GAL3−RFP after 3-day knockdown of LRRK2 (n = 50, 92, 67 and 100 cells, in order). **j**, Quantification of GAL3−RFP+ lysosomes in 143b (WT) and Rho0 cells after 48 h rtTA or MAPL expression (n = 18, 43, 20 and 47 cells, in order). **k**, Quantification of GAL3−RFP+ lysosomes in U2OS^GAL3−RFP after 3-day knockdown of GSDMs (n = 178, 138, 149 and 135 cells, in order) followed by 48 h of rtTA or MAPL expression. **l**, Quantification of cells with GAL3−RFP positive lysosomes after 5-day knockdown of GSDMD or GSDME followed by 2-day MAPL expression (n = 212, 215, 253, 189, 340 and 369 cells, in order) (left). Representative immunoblot of three independent experiments (right). **m**, Representative blots from immuno-isolation of lysosomes from U2OS cells expressing TMEM192-3xHA before immunoblotting (n = 3 independent experiments). P8, post 8,000g pellet. Graphs show means of independent repeats ± s.e.m., with analysis by one-way ANOVA with Tukey's multiple comparison test (**b,e,i,g,k,l**). Graphs represent means ± s.e.m., with comparisons by two-tailed, unpaired t-test (**c,h**).

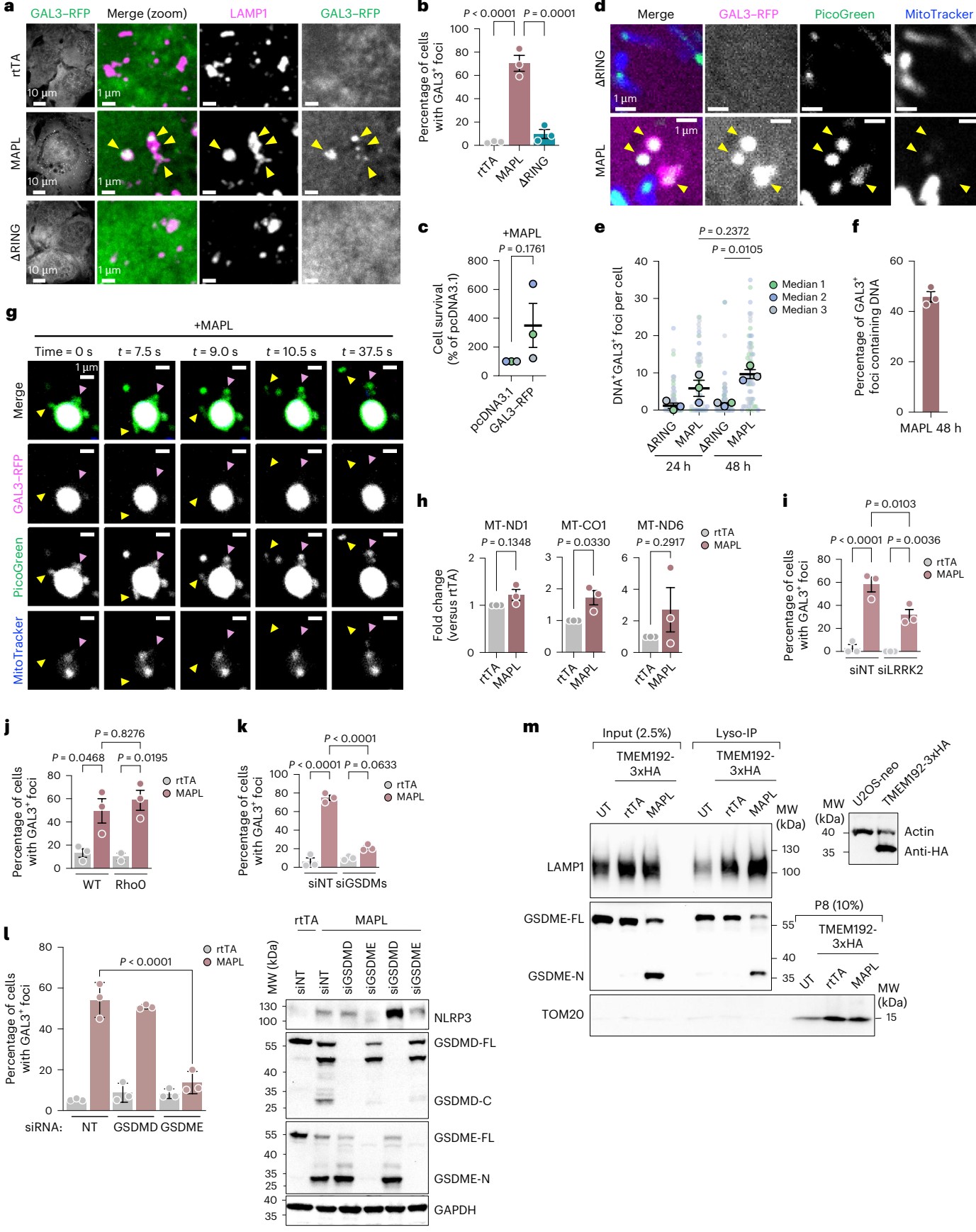

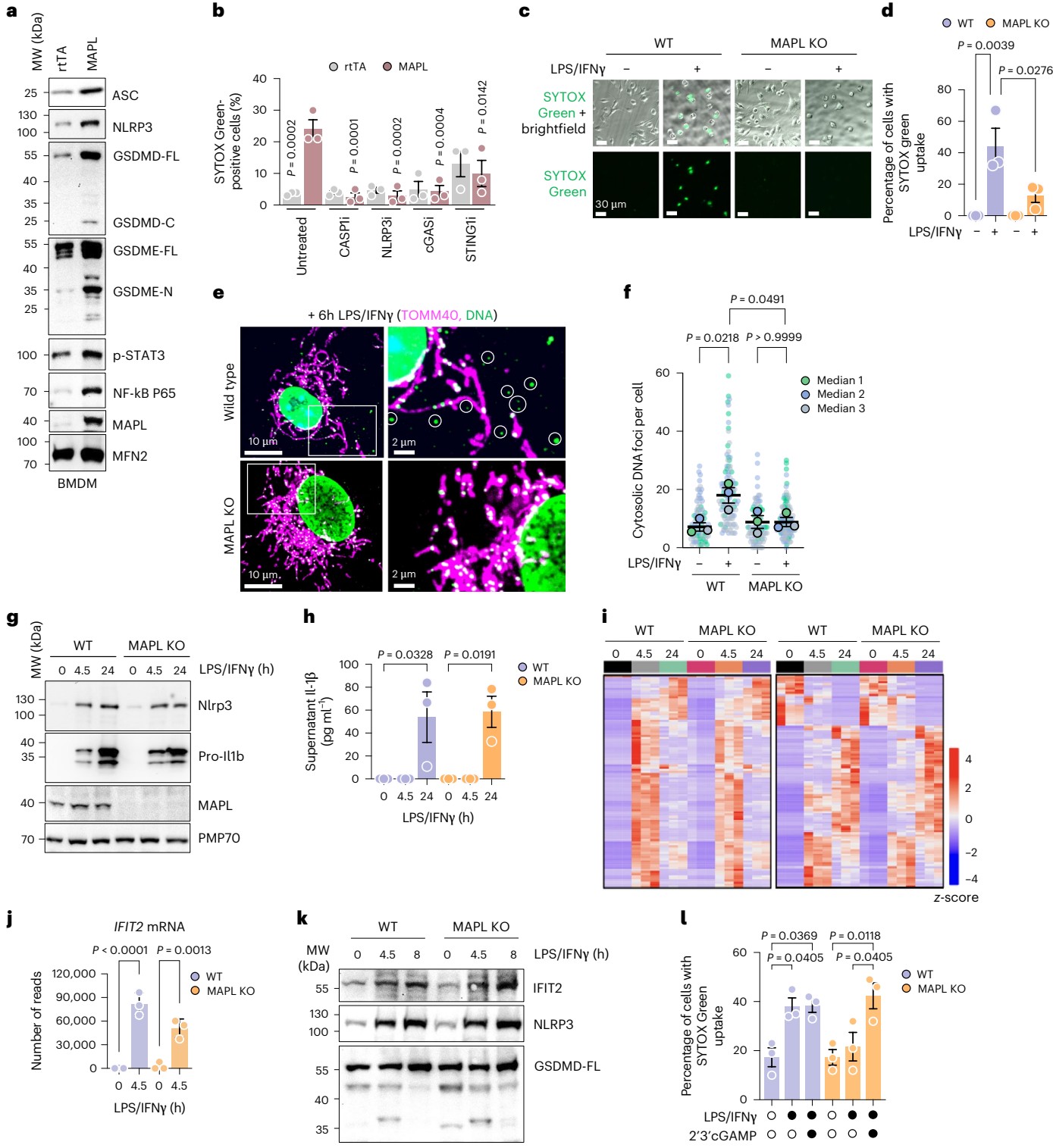

**Fig. 7 | MAPL regulates inflammatory cell death in primary macrophages.**
**a**, Immunoblot analysis following 48-h expression of rtTA or MAPL in WT BMDMs.
**b**, Quantification of MAPL-induced inflammatory cell death by SYTOX Green uptake in BMDMs treated with or without chemical inhibitors to caspase-1 (CASP1i, 25 μM YVAD), NLRP3 (NLRP3i, 70 μM MCC950), cGAS (cGASi, 5 μM G140) or STING1 (STING1i, 10 μM H151). $n = 3$ independent experiments ($n = 680$, 350, 632, 616, 587, 691, 425, 711, 561 and 568 cells, in order). $P$ values relative to untreated + MAPL. **c,d**, SYTOX Green uptake in WT and MAPL$^{-/-}$ BMDMs treated with LPS and IFNγ for -36 h (**c**), with quantification of three independent repeats (**d**) ($n = 523$, 655, 312 and 755 cells, in order). **e,f**, Confocal microscope images of cytosolic DNA (circles) in WT but not MAPL knockout (KO) BMDMs after 6 h LPS and IFNγ treatment (**e**), with quantification of three independent repeats (**f**)

($n = 95$, 113, 88 and 111 cells, in order). **g**, Representative immunoblot of WT and MAPL$^{-/-}$ BMDMs after LPS/IFNγ treatment ($n = 3$ independent experiments). **h**, ELISA assays to quantify IL-1β secretion in the media of BMDMs used for RNA-seq in **i**. **i**, RNA-seq, heatmap showing gene expression changes relative to untreated control BMDMs. **j**, RNA-seq, normalized reads for IFIT2 in each condition from three independent experiments. **k**, Representative immunoblot of WT and MAPL$^{-/-}$ BMDMs after LPS/IFNγ treatment ($n = 3$ independent experiments). **l**, Measurement of SYTOX Green uptake in WT and MAPL$^{-/-}$ BMDMs after LPS/IFNγ treatment with or without addition of 2'3'cGAMP for 36 h, with three independent repeats ($n = 820$, 946, 718, 1,066, 1,046 and 1,055 cells, in order). Graphs show means of independent repeats ± s.e.m., with analysis by one-way ANOVA with Tukey's multiple comparison test (**b,d,f,h,j,k**).

Although MAPL overexpression induced robust NLRP3 expression (Figs. 1 and 7a), loss of MAPL had no impact on LPS and IFNγ-induced NLRP3 or IL-1β expression, or secretion of IL-1β (Fig. 7g,h). MAPL protein expression remained unchanged upon LPS and IFNγ treatment (Fig. 7g). Complete RNA-seq analysis at 0, 4.5 and 24 h LPS and IFNγ treatment showed MAPL$^{-/-}$ and WT cells remained clustered together at each time point and showed few genotype-specific differences in any up- or down-regulated genes (Fig. 7i, Extended Data Fig. 8a and Supplementary Data Table 2). However, there was a small subset of genes that were induced to a lesser extent in MAPL$^{-/-}$ cells (Extended Data Fig. 8b) that included genes with antiviral functions including *Ifi44*, *Ifit2*, *Ifit1bl1* and *Oasl1*, and multiple members of the *p200* protein family, including *Ifi203* and *Ifi204*, which are cytosolic DNA sensors (Extended Data Fig. 8c). However, while *Ifit2* mRNA levels were 35% lower in LPS and IFNγ conditions in MAPL$^{-/-}$ (Fig. 7j), analysis of IFIT2 protein revealed full induction upon LPS and IFNγ in the absence of MAPL (Fig. 7k). Therefore, the impact of MAPL loss on cell death is unlikely due to any changes in canonical inflammatory signalling or transcriptional responses to infection.

As we observed that MAPL was critical for the generation of mtDNA$^+$ MDVs in response to LPS and IFNγ, we hypothesized that activation of cGAS–STING was required for the cell death arm of the pyroptotic pathway independent from any transcriptional response. Indeed, the addition of 2′3′ cGAMP, the product of cGAS enzymatic activity, resensitized MAPL$^{-/-}$ BMDMs to LPS and IFNγ-induced cell death (Fig. 7l), highlighting the involvement of cGAS–STING in pyroptosis. Thus, MAPL is required for pyroptotic cell death in response to TLR/IFNR activation due to its role in mediating mtDNA release and cGAS activation.

Finally, we examined whether the roles of VPS35 and LRRK2 were also conserved in LPS and IFNγ-induced pyroptosis in primary BMDMs. We isolated primary WT BMDMs and silenced VPS35 in these cells ex vivo as germline deletion in mice is embryonically lethal. The data show a loss of mtDNA$^+$ MDVs upon LPS and IFNγ treatment (Fig. 8a,b) and a strong protection against SYTOX Green uptake (Fig. 8c,d) in BMDMs depleted of VPS35. Primary BMDMs isolated from LRRK2$^{-/-}$ mice showed efficient release of mtDNA from mitochondria upon LPS and IFNγ treatment, placing LRRK2 downstream in the pathway (Fig. 8e,f). We also observed the presence of DNA within very enlarged lysosomes in LRRK2$^{-/-}$ treated BMDMs (Fig. 8g), consistent with the data in LRRK2-silenced U2OS cells. Notably, LRRK2$^{-/-}$ BMDM were fully protected against pyroptotic cell death, as measured by SYTOX Green uptake (Fig. 8h,i). Collectively, these data demonstrate that the MAPL-induced pyroptotic pathway uncovered in our CRISPR screen converges with canonical pathways triggered during immune cell death driven by LPS and IFNγ treatment.

## Discussion

The discoveries that in vivo deletion of mitochondrial outer membrane protein MAPL protects against cell death in the heart and nervous system[5,6], but induces liver cancer[4], revealed a critical question of how this SUMO/ubiquitin E3 ligase regulates cell survival from the mitochondrial surface. We identified a new mitochondria–lysosome axis essential for pyroptosis, which places lysosomes as the target of

mtDNA, and the point of its release into the cytosol. The data show that the SUMO/ubiquitination activity of MAPL leads to (1) the activation of caspases-3/7 which are known to cleave GSDME, and (2) VPS35-dependent formation of mtDNA$^+$ MDVs, which target a subset of lysosomes. The LRRK2-dependent insertion of GSDME pores into these lysosomes causes the release of mtDNA into the cytosol, and subsequent activation of both cGAS–STING and inflammasome-driven GSDMD cleavage, the final step essential for pyroptotic cell death. From the data presented we propose a staged model described in Fig. 8j that highlights our findings and the key open questions.

Our initial observation that ectopic expression of MAPL killed cells in a BAX/BAK-independent manner was a surprise given previous evidence that MAPL-driven SUMOylation of DRP1 promoted apoptosis[2]. Of note, we observed cleavage of both caspase-3 and caspase-7 upon MAPL expression, which was also BAX/BAK-independent (Fig. 1c, right), suggesting a non-apoptotic mechanism of caspase activation. Caspase-3/7 cleavage was unaffected upon the loss of either GSDMD or GSDME (Fig. 2k). It has been well established that caspase-3 can cleave GSDME in cells where it is expressed, which can switch the form of death from apoptosis to secondary pyroptosis or necroptosis[41,42]. In the case of ectopic MAPL expression, our data show that the pyroptotic pathway is dominant in the absence of apoptosis. In addition to the essential role of GSDMs in MAPL-induced death, we observe the activation of canonical pyroptotic makers including the expression of NLRP3, phosphorylation of STAT3 and NF-κB, release of IL-6 (and IL-1B in many cell types), and the SYTOX Green uptake as an indicator of cell surface pore formation. Although we did not observe the strong oligomerization of the induced NLRP3/ASC protein (relative to acute LPS/nigericin) visualized by BN–PAGE, we observed that the induced NLRP3/ASC pelleted in a TX-100 insoluble fraction, consistent with a phase-separated form of the complex[77].

The data showed that depletion of GSDME inhibited GSDMD cleavage (Figs. 2k and 6l). We also observed that GSDME was required for the permeabilization of a subset of lysosomes (revealed by GAL3 recruitment) carrying mtDNA (Fig. 6l). Moreover, biochemical isolation of lysosomes showed the incorporation of the N-terminal pore-forming fragment of GSDME within these organelles (Fig. 6m). We also directly observe the release of mtDNA into the cytosol from these structures in live-imaging experiments (Fig. 6g). Cytosolic mtDNA will drive activation of both cGAS–STING and the NLRP3 inflammasome[79]. Caspase-1 was identified among the top hits in the initial genome-wide screen for suppressors of MAPL-induced death (Fig. 1g), and inhibitors of caspase-1 blocked MAPL-induced cell death (Fig. 7b), supporting a role for caspase-1 and the inflammasome in cleaving GSDMD. We consider this the final step in MAPL-mediated pyroptosis, as GSDMD pores forming in the plasma membrane complete cell rupture.

Our data show that the arrival of mtDNA within lysosomes is an essential requirement for this pyroptotic death pathway (Figs. 4–6). We revealed how MAPL expression induces the formation of mtDNA$^+$ MDVs, which are delivered to a subpopulation of lysosomes. This was observed by confocal imaging and ultrastructural EM analyses and was dependent on established MDV machineries including the MIRO GTPases (Fig. 3b) and VPS35 (Fig. 5e,f). The release of mtDNA within

**Fig. 8 | VPS35 and LRRK2 regulate cell death. a–d**, VPS35 depletion in BMDMs by siRNA (siNT or siVPS35) for 3 days before 6-h LPS and IFNγ treatment and quantification of cytosolic DNA foci (circles) by confocal microscopy (**a**,**b**), $n = 3$ independent experiments ($n = 78, 73, 76$ and $91$ cells, in order) or 36-h treatment with LPS and IFNγ before measurement of cell death through SYTOX Green uptake (**c**,**d**) ($n = 3$ independent experiments; $n = 80, 75, 78$ and $93$ cells, in order). **e**–**g**, BMDMs were isolated from WT and LRRK2 KO mice and incubated with LPS and IFNγ for 6 h. Measurement of cytosolic DNA foci (circles) by confocal microscopy is seen from $n = 3$ independent experiments ($n = 80, 85, 63$ and $83$ cells, in order) (**e**,**f**) or by live-cell imaging (arrowhead) (**g**). **h**,**i**, Measurement of cell death through SYTOX Green uptake after 36-h treatment with LPS and IFNγ.

$n = 3$ independent experiments ($n = 82, 86, 64$ and $94$ cells, in order). Graphs show means of independent repeats ± s.e.m. with analysis by one-way ANOVA with Tukey's multiple comparison test (**b**,**d**,**f**,**i**). **j**, A model highlighting the two-step process for MAPL-induced death. MAPL expression results in the activation of caspase-3/7, which can cleave GSDME to drive pores in lysosomes, a process that requires LRRK2. MAPL also drives the MIRO1/2 and VPS35-dependent release of mtDNA$^+$ MDVs and subsequent transport to the lysosomes. Limited release of mtDNA activates cGAS–STING, which plays a non-transcriptional role in amplifying NLRP3 inflammasome activation and caspase-1 activity to cleave GSDMD to form pores in the plasma membrane. Question marks indicate mechanisms left to be resolved.

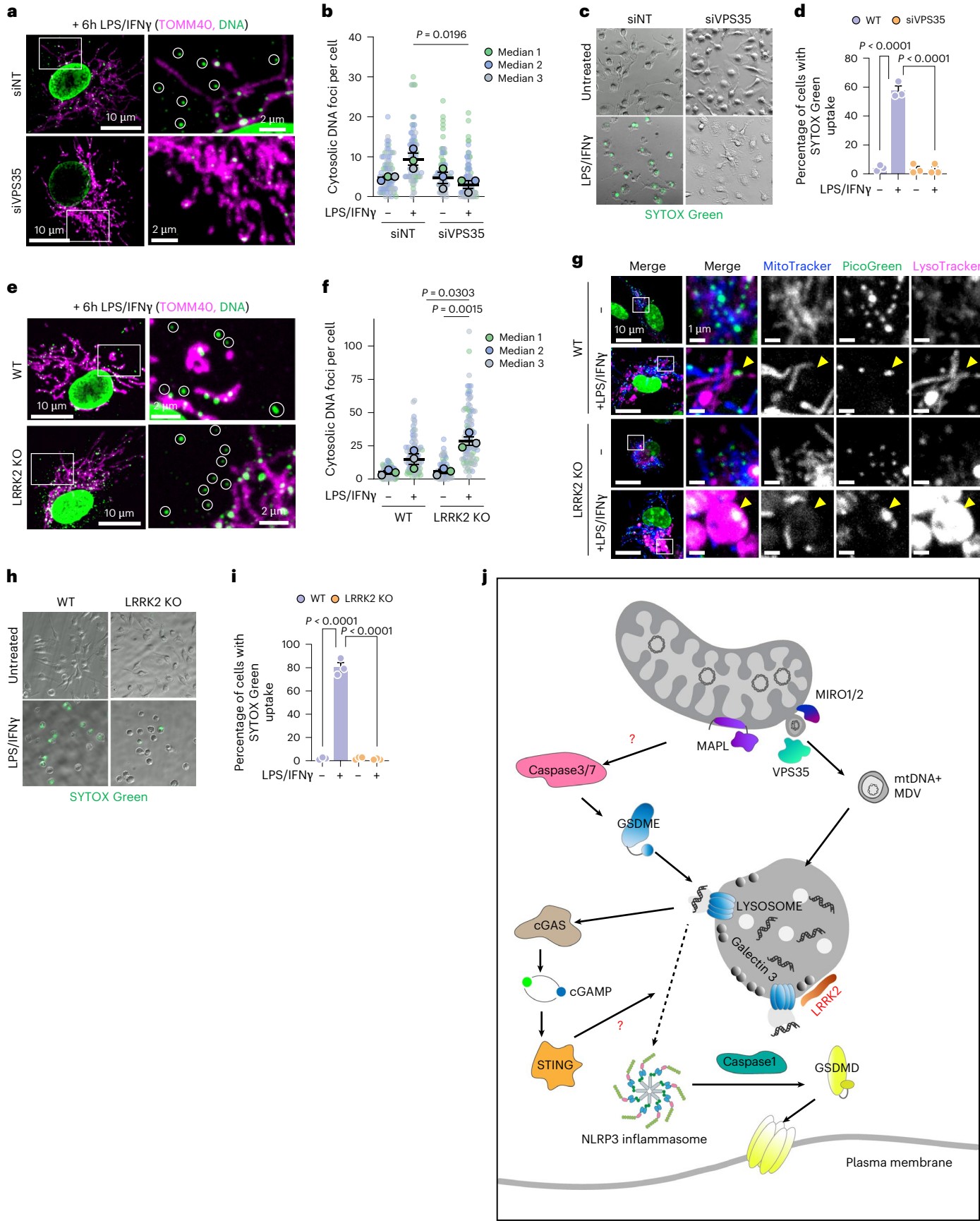

MDVs, or MDV-related pathways has been recently documented in response to altered metabolite (fumarate) accumulation and following mtDNA replication stress[43,60,63,80], indicating this as a common, signal-driven process. In contrast, other studies have shown mtDNA release from mitochondria through direct insertion/action of distinct pores including BAX/BAK, VDAC or GSDMs into the outer membrane[37–40,45,46,81–86]. Our data show that the loss of any of these three mitochondria pore-forming pathways had no impact on mtDNA trafficking induced by MAPL expression.

A key aspect of this discovery is the central role played by lysosomes in the release of mtDNA into the cytosol. This was highlighted by the large number of hits within the CRISPR screen where loss of a host of lysosomal proteins were protective against pyroptotic cell death when silenced (Fig. 5a–c). These included multiple PD-related proteins (LRRK2, VPS35, VPS13C, GBA and TMEM175). These PD genes have been shown to regulate lysosomal sorting pathways, lysosomal lipid flux and metabolism[87–91]. We consider that disruption of any of these components would cause an impairment in lysosomal identity and/or lipid composition of at least a subset of lysosomes, which may even be induced by the immune signalling downstream of TLR4-receptor activation. Pathogenic infection is known to rewire some endocytic trafficking routes, perhaps leading to this unique subpopulation[92]. Our identification of wider components within the endomembrane family, including Rab GTPases and sorting complex (HOPS, CORVET and ESCRT) machinery, support this hypothesis[55,56]. Whatever the origin, this lysosomal subpopulation (~15%) seems uniquely susceptible to receiving mtDNA⁺ MDVs (Fig. 5h) and then become breached in a manner dependent on LRRK2 and GSDME (Fig. 6i,l). Previous work characterized the preferential insertion of cleaved GSDMs into specific lipid environments, with a particular affinity for cardiolipin-enriched membranes[93]. Perhaps the delivery of double-membrane MDVs could contribute cardiolipin to specific lysosomal subcompartments, tagging them for GSDM pore assembly.

While mtDNA⁺ MDV-mediated delivery to lysosomes may commonly occur for degradation and turnover, our data imply that any breach of these lysosomes would launch a rapid innate immune signal. Indeed, cytosolic mtDNA and STING activation have been linked to lysosomal storage disorders[86], neurodegeneration, senescence and aging[94]. The abundance of evidence linking mtDNA to lysosomal disorders would be consistent with the idea that mtDNA can access the cytosol much more readily by crossing the single membrane of a damaged lysosome than by crossing both mitochondrial membranes.

A final step in our staged analysis of MAPL-regulated pyroptotic death was the requirement for cytosolic mtDNA. This seems to be due to two equally critical factors: first the activation of cGAS–STING and second, the cleavage of GSDMD to generate pores in the plasma membrane. cGAS–STING activation is considered critical in driving gene expression of IRF3/NF-κB transcriptional targets important in pyroptotic cell death[95]. However, our experiments in MAPL⁻/⁻ BMDMs suggest an additional, more direct cell biological mechanism. The protection against cell death granted by MAPL loss in LPS and IFNγ-treated BMDMs was eliminated upon addition of the cGAS product and the STING agonist 2'3'cGAMP (Fig. 7l). This is consistent with previous work that identified key non-transcriptional roles for cGAS–STING in the activation of the inflammasome. There, it was shown that activated STING translocation to the lysosome led to lysosomal breach before the activation of the inflammasome[96], although the mechanistic details of this remained unclear. Rho0 cells lacking all mtDNA (and cGAS–STING activation) still showed low levels of MAPL-induced GSDMD/GSDME cleavage and GAL3 recruitment to lysosomes, yet this was insufficient to induce cell death (Figs. 2d,e and 6l). This indicates that STING activation is not required for the initial breach of the lysosomes, which could allow lower levels of mtDNA release. Perhaps the initial mtDNA released is sufficient to activate cGAS–STING where

the arrival of STING at the lysosome could activate its proton channel activity[97], and help enlarge N-terminal GSDME pore formation/assembly, similar to NINJ1 roles in expanding the GSDMD pores at the plasma membrane[98,99]. This could allow the full breach and release of mtDNA from the lysosome and tip the balance to activate the NLRP3 inflammasome, producing more robust GSDMD cleavage and cell death.

Ultimately, our results provide mechanistic insights into the distinct events separating immune signalling from a previously uncharacterized mitochondrial vesicle transport pathway that is essential for pyroptotic death. We show that MAPL is requisite for LPS and IFNγ-induced mtDNA⁺ MDV delivery to lysosomes and subsequent release of mtDNA into the cytosol, where activation of cGAS–STING plays the critical downstream roles in cell death, independent of their transcriptional capacity. This is reminiscent of other cell death pathways in which mitochondria function not as mediators of transcriptional responses, but as executioners of death[100], as observed in our model of pyroptosis.

### Limitations of the study

The limitations of our study include the outstanding question of which substrates MAPL modifies upon exogenous expression to initiate caspase-3/caspase-7 cleavage and innate immune pathways. MAPL has been shown to SUMOylate NLRP3 (ref. 19) and the dsRNA sensor of viral infection RIG-I[18], which could play important functions during cell death. Notable questions also remain about the selection of mtDNA as an MDV cargo. Cargo selection may be linked to immune-related metabolic changes in the matrix. Canonical immune activation leads to the early induction of aconitate decarboxylase 1 (ACOD1, encoded by *immunoresponsive gene 1* (*Irg1*))[101], which rewires the tricarboxylic acid cycle through elevated itaconate production. Itaconate can alkylate cysteine residues, termed 2,3-dicarboxypropylation, or itaconation, of substrates[101]. This modification could facilitate mtDNA incorporation into MDVs, a process that was suggested to drive mtDNA release within the fumarate hydratase mutant cells[43]. Last it is unclear whether the identification of VPS35 and LRRK2 within this pathway represents a finding of important clinical relevance. Future work must delineate the impact of pathogenic variants in each of these genes on this immune pathway[37], and whether/how these functions contribute to the age-related loss of dopaminergic neurons.

### Online content

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

## Methods

### Ethics approval

Animal experimentation was conducted in accordance with the guidelines of the Canadian Council for Animal Care. Protocols were approved by the Animal Care Committees of McGill University.

### Culturing of cell lines

HUH-7 (RRID: CVCL_0336), U2OS (RRID: CVCL_0042), human fibroblasts (a gift from E. Shoubridge, McGill University) and BMK cells (a gift from G. C. Shore, McGill University) were cultured in Dulbecco's modified Eagle's medium (DMEM) containing 4.5 g l$^{-1}$ glucose, L-glutamine and sodium pyruvate (Wisent, 319-027-CL), supplemented with non-essential amino acids (Gibco, 11140050) and 10% fetal bovine serum (FBS; Wisent, 085-150). The 143b and Rho0 cells (143B rho-0 EtBr, RRID: CVCL_ZL52) (a gift from E. Shoubridge, McGill University) were validated by PCR (Extended Data Fig. 3) and cultured in the same conditions, with addition of 5 µg ml$^{-1}$ uridine. BMDMs were cultured in the same medium, with the FBS heat-inactivated at 56 °C before addition. Cell lines were cultured in a humidified incubator at 37 °C and 5% CO$_2$ and regularly tested for mycoplasma using the MycoAlert mycoplasma detection kit (Lonza, LT07-418).

### Isolation and culturing of BMDMs

MAPL KO mice were generated and maintained as previously reported (Extended Data Fig. 8d)[4]. Mice were from a C57BL/6J background, registered as RRID: MGI:7524577. To prepare BMDMs, marrow was flushed from mouse tibias and femurs using a 25-gauge needle with DMEM. The resulting homogenate was filtered through 40-µm mesh filters. Cells were collected by centrifugation and plated on uncoated culture dishes in DMEM, 10% heat-inactivated FBS and 25% L929 cell conditioned supernatant (in DMEM, collected 10 days post-confluency). The resulting macrophages were collected after 8 days in culture for experiments. BMDMs were routinely treated with 1.5 µg ml$^{-1}$ LPS (InvivoGen, tlrl-3pelps) and 0.2 µg ml$^{-1}$ IFNγ (PeproTech, 315-05) for the indicated times. When also used, 2'3'cGAMP (50 µg ml$^{-1}$, InvivoGen, tlrl-nacga23-02) was added at the same time as LPS. Inhibitors to caspase-1 (25 µM CASP1i, Ac-YVAD-cm, InvivoGen inh-yvad), NLRP3 inhibitor (70 µM NLRP3i MCC950, InvivoGen inh-mcc), cGAS inhibitor (5 µM cGASi, G140, InvivoGen inh-g140) or STING1 inhibitor (10 µM STING1i, H151, InvivoGen inh-h-151) and pan-caspase (20 µM ZVAD-FMK, Enzo Life ALX-260-020) were used at the noted concentrations during treatment with LPS and IFNγ, in addition to pre-treatment for 3 h before addition of LPS and IFNγ.

### Transfection and generation of stable cell lines

U2OS cells (RRID: CVCL_0042) were transfected with 2.5 µg of plasmid per well (six-well plate) at approximately 80% confluence using Lipofectamine 2000 (Thermo Fisher, 11668019) according to the manufacturer's instructions. To generate stable expressors, the following plasmids were used for transfection: pmRFP-C1-Galectin-3 (RRID: CVCL_F0QJ) (U2OS$^{RFP-Gal3}$) (a kind gift from M. Gutierrez[67]) pOCT−GFP (U2OS$^{OCT-GFP}$)[102] (a gift from J. Silvius, McGill University) and TMEM192-3xHA (U2OS$^{TMEM192-3xHA}$) (RRID: CVCL_F0QI, Addgene 102930) and pcDNA3.1 (U2OS$^{neo}$) (Thermo Fisher, V79020). Two days after transfection, cells were sparsely plated in 150-mm dishes in medium containing 750 µg ml$^{-1}$ G418 or 1 µg ml$^{-1}$ puromycin until individual colonies were forming. Single clones were subsequently isolated. Confirmation of exogenous gene expression was determined by confocal microscopy and/or immunoblotting.

### RNA interference

siRNA-mediated knockdown of genes was facilitated by Lipofectamine RNAiMAX (Invitrogen, 13778150) according to manufacturer's instructions and confirmed by immunoblotting. siRNA was used at a final concentration of 20 nM. The following human On-TARGET plus individual or Smart Pool siRNAs (Dharmacon) were used: cGAS/MB21D1 (L-015607-02), GSDMC (L-014716-02), GSDMD (L-016207-00), GSDME/DFNA5 (L-011844-00), LRRK2 (L-006323-00), MIRO1 (L-010365-01), MIRO2 (L-008340-01) and VPS35 (J-010894-05). Non-targeting siRNA (D-001810-10) was used as a control.

### Blue native PAGE to evaluate NLRP3 oligomer

A total of 2 ×10$^6$ cells per condition were collected, washed once with PBS and solubilized for 20 min at 4 °C with 0.5% digitonin in a buffer containing 75 mM Bis-Tris (pH 7.0), 1.75 M aminocaproic acid and 2 mM EDTA, followed by centrifugation at 20,000g for 20 min, at 4 °C. The supernatant was run on 4–15% polyacrylamide gradient gels. Separated proteins were transferred to a nitrocellulose membrane using a semi-dry system (Bio-Rad) and immunoblot analysis was performed with the indicated antibodies.

### Cells lysates, PAGE and immunoblotting

Cells were solubilized in lysis buffer comprising 50 mM HEPES, pH 7.4, 150 mM NaCl, 1% Triton X-100 and 1 mM EDTA followed by centrifugation for 10 min at 10,000g and 4 °C. Total protein concentration was determined using a Bradford assay (Bio-Rad 5000006EDU) before samples were mixed with sample buffer (50 mM Tris, pH 6.8, 0.1% glycerol, 2% SDS and 100 mM β-mercaptoethanol) and heated to 95 °C for 5 min. The protein extract was separated with SDS−PAGE using a 4–20% gradient gel, before transfer to 0.22-µm-pore supported nitrocellulose membrane (Bio-Rad). Western-Lightning PLUS-ECL (PerkinElmer, NEL105001EA) with an INTAS ChemoCam (INTAS Science Imaging) was used to visualize bands, before processing with ImageJ software. The information of the primary and secondary antibodies and the dilutions is provided in Supplementary Table 3.

### Cell death and pyroptosis assays

Cell survival was determined using Promega CellTiter-Glo v.2.0 Luminescence Cell Viability Assay kit (Promega, G9241) as described by the manufacturer. In brief, cells were plated in triplicate in a 96-well culture dish and cultured overnight (1 × 10$^4$ cells per well). Cells were infected with appropriate adenovirus vectors (20 plaque-forming units (p.f.u.) per cell) for the corresponding time (typically 48–96 h) after which CellTiter-Glow reagent was added and luminescence was determined using a Promega GloMax 96 Microplate Luminometer.

Pyroptotic cell death was determined via SYTOX Green (Invitrogen S7020) uptake. Monolayer cells plated on glass-bottom dishes were treated for indicated time followed by addition of SYTOX Green at final concentration of 1 µM directly in the medium and immediately imaged. Images were acquired using Zeiss Axio Observer (RRID: SCR_021351) and analysed manually using Fiji (ImageJ, RRID: SCR_003070).

### Detection of extracellular cytokine

BMDMs or U2OS cells were plated on six-well plates (2 × 10$^6$ cells per well), treated under the indicated conditions in 1 ml medium per well. Following treatments, medium was collected and centrifuged at 1,000g to remove floating cells. The cleared medium was subjected to IL-1β (R&D System DY401-05) or IL-6 (R&D System DY401-05) ELISA detection as described by the manufacturer.

### Preparation of adenovirus and infection of cells

Adenovirus expressing MAPL−Flag (MAPL) and MAPL−ΔRING−Flag (ΔRING) were custom generated for us by Vector Biosystems[4]. Ad−rtTA and Ad−tBID (truncated Bid) were gifts from G. C. Shore (McGill University). The virus was expanded by one round of infection of Ad−293 cells (Agilent, 240085, RRID: CVCL_98040) at 5 p.f.u. per cell in complete growing medium for 3–5 days. Cells were collected and lysed by three rounds of freeze−thaw. Viral titre (p.f.u.) was determined by plaque-forming assay. In brief, serial dilution of the viral stock was used to infect Ad−293 for 2 h after which time 1.25% low-melting agarose was

overlaid. After 2–3 weeks, visible plaques were counted to determine p.f.u. per ml. For transient expression, cells were routinely incubated with adenovirus encoding rtTA, MAPL or ΔRING at 10–30 p.f.u. per cell in complete medium. Expression was confirmed by immunoblotting and/or immunofluorescence.

## Pooled genome-wide CRISPR knockout screen

Functional genetic screens were performed in HUH-7 cells using the human CRISPR Brunello lentiviral pooled library (four unique single guide RNAs (sgRNAs) per gene, comprising a total of ~76,000 sgR-NAs)[103]. HUH-7 cells (10 × 150-mm dishes with $8 × 10^6$ cells per dish) were infected with each pooled library virus at low multiplicity of infection achieving single viral integration and selected in medium containing puromycin (2 µg ml$^{-1}$) for 3 days. Post-puromycin selection cells were passaged for another 7 days to allow complete editing. Cells were trypsinized, pooled, counted and plated into 100 × 150-mm dishes ($5 × 10^5$ cells per dish). Then, 50 plates were infected with Ad–rtTA and 50 plates were infected with Ad–MAPL–Flag adeno virus at 10 p.f.u. per cell for 2 days. Medium was changed every 2–3 days. For Ad–rtTA, cells reached confluency in 1.5 weeks and for Ad–MAPL–Flag, ~3 weeks. Cells were collected, trypsinized to single cells, pooled and mixed well. Approximately $20 × 10^6$ cells were used for isolating genomic DNA with High Pure PCR Template Preparation kit (Roche) as described by the manufacturer. Library preparation for next-generation sequencing was conducted as described previously[104]. In brief, the gRNA sequences were amplified from genomic DNA by PCR using Phusion HF DNA polymerase (Thermo Fisher Scientific) using a two-step amplification adding a unique 6-bp index per sample and sequencing adaptor sequences. PCR products were purified using the High Pure PCR Product Purification kit (Roche) and quantified using the Quant-iT PicoGreen dsDNA Assay kit (Thermo Fisher Scientific, P7581) before sequencing on a HiSeq2500 System (Illumina). Sequencing reads were mapped to the library using xcalibr (https://www.thermofisher.com/order/catalog/product/OPTON-30965) and counts were then analysed with MAGeCK (https://sourceforge.net/p/mageck/wiki/Home/, v.0.5.8)[105] using the robust rank aggregation algorithm to identify genes whose perturbation (knockout or overexpression) primarily enhanced fitness in the MAPL overexpressing group but not the control group. An sgRNA score was attributed to each sgRNA representing its relative abundance within each group. Guides with more than one mismatch and guides with fewer than 50 reads in the rtTA library were filtered out. Reads of the remaining guides were normalized to the total (before filtering) sequencing library size for rtTA and MAPL overexpression libraries and ratios of normalized averages (MAPL/rtTA) were calculated. The differential scores were calculated as $\log_2$(ratio of normalized averages) and ranked. Values with differential scores equalling zero were excluded from graphing. Calculations were computed using RStudio (RRID: SCR_000432, https://posit.co/, v.2021.09.1), and graphs were plotted using Prism v.10 (GraphPad Prism, RRID: SCR_002798).

## Immunofluorescence and image acquisition

For immunofluorescent microscopy, cells were grown on uncoated glass coverslips (Thermo Fisher, 12-545-81). After treatment, the medium was removed from the coverslips and cells were directly fixed with 5% (wt/vol) paraformaldehyde (PFA; Sigma, P6148) in PBS (prewarmed to 37 °C) for 15 min at 37 °C. Coverslips were washed three times in PBS, before 10 min of quenching with 50 mM NH$_4$Cl. Cells were washed again three times in PBS followed by permeabilization using 0.1% Triton X-100 in PBS for 10 min. After three washes in PBS, cells were blocked with 5% FBS in PBS for 10 min, then sequentially incubated in primary antibody (prepared in 5% FBS in PBS) for 1 h, or added together for 2 h. During sequential incubations, between primary antibodies coverslips were washed three times in PBS. After the final primary antibody and washes, cells were incubated in the appropriate secondary antibodies (1:2,000) for

1 h. Cells were then washed twice in dH$_2$O, before the addition of 1 µg ml$^{-1}$ 4,6-diamidino-2-phenylindole (DAPI; Invitrogen, D1306) in dH$_2$O for 10 min to stain nuclei. After one further wash in dH$_2$O, coverslips were mounted onto microscope slides (VWR, 82003-412) with approximately 10 µl of mounting medium (Dako, S302380-2). Images were acquired using an Olympus IX83 confocal microscope containing a spinning disk system (Andor/Yokogawa CSU-X) using an Olympus UPLANSAPO Å-100/1.40 NA oil objective and Andor Neo sCMOS camera and MetaMorph software (MetaMorph Microscopy Automation and Image Analysis Software (RRID: SCR_002368). Laser lines corresponding to 405 nm, 488 nm, 561 nm and 637 nm were used, with excitation and exposure times maintained across samples for each independent experiment. Z-stacks of 0.2 µm per stack were typically used. Details of the primary and secondary antibodies and the dilutions are provided in Supplementary Table 4.

## Live-cell microscopy

Cells were plated on uncoated glass-bottom dishes (MatTek, P35G-1.5-20C, size no. 1.5) then treated as described. For adenoviral expression, cells were typically incubated with 10–20 p.f.u. per cell of corresponding virus for 6 h before three washes in PBS and replacement with fresh medium. Cells were then incubated for 24 or 48 h. For co-staining mtDNA and the mitochondrial network, cells were then washed three times using 1× PBS, before addition of complete medium containing 2.5 µl ml$^{-1}$ Quant-iT PicoGreen (Thermo Fisher, P7589) for 30 min. Cells were then washed three times in complete medium, then incubated in 10 nM MitoTracker Deep Red FM (Invitrogen, M22526) for 30 min before imaging. For labelling late endosomes and lysosomes, cells were incubated in 100 nM LysoTracker Red FM (Invitrogen L7528) for 30 min at the same time as MitoTracker staining. For labelling lysosomes, cells were incubated for 6 h with 0.1 mg ml$^{-1}$ Dextran Alex Fluor 555, 10,000 MW anionic, fixable (Invitrogen, D34679) then chased for 18 h in fresh medium. Before imaging, HEPES buffer was added to a final concentration of 25 mM. Images were acquired using an Olympus IX83 confocal microscope containing a spinning disk system (Andor/Yokogawa CSU-X) using an Olympus Olympus UPLANSAPO Å-100/1.40 NA oil objective and Andor Neo sCMOS camera and Meta-Morph software. Cells were imaged at 37 °C in a humidified chamber controlled by an INU TOKAI HIT system. For videos, a rate of 1 frame per 1.5 s was typically used. Acquisition parameters were controlled to avoid bleed-through across channels and remained consistent across experimental conditions.

## Image processing and quantification

Images were processed using Fiji ImageJ software (RRID: SCR_003070). For counting the number of cytosolic mtDNA foci, a maximum intensity projection was generated, before automated macros enabled unbiased quantification. First, a mask of the nucleus (DAPI signal), mitochondria (TOMM40 or OCT–GFP signal) and mtDNA (DNA signal) were generated. The mtDNA foci outside mitochondria were calculated by subtracting the nuclear and mitochondrial masks from the mtDNA mask, leaving only non-nuclear and non-mitochondrial foci. Cell perimeter was drawn by hand for each cell, and foci within the perimeter were counted. The number of mtDNA$^+$ lysosomes was counted by eye, after manually thresholding the lysosome and DNA channels. For these experiments, the mitochondrial network was also taken into consideration to prevent false positive results. A single z-plane was taken into consideration for these experiments to ensure colocalization. Quantification of lysosomal damage was also carried out by eye. Cells with extensive lysosomal damage (ten or more galectin-3 foci per cell) were considered GAL3-positive cells. TOM40$^-$ complex I$^+$mtDNA$^+$ vesicles were identified manually, as were GAL3–RFP and mtDNA$^+$GAL3–RFP$^+$ foci. Lysosome size was measured by LAMP1 immunofluorescence using an automated macro. LAMP1 signal was masked and those >0.1 µm$^2$ were counted.

## RNA isolation and qPCR

Total cellular RNA was prepared using Nucleospin (Takara, 740955.50) as described by the manufacture, including treatment with DNase for 15 min. Reverse transcription using the High Capacity cDNA Reverse Transcription kit (Life Technologies) was completed with random primers according to manufacturer instructions. Gene expression assays were designed with the Universal Probe Library from Roche. qPCR reactions used 5–25 ng of cDNA, the TaqMan Advanced Fast Universal PCR Master Mix (Life Technologies), 2 μM of each primer and 1 μM of the corresponding UPL probe. Viia7 qPCR machine (Life Technologies) was used to detect amplification, with a programme involving an initial step of 95 °C for 3 min, followed by 40 cycles at 95 °C (5 s) and 60 °C (30 s). Reactions were run in duplicate and experiments were repeated three times independently. Relative abundance was determined relative to an endogenous control genes (TERT and PPIA) using the ΔΔCT approach (ΔCT = Cttarget − CtCTRL) before comparison with a calibrator sample for example an untreated control (ΔΔCT = ΔCtSample − ΔCtCalibrator). Relative expression was then calculated using the formula RQ = 2 − ΔΔCT. Reverse transcription and qPCR was performed at the Institute for Research in Immunology and Cancer, Montreal, Canada. The sequences of the oligomers used are provided in Supplementary Table 5.

## PCR validation of Rho0 cells

A total of $1 \times 10^6$ cells per condition was collected and lysed in 0.5 ml of 50 mM Tris-HCl, pH 8.0, 100 mM EDTA, pH 8.0, 100 mM NaCl, 1% SDS and 0.5 mg ml$^{-1}$ proteinase K (Sigma, P2308). Following 1 h of incubation at 55 °C, the lysate was extracted with equal volumes of phenol:chloroform:isoamyl alcohol (25:24:1, Sigma, 77617). DNA from 500 μl of the aqueous phase was precipitated with 1 ml of 100% ethanol and washed once with 70% ethanol. Then, 50 ng of the resulting DNA was used for PCR (30 cycles of 96 °C:59 °C:72 °C) using SYBR Green Supermix (Bio-Rad, 1725271). The sequences of the oligomers used are provided in Supplementary Table 5.

## Transmission electron microscopy

Cells were plated on glass coverslips and treated with adenovirus (20 p.f.u. per cell) for 24 h. Cells were fixed with 2.5% glutaraldehyde, 0.1 M sodium cacodylate and 0.1% CaCl$_2$ for 1–2 days at 4 °C. Coverslips were treated with 1% OSO$_4$ and 1.5% ferrocyanide for 45 min at 4 °C. Following three washes with dH$_2$O, coverslips were incubated with 5% uranyl acetate (in dH$_2$O) for 45 min. Coverslips were then washed extensively with ethanol and acetone before embedding in Spurr epoxy resin (30 g ERL-4221 3,4-epoxycyclohexanemethyl 3,4-epoxycyclohexanecarboxylate, 18 g direct exposy resin, 78 g nonenyl succinic anhydride (NSA) and 1.35 ml 2-dimethylaminoethanol (DMAE) all from Mecalab). Samples were sliced and mounted on grids by McGill university Facility for Electron Microscopy Research and images were acquired with FEI Tecnai G2 Spirit Twin 120 kV Cryo-TEM.

## Immuno-isolation of lysosomes

U2OS cells stably expressing TMEM192-3xHA[106] were seeded overnight in 10-cm dishes (two dishes per condition, $2 \times 10^6$ cells per dish). U2OS cells expressing RFP–GAL3 were also set up as a control not expressing TMEM192-3xHA. Lyso-IP U2OS cells were infected with 30 p.f.u. per cell adenovirus encoding ΔRING–Flag or MAPL–Flag. After 6 h, virus was removed, and cells were washed three times in 1× PBS before addition of fresh medium. Cells were left for a further 18 h, before collection in PBS. Lyso-IP was undertaken as previously described[107]. In brief, Cells were centrifuged once, then resuspended in 1 ml KPBS (136 mM KCl, 10 mM KH$_2$PO$_4$ and 50 mM sucrose, pH 7.2). Then, 2.5% input was isolated and resuspended in 1× Laemmli buffer. The remaining samples were lysed using 30 strokes in a Teflon-glass homogenizer. The homogenates were then centrifuged at 1,000$g$ for 5 min at 4 °C. The supernatant was collected and re-centrifuged at 8,000$g$ for 5 min at 4 °C. The 8,000$g$ supernatant was pre-absorbed with 75 μl of Protein G magnetic beads at 4 °C for 20 min with rotation. The pre-absorbed supernatant was added to 75 μl anti-HA magnetic beads (prepared by washing three times in KPBS) and placed on rotation for 20 min at 4 °C. The beads were then washed three times in KPBS, before elution in 2× Laemmli buffer by heating to 95 °C for 10 min. Samples were separated via SDS–PAGE (4–14% gel) before immunoblotting.

## Isolation of cytosolic fractions for mtDNA qPCR

A total of $2 \times 10^6$ U2OS cells per experimental condition were treated with Ad–rtTA or Ad–MAPL for 24 h. Treated cells were collected and washed once with PBS. Each cell pellet was resuspended in 500 μl digitonin lysis buffer (50 mM HEPES, pH 7.4, 150 mM NaCl, 18 μg ml$^{-1}$ digitonin and 1× protease inhibitors cocktail CLAAAP, Roche 493132001) and incubated on a tube rotator for 10 min at 4 °C. Samples were centrifuged at 950$g$ for 5 min at 4 °C. The supernatant was transferred to a new 1.5-ml tube and centrifuged at 17,000$g$ for 5 min at 4 °C. The resulting supernatant was incubated with 4 μl of 20 mg ml$^{-1}$ proteinase K at 55 °C for 1 h. Equal volumes of phenol:chloroform:isoamyl alcohol (Sigma, 77617) were added and vortexed vigorously. Samples were centrifuged 21,000$g$ for 5 min, room temperature and the top aqueous phase was collected. DNA was precipitated by add 2.5× volumes of 100% ethanol and incubated at −20 °C overnight. DNA was centrifuged for 20 min at max speed, 4 °C followed by two washes with 95% ethanol. DNA concentration was determined using nanodrop. Fractions were assessed by immunoblot to confirm absence of contamination. Cytosolic mtDNA levels were determined from qPCR data using the approach described in Basic Protocol 2 (ref. [108]).

## RNA-seq

BMDM samples prepared for RNA-seq were first analysed by western blot and ELISA to confirm the LPS and IFNγ treatment-induced expression of NLRP3 and IL-1β at 4.5 and 24 h in control cells. IL-1β Duoset ELISA (R&D Systems DY401-05) was used to determine the release of IL-1β in the extracellular supernatant of BMDMs treated for RNA-seq as described by the manufacture. Total cellular RNA was prepared using Nucleospin (Takara, 740955.50) as described by the manufacture. The mRNA library was prepared using PolyA selection (Dynabeads Thermo Fisher Scientific) followed by KAPA RNA HyperPrep protocol, using xGene Dual Index UMI Adaptor (IDT). The 2100 Bioanalyser system was used to measure RNA integrity. Library was prepared using PolyA selection (Dynabeads, Thermo Fisher Scientific) before KAPA RNA HyperPrep using xGen Dual Index UMI Adaptors (IDT). The Nextseq500-1 Flowcell High Output was used for sequencing, with 75 cycles single end reads (maximum 1 × 85 nucleotides). Library preparation and sequencing was completed in the Institute for Research in Immunology and Cancer, Montreal, Canada. For analysis, Galaxy platform[109] and RStudio v.2021.09.1 were used. Reads were filtered to remove adaptor sequences, contamination and low-quality reads using Trimmomatic (http://www.usadellab.org/cms/?page=trimmomatic, version: v0.39, RRID: SCR_011848) (parameters ILLUMINACLIP 2:30:10 (AGATCGGAAGAGCACACGTCTGAACTCCAGTCAC) LEADING:15 TRAILING:15 MINLEN:20) and FastaQC[110]. Reads were mapped using HISAT2 (ref. [111]) to reference mouse genome build mm10, and counted using featureCounts[112]. Differential expression analysis was carried out using DeSeq2 (ref. [113]). Heatmaps were generated using ComplexHeatmaps[114], PCA plot values were computed in R and plotted using Prism v.9 and scatter-plots were also plotted using Prism v.9. For visualization of read alignment, BAM indexes were generated using SAMtools[115] and visualized using WashU Epigenome Browser (http://epigenomegateway.wustl.edu/, RRID: SCR_006208)[116].

## Statistics and reproducibility

Statistical analysis was performed on independent repeat values (and not technical repeats) using GraphPad Prism software (v.10.5.0 (673)).

All experiments were performed independently at least twice, as indicated in the figure legends, and quantifications and/or representative images are provided. No statistical method was used to predetermine the sample size. No data were excluded from the analyses. All numerical source data, statistics about the performed statistical test and unprocessed blots are provided in the source data. The experiments were not randomized. The investigators were not blinded to allocation during experiments and outcome assessment. All numerical source data, statistics about the performed statistical test and unprocessed blots are provided in the source data.

## Reporting summary

Further information on research design is available in the Nature Portfolio Reporting Summary linked to this article.

## Data availability

The data, code and key laboratory materials used and generated in this study are listed in a key resource table alongside their persistent identifiers on Zenodo at https://doi.org/10.5281/zenodo.17180353 (ref. 117). Protocols for this manuscript can be found at https://doi.org/10.17504/protocols.io.8epv5x266g1b/v1. (ref. 118). Raw sequencing data are deposited in the Gene Expression Omnibus under accession code GSE301127. Source data are provided with this paper.

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

## Acknowledgements

An earlier version of this manuscript was posted to *BioRxiv* on 12 September 2023 (https://doi.org/10.1101/2023.09.11.557213). We thank all members of the Desjardins Aligning Science Across Parkinson's (ASAP) team and members of the laboratory for critical comments throughout the course of this study. The study is funded by the joint efforts of The Michael J. Fox Foundation for Parkinson's Research (MJFF) and the ASAP initiative. MJFF administers the grant ASAP-000525 on behalf of ASAP and itself. For the purpose of open access, the author has applied a CC BY public copyright license to all Author-Accepted Manuscripts arising from this submission. We acknowledge funding support from ASAP (M.D. and H.M.M.); an EMBO Postdoctoral Fellowship ALTF 971-2021 (J.J.C.); the Sigrid Juselius Fellowship, Redpoll Postdoctoral Fellowship, Finnish Cultural Foundation Fellowship and Maud Kuistila Memorial Foundation Fellowship (O.I.).

## Author contributions

Conceptualization: M.N., M.D. and H.M.M. Methodology: M.N., J.J.C., O.I., V.G., A.J., S.H., G.M., C.T.R., A.M. and H.M.M. Investigation: M.N., J.J.C., O.I. and A.J. Visualization: M.N., J.J.C., O.I. and H.M.M. Funding acquisition: H.M.M. Project administration: H.M.M. Supervision: H.M.M. Writing – original draft: J.J.C. and H.M.M. Writing – review & editing: M.N., J.J.C., O.I., S.H., A.M., M.D. and H.M.M.

## Competing interests

The authors declare no competing interests.

## Additional information

**Extended data** is available for this paper at https://doi.org/10.1038/s41556-025-01774-y.

**Correspondence and requests for materials** should be addressed to Heidi M. McBride.

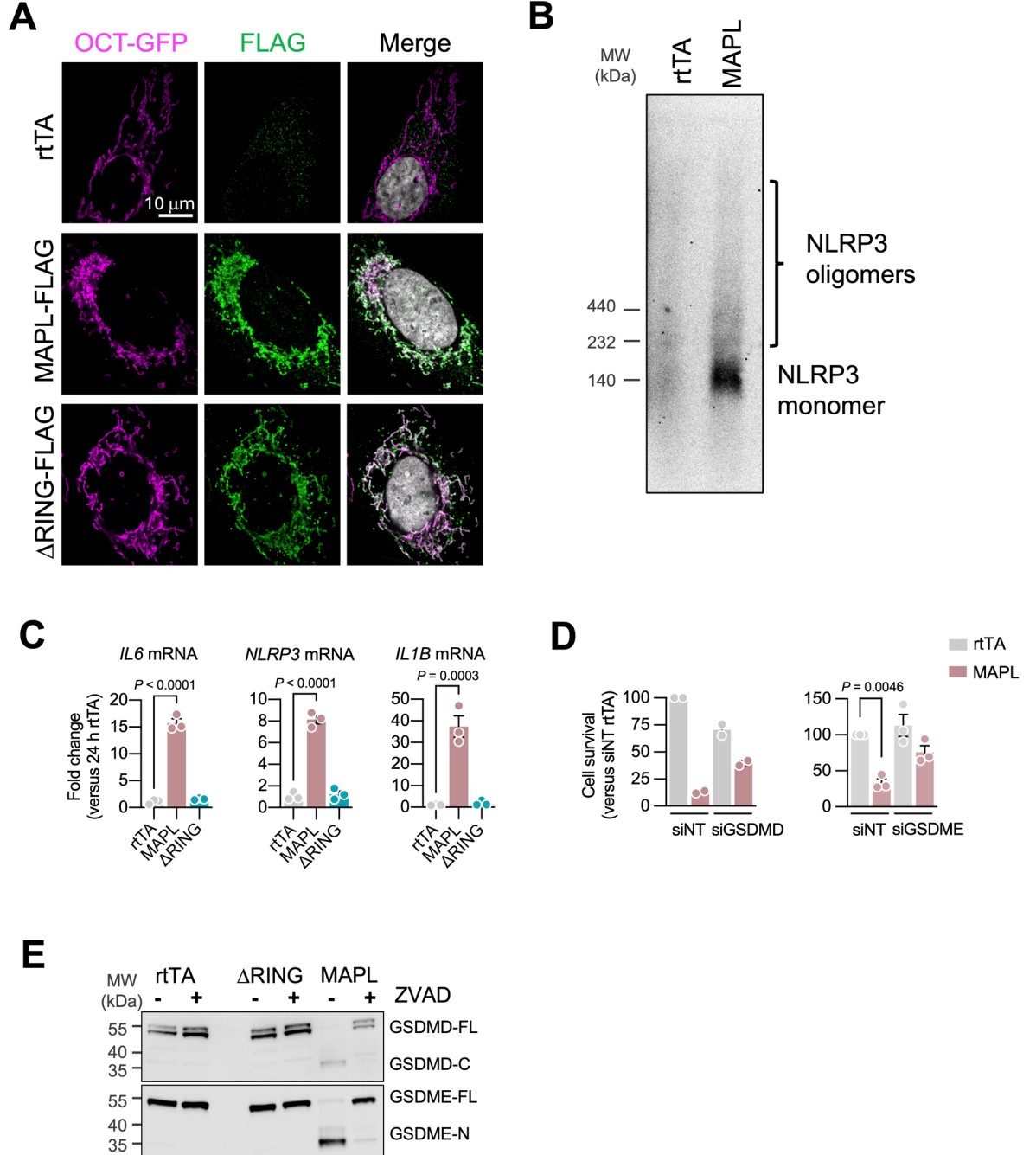

**Extended Data Fig. 1 | Confirmation of MAPL expression on mitochondria and depletion of GSDMs effect.** (**a**) Immunofluorescence using anti-FLAG antibody showing delivery of MAPL–FLAG and MAPL-ΔRING–FLAG to mitochondria in U2OS^OCT–GFP 24 h after expression. (**b**) Blue native analysis of U2OS^neo cells after 24-h expression of MAPL. (**c**) qRT–PCR assessment of IL-6, NLRP3, and IL1B mRNA levels in U2OS^neo (24 h after expression) (48 h after expression). N = 3 independent experiments. (**d**) A second death assay using CellTitre^TM Glo

Assay in U2OS after 3-day knockdown of GSDMD or GSDME then 48 h rtTA or MAPL expression. Non-targeting siRNA (siNT) was used as a control. N = 2–3 independent experiments. (**e**) Immunoblot analysis of GSDMD and GSDME cleavage following 48 h expression of MAPL or MAPL-ΔRING in the presence or absence of pan-caspase inhibitor ZVAD. For all graphs, bars represent means of independent repeats ± SEM, with comparisons by One-way ANOVA.

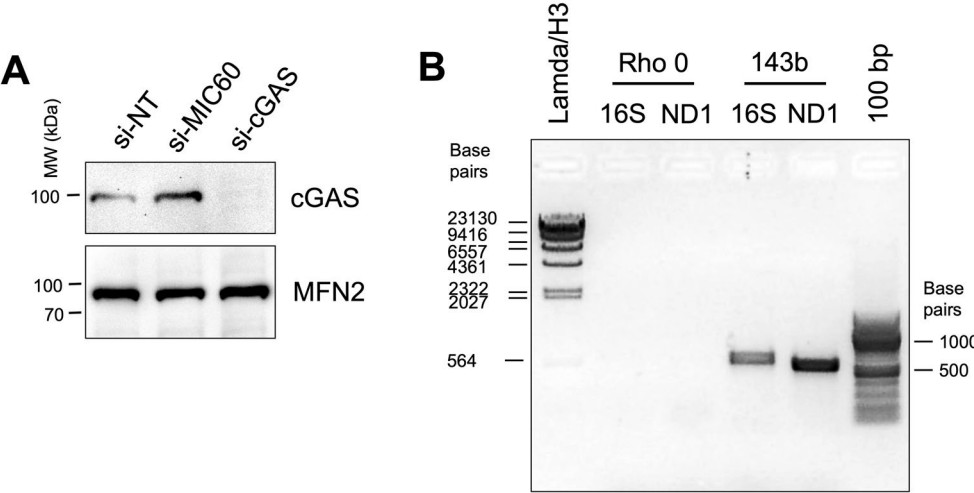

**Extended Data Fig. 2 | Confirmation of cGAS KD and validation of loss of mtDNA in Rho0 cells.** (**a**) Confirmation of cGAS knockdown by immunoblot. (**b**) Validation of Rho0 cells PCR analysis of wild-type (WT) 143b cells, and Rho0 143b cells, validating loss of mtDNA in Rho0 cells.

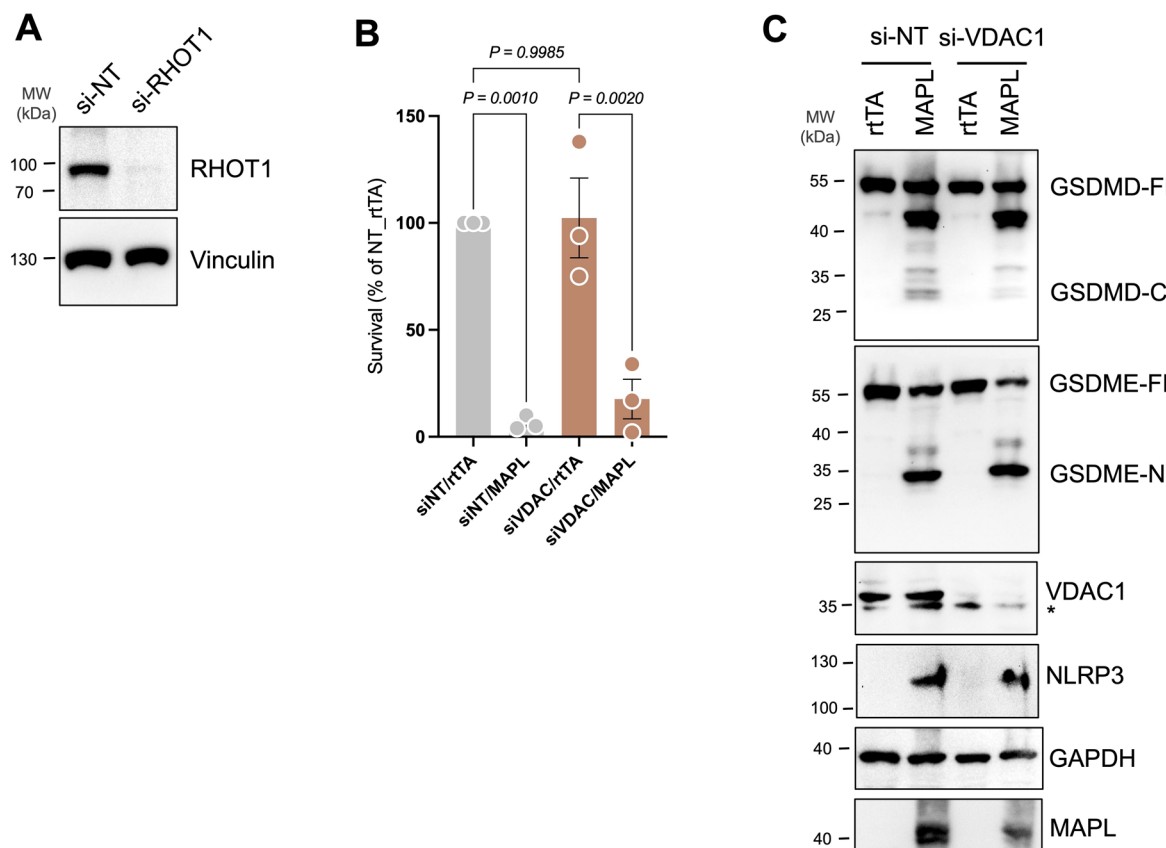

**Extended Data Fig. 3 | Validation of MIRO1 KD and VDAC KD.** (**a**) Confirmation of MIRO1 knockdown (**b**) CellTitre Glo Assay measuring cell death in control U2OS (NT) or U2OS depleted of VDAC1 (VDAC) after expression of rtTA, MAPL for 48 h. N = 3 independent experiments. P values are relative to rtTA.

(**c**) Immunoblot analysis of representative experiment validating efficacy of VDAC1 knockdown. There is no impact on a series of pyroptotic markers upon loss of VDAC1. For graph, bars represent means of independent repeats ± SEM, with comparisons by One-way ANOVA.

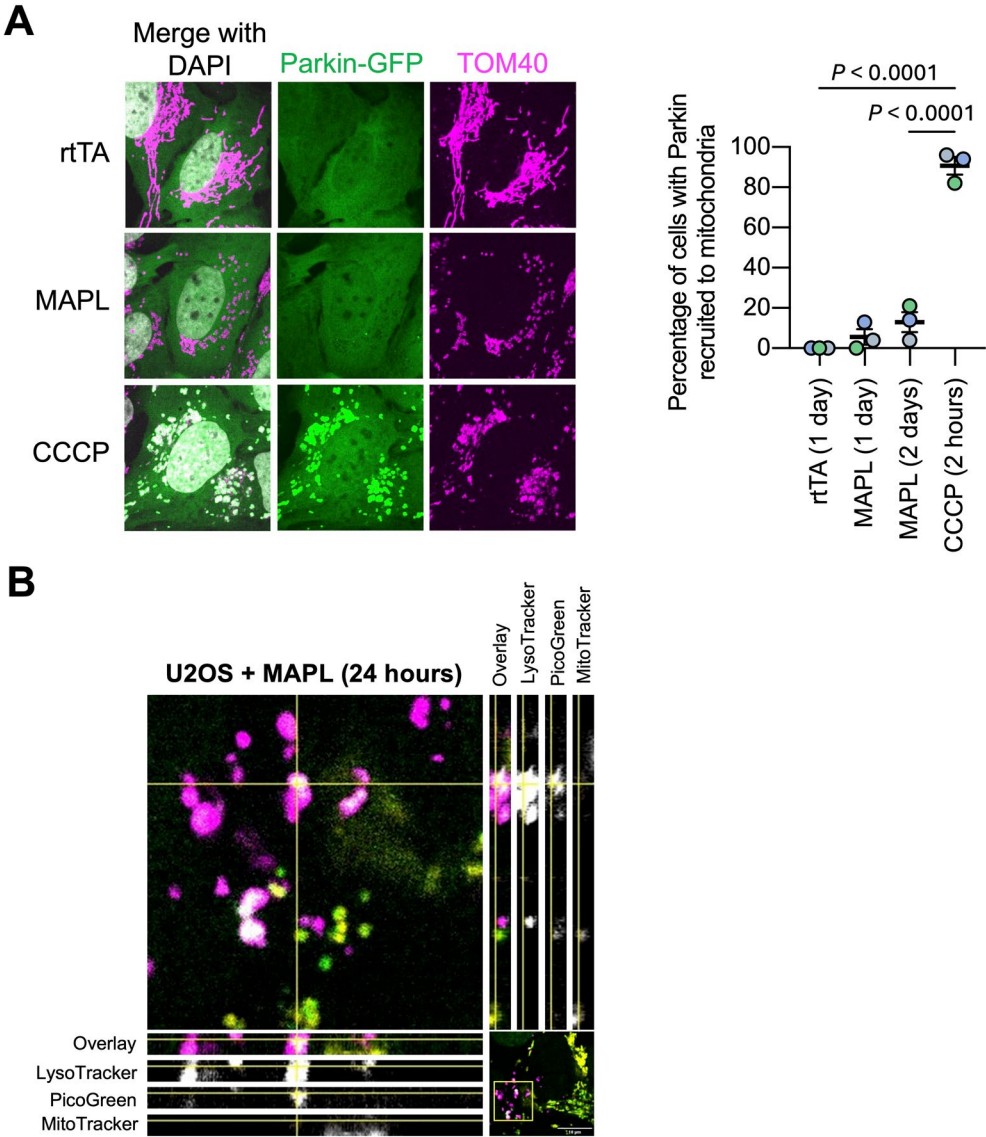

**Extended Data Fig. 4 | MAPL expression does not induce Parkin recruitment to mitochondria. (a)** MAPL overexpression does not induce Parkin-mediated mitophagy rtTA or MAPL was expressed in U2OS cells expressing Parkin-GFP. 100 μM CCCP treatment was used as a positive control for Parkin-GFP recruitment to mitochondria. Quantification of three independent experiments is shown. **(b)** 3-D analysis of Z-stack images showing PicoGreen signal within lysosomal lumen. MAPL was expressed in U2OS cells for 24 h before addition of PicoGreen (DNA), MitoTracker (mitochondria) and LysoTracker (lysosomes). Z-stacks (0.2 mm) were analysed in 3-D using FIJI Image J software, enabling visualization of DNA within the lysosomal lumen.

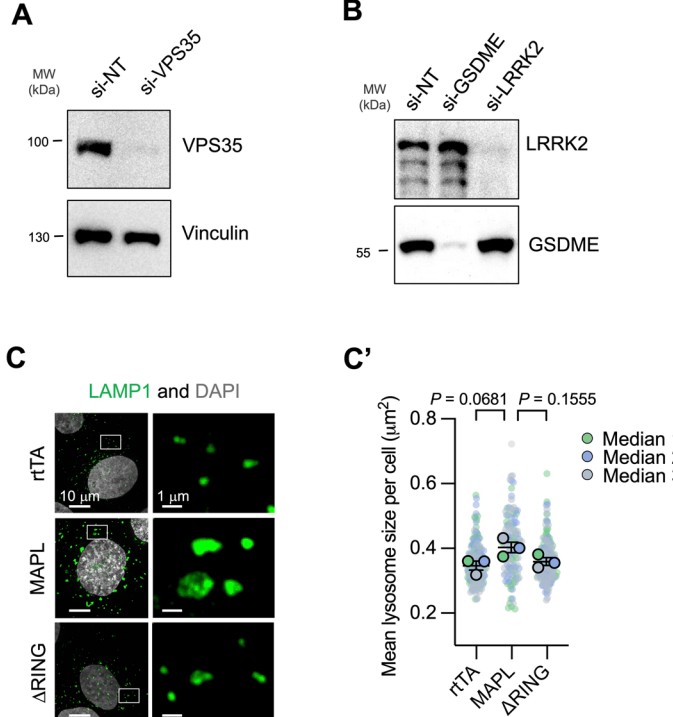

**Extended Data Fig. 5 | Effect of depletion of VPS35, LRRK2 and GSDME on lysosome size. (a, b)** Immunoblotting confirming knockdown of target genes in U2OS cells (**a**) VPS35, (**b**) GSDME and LRRK2 using siRNA. Non-targeting siRNA (siNT) was used as a control (**c**) MAPL overexpression causes a RING-dependent increase in lysosomal size rtTA, MAPL or ΔRING were expressed in U2OS^OCT–GFP cells for 48 h before assessment of lysosomal size. N = 3 independent experiments.

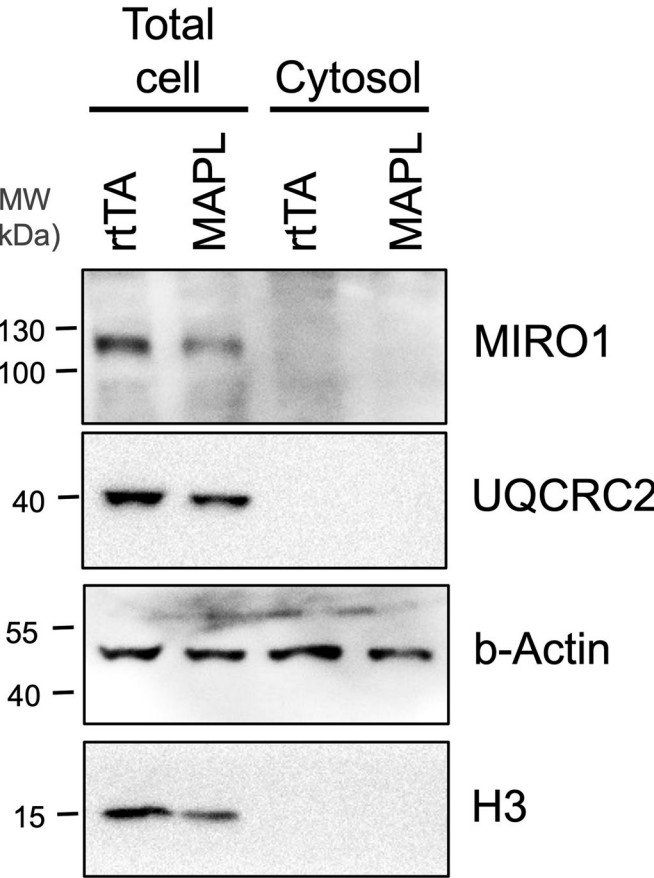

**Extended Data Fig. 6 | Validation of cytosolic fractions for mtDNA qPCR.** Validation of cytosolic fractions by immunoblotting Immunoblot analysis of cytosolic U2OS fractions used for analysis of cytosolic mtDNA by qPCR (Fig. 6h). N = 3 independent experiments.

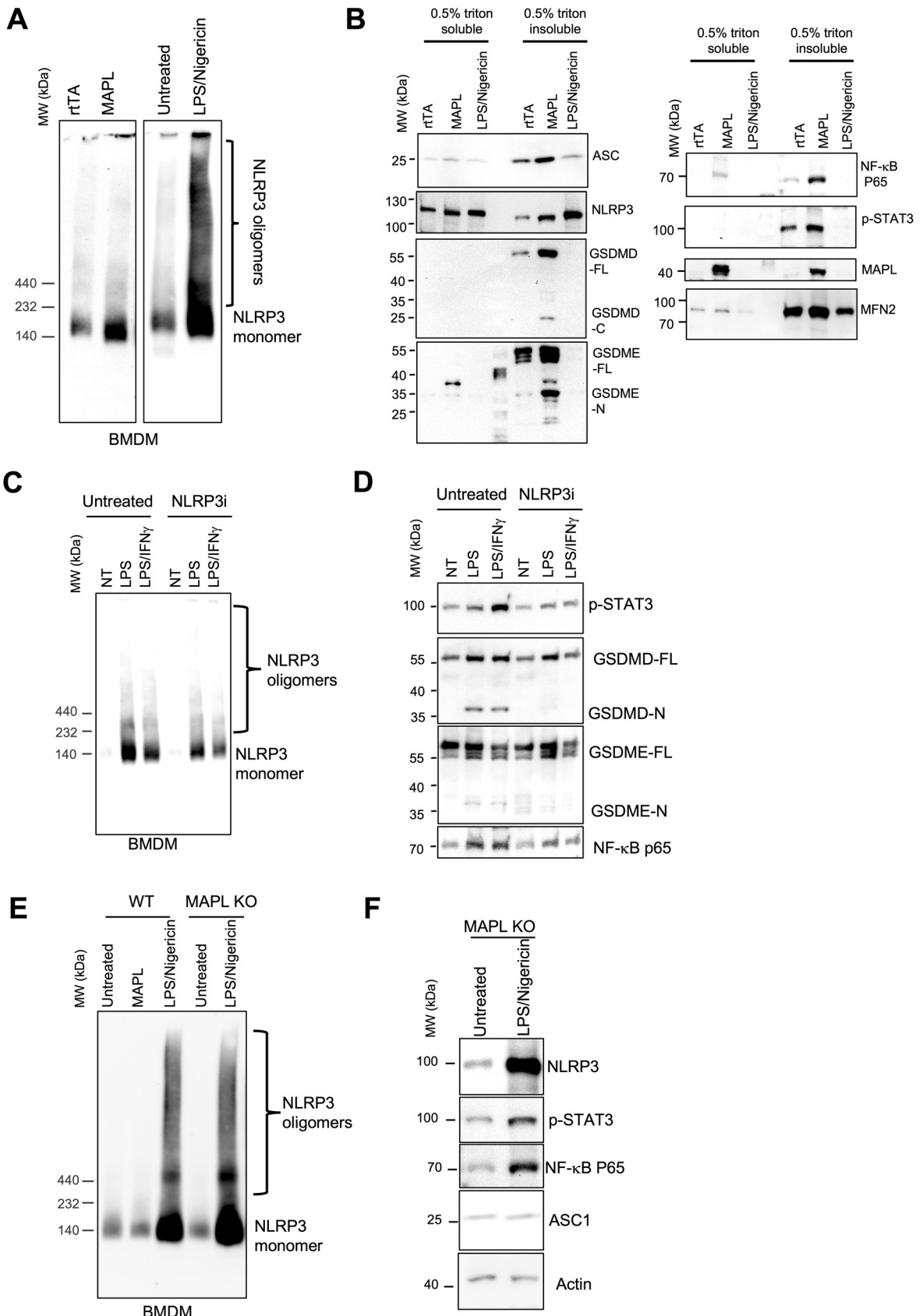

**Extended Data Fig. 7 | See next page for caption.**

**Extended Data Fig. 7 | Nigericin and LPS/IFNγ induced pyroptotic signals.**
(**a**) LPS/Nigericin (1 h) or MAPL treated BMDM cells (48 h) were solubilized with 0.5% digitonin and processed for BN-PAGE. Immunoblot against NLRP3 reveal oligomers induced by nigericin but not upon MAPL expression. (**b**) As in (**a**) but cells were solubilized in 0.5% triton X-100 and detergent-soluble and detergent-insoluble material processed for Western blot analysis of ASC, NLRP3, NF-kB and p-STAT3. (**c**) NLRP3 inhibitor blocks pyroptosis in control BMDM. BN-PAGE analysis showed a reduction of NLRP3 oligomerization upon LPS/IFNγ treatment in the presence of NLRP3 inhibitor (NLRP3i, MCC950). (**d**) As in (**c**) where Western blot analysis shows effective inhibition of p-STAT3, NF-κB, and GSDM cleavage using the NLRP3 inhibitor. (**e**) Nigericin induced the oligomerization of NLRP3 as analysed by BN-PAGE, which was unaltered in MAPL KO BMDM (**f**) Additional pyroptotic markers p-STAT3 and NF-κB were also unaltered in MAPL KO.

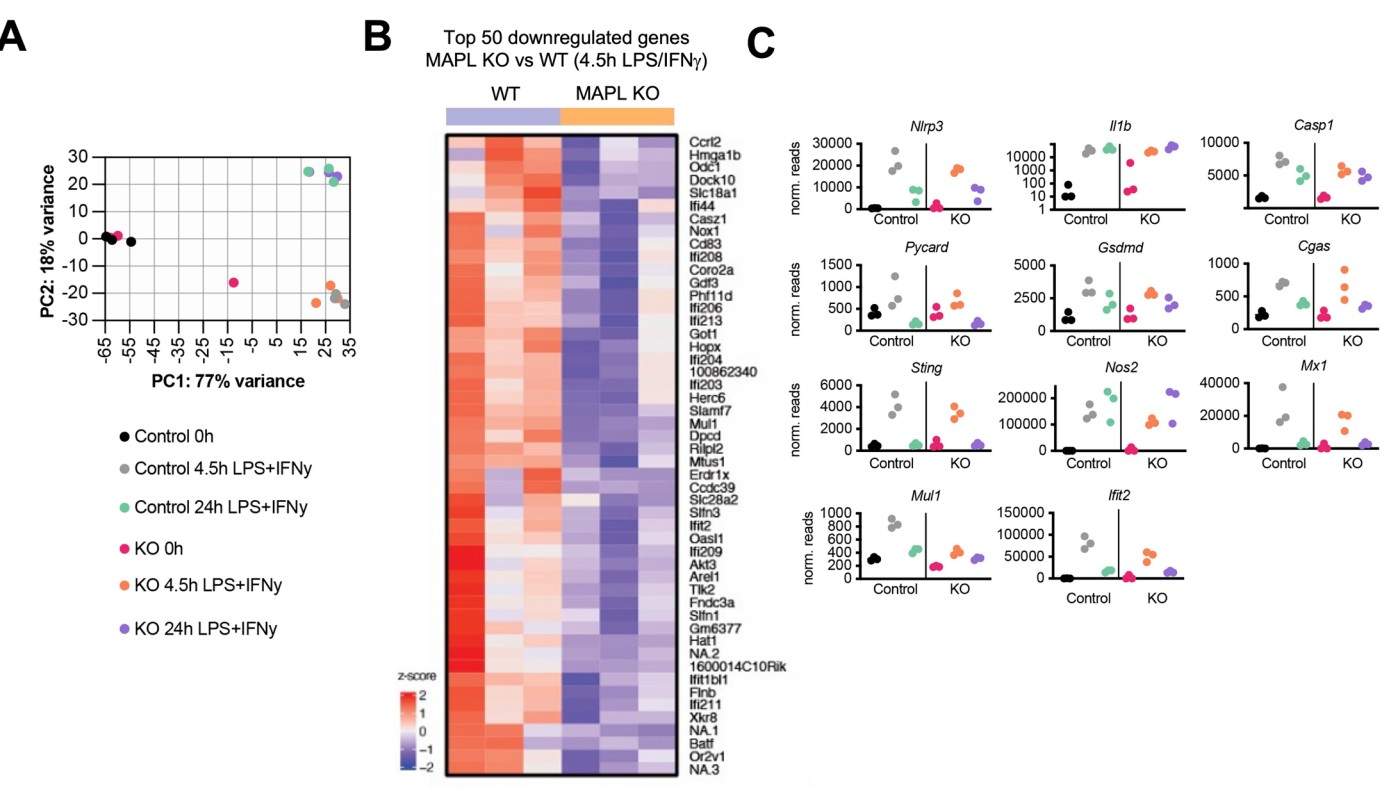

**B** Top 50 downregulated genes MAPL KO vs WT (4.5h LPS/IFNγ)

**Extended Data Fig. 8 | RNA-Seq analysis of MAPL KO BMDM. (a)** RNA-Seq of WT and MAPL$^{-/-}$ BMDMs. Principal component analysis of RNA-Seq from wild-type (control) and MAPL$^{-/-}$ BMDMs treated with LPS/IFNg for 0 h, 4.5 h or 24 h. **(b)** RNA-Seq comparing gene expression between WT and MAPL-/- BMDMs treated with LPS and IFNγ for 4.5 h showing the top 50 downregulated genes.

**(c)** Normalized reads for corresponding genes in each condition. **(d)** Visualization of RNA-seq read alignment to Mul1 (MAPL-encoding) for representative Control and MAPL-/- BMDM samples. The yellow highlight indicate exon2, that is flanked by loxP sites and excised in MAPL-/- resulting in a frame-shift, as previously described[4].

# Reporting Summary

## Statistics

For all statistical analyses, confirm that the following items are present in the figure legend, table legend, main text, or Methods section.

| n/a | Confirmed | |
|---|---|---|
| ☐ | ☒ | The exact sample size (*n*) for each experimental group/condition, given as a discrete number and unit of measurement |
| ☐ | ☒ | A statement on whether measurements were taken from distinct samples or whether the same sample was measured repeatedly |
| ☐ | ☒ | The statistical test(s) used AND whether they are one- or two-sided *Only common tests should be described solely by name; describe more complex techniques in the Methods section.* |
| ☐ | ☒ | A description of all covariates tested |
| ☐ | ☒ | A description of any assumptions or corrections, such as tests of normality and adjustment for multiple comparisons |
| ☐ | ☒ | A full description of the statistical parameters including central tendency (e.g. means) or other basic estimates (e.g. regression coefficient) AND variation (e.g. standard deviation) or associated estimates of uncertainty (e.g. confidence intervals) |
| ☐ | ☒ | For null hypothesis testing, the test statistic (e.g. *F*, *t*, *r*) with confidence intervals, effect sizes, degrees of freedom and *P* value noted *Give P values as exact values whenever suitable.* |
| ☒ | ☐ | For Bayesian analysis, information on the choice of priors and Markov chain Monte Carlo settings |
| ☒ | ☐ | For hierarchical and complex designs, identification of the appropriate level for tests and full reporting of outcomes |
| ☒ | ☐ | Estimates of effect sizes (e.g. Cohen's *d*, Pearson's *r*), indicating how they were calculated |

*Our web collection on statistics for biologists contains articles on many of the points above.*

## Software and code

Policy information about availability of computer code

| Data collection | Methods outline all softwares used with relevant citations. Confocal images and analysis was done with MetaMorph software (MetaMorph Microscopy Automation and Image Analysis Software (RRID:SCR_002368). RNAseq reads were mapped using HISAT2 to reference mouse genome build mm10, and counted using featureCounts. Differential expression analysis was done using DeSeq2. Heatmaps were generated using ComplexHeatmaps, PCA plot values were computed in R and plotted using Prism9, and scatter dot plots were also plotted using Prism 9. For visualisation of read alignment, BAM indexes were generated using using SAMtools, and visualised using WashU Epigenome Browser. |
|---|---|
| Data analysis | Images were processed using FIJI Image J software (ImageJ (RRID:SCR_003070). For the genome-wide screen, sequencing reads were mapped to the library using xcalibr (https://www.thermofisher.com/order/catalog/product/OPTON-30965) and counts were then analyzed with MAGeCK (https://sourceforge.net/p/mageck/wiki/Home/, version 0.5.8) using the Robust Rank Aggregation (RRA) algorithm to identify genes whose perturbation (knockout or overexpression) primarily enhanced fitness in the MAPL overexpressing group but not the control group. For RNAseq, calculations were computed using Rstudio (RRID:SCR_000432, https://posit.co/, version 2021.09.1), and graphs plotted using Prism10 (GraphPad Prism, RRID:SCR_002798, http://www.graphpad.com/) |

For manuscripts utilizing custom algorithms or software that are central to the research but not yet described in published literature, software must be made available to editors and reviewers. We strongly encourage code deposition in a community repository (e.g. GitHub). See the Nature Portfolio guidelines for submitting code & software for further information.

## Data

Policy information about availability of data

All manuscripts must include a data availability statement. This statement should provide the following information, where applicable:
- Accession codes, unique identifiers, or web links for publicly available datasets
- A description of any restrictions on data availability
- For clinical datasets or third party data, please ensure that the statement adheres to our policy

Data availability statement is included in manuscript. RNAseq datasets are uploaded to GEO repository, and has been released. https://www.ncbi.nlm.nih.gov/geo/query/acc.cgi?acc=GSE301127. Data and materials are also deposited at https://zenodo.org/records/16387131 and has been released to the public

## Research involving human participants, their data, or biological material

Policy information about studies with human participants or human data. See also policy information about sex, gender (identity/presentation), and sexual orientation and race, ethnicity and racism.

| | |
|---|---|
| Reporting on sex and gender | Gender is not relevant. Manuscript is primarily using cultured cell lines that include both male and female lines (U2OS - female, BMDM were taken from both male and female mice. |
| Reporting on race, ethnicity, or other socially relevant groupings | N/A |
| Population characteristics | N/A |
| Recruitment | N/A |
| Ethics oversight | McGill Animal Care protocols approved for mouse colonies used in the generation of primary BMDM. |

Note that full information on the approval of the study protocol must also be provided in the manuscript.

# Field-specific reporting

Please select the one below that is the best fit for your research. If you are not sure, read the appropriate sections before making your selection.

☒ Life sciences  ☐ Behavioural & social sciences  ☐ Ecological, evolutionary & environmental sciences

For a reference copy of the document with all sections, see nature.com/documents/nr-reporting-summary-flat.pdf

# Life sciences study design

All studies must disclose on these points even when the disclosure is negative.

| | |
|---|---|
| Sample size | No statistical method was used to predetermine sample size. At least three independent experiments were performed to test for statistical significant differences. Minimum of 30 cells imaged per condition, in triplicate with at least 3 biological replicates. The design achieved sufficient size for statistical analysis as reported in previous publications within the field. |
| Data exclusions | No experimental data points were excluded. Only data from experiments with appropriate negative/positive controls is included in this study. |
| Replication | At least three independent experiments were performed to test for statistical significance and verify reproducibility of the experiments. We can confirm that all attempts at replication were successful. For videos, microscopy images and western blots only representative data is shown. |
| Randomization | No randomization was performed in this study. The experimental set-up included clearly defined groups/ conditions for comparison. To avoid human bias, automated software with identical settings between conditions was used for quantifications. Image analysis was performed randomized and blinded. |
| Blinding | Not necessary for most parts of this study due to automated quantification by a software with identical parameter settings. Image analysis was performed randomized and blinded. |

# Reporting for specific materials, systems and methods

We require information from authors about some types of materials, experimental systems and methods used in many studies. Here, indicate whether each material, system or method listed is relevant to your study. If you are not sure if a list item applies to your research, read the appropriate section before selecting a response.

## Materials & experimental systems

| n/a | Involved in the study |
|:---:|---|
| ☐ | ☒ Antibodies |
| ☐ | ☒ Eukaryotic cell lines |
| ☒ | ☐ Palaeontology and archaeology |
| ☐ | ☒ Animals and other organisms |
| ☒ | ☐ Clinical data |
| ☒ | ☐ Dual use research of concern |
| ☒ | ☐ Plants |

## Methods

| n/a | Involved in the study |
|:---:|---|
| ☒ | ☐ ChIP-seq |
| ☒ | ☐ Flow cytometry |
| ☒ | ☐ MRI-based neuroimaging |

# Antibodies

| Antibodies used | Primary antibody Company Catalogue Number/RRID Dilution |
|---|---|
| | Actin Sigma A2228/RRID:AB_476697 1:2000 |
| | AIF(E1) Santa Cruz Sc-13116/ RRID:AB_626654 1:1000 |
| | Caspase 1 Adipogen AG-20B-0042-C100/ 1:500 |
| | Cleaved caspase 3 Cell Signaling 96661S/RRID:AB_2341188 1:1000 |
| | Cleaved caspase 7 Cell Signaling 9491/ RRID:AB_2068144 1:1000 |
| | cGAS Cell signaling 15102/RRID:AB_2732795 1:1000 |
| | FLAG Sigma F1804/RRID:AB_262044 1:1000 |
| | GSDMD Abcam ab210070 /RRID:AB_2893325 1:1000 |
| | GSDME Abcam ab215191/RRID:AB_2737000 1:1000 |
| | Histone 3 Abclonal A2348/RRID:AB_2737000 1:1000 |
| | HA Sigma H9658/RRID:AB_260092 1:1000 |
| | IL1b Bio-Techne AB-401-NA/ 1:1000 |
| | LAMP1 Cell Signaling 9091/RRID:AB_2687579 1:1000 |
| | LRRK2 Abcam ab133474/RRID:AB_2713963 1:1000 |
| | MAPL Sigma HPA017681/RRID:AB_1848699 1:1000 |
| | MFN2(XX-1) Santa Cruz Sc-100560/RRID:AB_2235195 1:1000 |
| | NLRP3 Cell Signaling 15101/RRID:AB_2722591 1:1000 |
| | NF-kB p65 Abcam ab16502/RRID:AB_443394 1:1000 |
| | PMP70 Abcam ab3421/RRID:AB_2219901 1:2000 |
| | RHOT1 Sigma HPA010687/RRID:AB_1079813 1:1000 |
| | p-STAT3 Abclonal AP0474/RRID:AB_2771567 1:1000 |
| | TOM20 Sigma HPA010687/RRID:AB_1080326 1:1000 |
| | UQCR2 Proteintech 83667-2/ RRID:AB_3671273 1:1000 |
| | Vinculin Sigma V4505/RRID:AB_477617 1:1000 |
| | VPS35 Abnova H00055737/RRID:AB_566269 1:1000 |
| | |
| | Secondary antibody (IF) Company Catalogue Number Dilution |
| | HRP-anti-mouse Cytiva NA931/RRID:AB_772210 1:5000 |
| | HRP-anti-rabbit Cytiva NA934/RRID:AB_772206 1:5000 |
| | |
| | Primary antibody Company Catalogue Number/RRID Dilution |
| | Actin Sigma A2228/RRID:AB_476697 1:2000 |
| | AIF(E1) Santa Cruz Sc-13116/ RRID:AB_626654 1:1000 |
| | Caspase 1 Adipogen AG-20B-0042-C100/ 1:500 |
| | Cleaved caspase 3 Cell Signaling 96661S/RRID:AB_2341188 1:1000 |
| | Cleaved caspase 7 Cell Signaling 9491/ RRID:AB_2068144 1:1000 |
| | cGAS Cell signaling 15102/RRID:AB_2732795 1:1000 |
| | FLAG Sigma F1804/RRID:AB_262044 1:1000 |
| | GSDMD Abcam ab210070 /RRID:AB_2893325 1:1000 |
| | GSDME Abcam ab215191/RRID:AB_2737000 1:1000 |
| | Histone 3 Abclonal A2348/RRID:AB_2737000 1:1000 |
| | HA Sigma H9658/RRID:AB_260092 1:1000 |
| | IL1b Bio-Techne AB-401-NA/ 1:1000 |
| | LAMP1 Cell Signaling 9091/RRID:AB_2687579 1:1000 |
| | LRRK2 Abcam ab133474/RRID:AB_2713963 1:1000 |
| | MAPL Sigma HPA017681/RRID:AB_1848699 1:1000 |
| | MFN2(XX-1) Santa Cruz Sc-100560/RRID:AB_2235195 1:1000 |
| | NLRP3 Cell Signaling 15101/RRID:AB_2722591 1:1000 |
| | NF-kB p65 Abcam ab16502/RRID:AB_443394 1:1000 |
| | PMP70 Abcam ab3421/RRID:AB_2219901 1:2000 |
| | RHOT1 Sigma HPA010687/RRID:AB_1079813 1:1000 |
| | p-STAT3 Abclonal AP0474/RRID:AB_2771567 1:1000 |
| | TOM20 Sigma HPA010687/RRID:AB_1080326 1:1000 |
| | UQCR2 Proteintech 83667-2/ RRID:AB_3671273 1:1000 |
| | Vinculin Sigma V4505/RRID:AB_477617 1:1000 |
| | VPS35 Abnova H00055737/RRID:AB_566269 1:1000 |

Secondary antibody (IF) Company Catalogue Number Dilution
HRP-anti-mouse Cytiva NA931/RRID:AB_772210 1:5000
HRP-anti-rabbit Cytiva NA934/RRID:AB_772206 1:5000

| Validation | Many of the antibodies were validated in our study using silencing or knockout lines, including MAPL, LRRK2, VPS35, VDAC1, MIRO1, MIRO2, GSDMD, GSDME, cGAS. Other antibodies are well established, company validated on their websites, and routinely published |

# Eukaryotic cell lines

Policy information about cell lines and Sex and Gender in Research

| Cell line source(s) | U2OS cells (RRID:CVCL_0042)<br>TMEM192-3xHA (U2OSTMEM192-3xHA) (RRID:Addgene 102930)<br>pmRFP-C1-Galectin-3 (U2OSRFP-Gal3)<br>pcDNA3.1(U2OSneo) (Thermo Fisher, V79020)<br>HUH-7 (RRID:CVCL_0336),<br>143B Human osteosarcoma cell line (ATCC CRL-8303, gift from E. Shoubridge, McGill University)<br>BMK and BMK Bax/Bak-/- (Eileen White, see PMID: 12242152) |
| Authentication | No additional authentication was performed. |
| Mycoplasma contamination | All cell lines were frequently tested for mycoplasma contamination. No mycoplasma contamination was detected during the duration of this study. |
| Commonly misidentified lines (See ICLAC register) | No ICLAC registered commonly misidentified cell line was used in this study. |

# Animals and other research organisms

Policy information about studies involving animals; ARRIVE guidelines recommended for reporting animal research, and Sex and Gender in Research

| Laboratory animals | MAPL KO on C57BL/6J background, registered as RRID:MGI:7524577<br>LRRK2 KO on C57BL/6J background, registered as .RRID:IMSR_JAX:012444 |
| Wild animals | No wild animals used. All breedings done from heterozygous animals so wild type mice came from littermate controls |
| Reporting on sex | BMDM were isolated from mixed male and female. Experiments on male or female BMDM individually did not change any of the results across the replicates. The manuscript will report exactly which replicates are from which sex BMDM, and when they are pooled together. |
| Field-collected samples | N/A |
| Ethics oversight | Animal experimentation was conducted in accordance with the guidelines of the Canadian Council for Animal Care. Protocols were approved by the Animal Care Committees of McGill University. |

Note that full information on the approval of the study protocol must also be provided in the manuscript.

# Plants

| Seed stocks | N/A |
| Novel plant genotypes | N/A |
| Authentication | N/A |

