## [Peer Review File · Nature Cell Biology]

MAPL regulates gasdermin-mediated release of mtDNA from lysosomes to drive pyroptotic cell death

Corresponding Author: Professor Heidi McBride

Version 0:

Decision Letter:

*Please delete the link to your author homepage if you wish to forward this email to co-authors.

Dear Heidi,

Thank you again for submitting your manuscript, "The mitochondrial ligase MAPL is an inflammatory rheostat that regulates immune signalling and cell death" to Nature Cell Biology and thank you for your patience with the peer review process. The manuscript has now been seen by 3 referees, who are experts in mitochondria and cell death (Referee #1); mtDNA, inflammation (Referee #2); and mitochondria, dynamics (Referee #3). As you will see from their comments (attached below), they found this work of potential interest but have raised substantial concerns, which in our view would need to be addressed with considerable revisions before we can consider publication in Nature Cell Biology.

We have now discussed the referee reports in detail within the editorial team to identify key referee points that should be addressed with priority to bolster the model and conclusions, as opposed to requests that are overruled as being beyond the scope of the current study. To guide the scope of the revisions, I have listed these points below. As you know, our standard revision period is six months, and we are committed to providing a fair and constructive peer-review process, so please feel free to contact me if you would like to discuss any of the referee comments further or if you anticipate any issues or delays addressing the reviews.

You will see that the reviewers had reservations about the strength of the conclusions (in particular regarding the mode of cell death) and the depth of investigation of the proposed pathway. We found their points relevant and valid; their concerns would need to be addressed rigorously experimentally. Reconsideration of the study for this journal and re-engagement of referees will depend on the strength of these revisions. In our view, efforts are needed to address the following points:

1- The reviewers asked for further analyses confirming that the form of cell death is pyroptosis and for more detailed analyses of the pathway:

Rev#1 "Figure 1.."; "Key experiments are shown in Figure 7.."; "additional aspects that need to be investigated.." paragraphs

Rev#2 points #1, #2, #4

2- The reviewers found the order of events unclear between MAPL, mtDNA exit from mitochondria, Gasdermin activation. They asked whether STING is involved. Further analyses are needed to clarify the mechanism:

Rev#1 "Figure 2.." "Does suppression of STING".. "The authors' model proposes.." paragraphs

Rev#2 point #5

Rev#3 points #1, #2, #8; point #7 if possible

3- The reviewers additionally felt that more work is needed to implicate MDVs and clarify the model:

Rev#2 point #3

Rev#3 points #4, #5, #6

4- All other referee concerns pertaining to strengthening existing data, providing controls, methodological details, clarifications and textual changes, should also be addressed.

5- Finally, please pay close attention to our guidelines on statistical and methodological reporting (listed below) as failure to do so may delay the reconsideration of the revised manuscript. In particular, please provide:

We would be happy to consider a revised manuscript that would satisfactorily address these points, unless a similar paper is published elsewhere, or is accepted for publication in Nature Cell Biology in the meantime.

In contrast, although we agree with Referee #3 (point #3) that purifying MDVs may provide valuable insights, we consider this point to be beyond the scope of the present study. Thus, addressing it experimentally will not be necessary for reconsideration of the manuscript at this journal.

- ensure that it conforms to our format instructions and publication policies (see below and www.nature.com/nature/authors/).
- provide a point-by-point rebuttal to the full referee reports verbatim, as provided at the end of this letter.
- provide the completed Editorial Policy Checklist (found here <https://www.nature.com/authors/policies/Policy.pdf>), and Reporting Summary (found here <https://www.nature.com/authors/policies/ReportingSummary.pdf>). This is essential for reconsideration of the manuscript and these documents will be available to editors and referees in the event of peer review. For more information see <http://www.nature.com/authors/policies/availability.html> or contact me.

Nature Cell Biology is committed to improving transparency in authorship. As part of our efforts in this direction, we are now requesting that all authors identified as 'corresponding author' on published papers create and link their Open Researcher and Contributor Identifier (ORCID) with their account on the Manuscript Tracking System (MTS), prior to acceptance. ORCID helps the scientific community achieve unambiguous attribution of all scholarly contributions. You can create and link your ORCID from the home page of the MTS by clicking on 'Modify my Springer Nature account'. For more information please visit <http://www.springernature.com/orcid>.

Link Redacted

We hope that you will find our referees' comments and editorial guidance helpful. Please do not hesitate to contact me if there is anything you would like to discuss. Thank you again for considering NCB for your work.

Best wishes,

Melina

Melina Casadio, PhD
Senior Editor, Nature Cell Biology
ORCID ID: <https://orcid.org/0000-0003-2389-2243>

Reviewers' Comments:

Reviewer #1:

Remarks to the Author:

Nguyen and colleagues investigate how the mitochondrial E3 ligase MAPL induces cell death largely defining the mechanistic basis through MAPL overexpression before investigating (and demonstrating) a role for endogenous MAPL in pyroptosis induced in BMDMs. From their data, the authors propose a model that activated MAPL leads to mtDNA release from mitochondria, via lysosomes that are permeabilized by Gasdermin D. The mtDNA then, by binding cGAS activates NLRP3, leading to inflammasome activation and pyroptosis. The study is undoubtedly interesting and timely and the authors' data support some aspects of the model they propose. That said, I think some key questions remain outstanding, moreover some aspects require more rigorous demonstration. These points are outlined below:

- Figure 1, the authors convincingly show that GSDMD contributes to MAPL induced killing however additional data is required to demonstrate that pyroptosis is occurring (since caspase-8 can also induce GSDMD cleavage/activation under non-pyroptotic conditions), key to address whether NLRP3 and caspase-1 are also required for MAPL induced cell killing, this would give more reassurance that cell death is happening via pyroptosis

- Figure 2. The data linking cGAS activation to inflammasome activity is intriguing but one question is how the authors propose that cGAS is regulating NLRP3 activity? Is cGAS (via STING dependent IFN signaling) potentially involved in priming cells in response to mtDNA sensing (upon MAPL activation) facilitating NLRP3 expression.

Does suppression of STING (RNAi or pharmacological inhibition phenocopy cGAS loss) think important to define whether cGAS is regulating NLRP3 activity in a STING dependent manner.

- The authors model proposes that MAPL lies upstream of mtDNA release leading to GSDMD activation, yet they convincingly show that

GSDMD is required for mtDNA release from lysosomes, this is at odds with the model they present and requires clarification.

- Key expts. are shown in Figure 7, looking at the role of endogenous MAPL in LPS induced pyroptosis in primary BMDMs. LPS directly activates caspase-11, thus potentially a different mechanism (for MAPL involvement) is at play here relative to what is described earlier in the paper (NLRP3 dependent – though this needs tested). Is LPS activating pyroptosis in this model dependent on NLRP3 or direct activation of caspase-11 ?

Additional aspects that need to be investigated (to validate the model proposed) in these expts. are comparison of NLRP3 levels between treated MAPL proficient/deficient BMDMs, caspase-1 activity, GSDMD cleavage and the importance of mtDNA to pyroptosis induced under these circumstances, this will help solidify the model that MAPL impacts on inflammasome activity in an mtDNA dependent manner.

Reviewer #2:

Remarks to the Author:

In this manuscript, Nguyen et al. report a novel role for the outer mitochondrial membrane SUMO ligase MAPL in promoting activation of innate immune sensors cGAS and NLRP3, which execute inflammatory cell death upon MAPL expression. Through extensive CRISPR screens they identify transcriptional upregulation of inflammatory pathways in the presence of MAPL overexpression, which require MAPL SUMO ligase function and are linked to mtDNA release. Cells expressing MAPL have increased formation of mtDNA-containing MDVs that merge with lysosomes, which become leaky and allow mtDNA to access to the cytosol, where it is sensed by cGAS and NLRP3. Overall, this is an interesting and impactful study that adds to our understanding of mechanisms of mtDNA release and activation of cell-intrinsic innate immunity. The findings may help to advance knowledge on how mtDNA release simultaneously engages cGAS and NLRP3 to drive type I interferon, inflammation, and cell death. However, as presented, the paper has several mechanistic gaps and some additional experiments are required to solidify the story for NCB.

Major points:

1. MAPL overexpression drives cell death, which is clear. What is less clear is whether this is truly pyroptosis driven by canonical NLRP3 activation, IFN-dependent necroptosis, or panoptosis involving aspects of both. Prior work has shown that MAPL overexpression activates NF- κ B, and therefore may bypass the priming step usually driven by a PAMP or DAMP. NLRP3 RNA increases after MAPL overexpression consistent with priming, but it is unclear if other inflammasome components are upregulated in MAPL overexpressing cells. The authors should immunoblot for NLRP3, ASC, caspase-1, and pro-IL-1 β in the presence and absence of both RING mutant and FL MAPL to determine if MAPL SUMO ligase activity and signaling drives inflammasome priming. ELISAs measuring IL-1 β or IL-18 release would also add support for a pyroptotic cell death mechanism. Additionally, a pan caspase inhibitor, ZVAD, was used; however, use of a specific caspase-1 inhibitor such as Ac-YVAD-oph or an NLRP3 inhibitor such as MCC950 would more strongly support pyroptotic mechanisms. These experiments are needed in Figures 1, 5, and 7 to definitively show that pyroptotic cell death is occurring in MAPL overexpression cells (less in MAPL Kos) and is altered with the knock down of VPS35 and LRRK2.
2. Figure 2 suggests that knocking down cGAS or depleting DNA alleviates the elevated cell death in MAPL overexpressing cells. In line with the above points, does cGAS knockdown or EtBr lower IL-6 or NLRP3 transcripts? MAPL overexpressing cells also have less mtDNA by PCR, but the IF images don't clearly show this. Can the authors quantitate both mitochondria-localized mtDNA nucleoids and cytosolic nucleoids after MAPL overexpression?
3. The electron micrographs displayed in Figure 3 clearly show MDV formation in MAPL overexpressing cells; however, this figure lacks the rTA control +/- VPS35 siRNA. Also, apart from having MDVs, the mitochondria in the siNT+MAPL overexpression cells have very distorted cristae as compared to the siVPS35 cells. MAPL expression seems to drive many mitochondrial morphological alterations, and it is unclear how knockdown of VPS35 and MIROs, which should only limit MDV formation, rescues these phenotypes. Can the authors clarify what is going on here?
4. The use of MAPL KO primary BMDMs in Figure 7 significantly adds to this study. It would greatly strengthen the previous findings to add MAPL/MAPL ring mut. overexpression to WT BMDMs through a doxycycline inducible retroviral transduction system. THP-1 monocyte/macrophages could also be used if BMDM transduction is technically challenging. In line with this, does LPS/IFN γ upregulate MAPL in BMDMs to initiate mtDNA release, or does it change its activity on mitochondria? Figure 7 should also include classical inflammasome activation protocols (LPS+ATP; LPS+nigericin; LPS+DNA transfection) to determine whether loss of MAPL minimizes inflammasome activation in BMDMs.
5. The terminal steps of the mechanism are unclear in the sense that in order to permeabilize lysosomal membranes, gasdermins need to first be cleaved. However, NLRP3 activation in the model is downstream of the GSDM pore formation that allows mtDNA to leak and be sensed. It is well known that alterations in mitochondrial membrane dynamics allow for direct mtDNA release from the mitochondria. Due to MAPL's role in altering membrane dynamics, is it possible that some mtDNA may leak upstream of MDV release resulting in cGAS sensing and subsequent upregulation of interferons, NF- κ B, and galectins that may prime and promote the downstream leak of mtDNA from the lysosome? Perhaps the DNA leaving directly from mitochondria is modified or localized to uniquely activate cGAS, whereas lysosomal mtDNA entering the cytosol is a better ligand for NLRP3? If this is true, knockdown of VPS35 or NLRP3 should rescue cell death, but the NF- κ B and IFN-I signatures, including upregulation of inflammasome components, should remain intact. This is a critical aspect of the story that is unclear, but if expanded on, would greatly add to our understanding of how MAPL acts as a key factor integrating mtDNA release with both cGAS and NLRP3 activation.

Minor concerns:

1. While the data are largely convincing, there are some concerns with rigor based on the figures and methods provided. These concerns are addressable with more information and quantitation.
 - a. While many of the graphs contain points indicating repetition, (apart from Figure 1) the figure legends lack information detailing the number and type of replicates.
 - b. All microscopy, including Figure 1K & 3D, should be quantified. This quantification should include more than n=2 as in Figure 5.
 - c. Details on how the quantitation of the microscopy was completed should be expanded. This is most unclear in graphs labeled "(% cells with enlarged lysosomes)." Does this mean these cells have 1 large lysosome or are all the lysosomes enlarged? What is the threshold for large lysosomes?
 - d. Some of the graphs lack statistics to match the statements provided (i.e. Figure 1D, Figure 2B/E/F/G/H, Figure 5D, Figure 6C). Additionally, a description of what statistics were performed are missing from Figures 5 & 7.

2. Improvements should be made to the included supplementary figures:

- a. Supplement 1B-E are not referred to in the text.
- b. Supplement B-C are not referred to in the figure legend.
- c. There is a figure labeled supplement 2, but the provided legend is labeled supplement 3. This is referred to as supplement 3 in the text.
- d. The figure labeled supplement 2 with the supplement 3 legend does not provide enough information. Is there a difference between the two panels in this figure?

Reviewer #3:

Remarks to the Author:

In this manuscript titled "The mitochondrial ligase MAPL is an inflammatory rheostat that regulates immune signalling and cell death", Mai Nguyen et al. report that MAPL induces mtDNA trafficking in MDVs to lysosome, followed by the release of mtDNA into the cytosol due to lysosomal permeabilization, the released mtDNA then leads to inflammation that induces pyroptosis. The findings shown in this manuscript are potentially interesting to the readers. However, the data in the manuscript are preliminary and do not support the conclusions well. Consequently, a substantial addition of direct evidences is required to solidify these findings.

Major comments

1. The major events (processes) of mtDNA release upon MAPL expression require direct evidences (images and videos): 1) the process (images and videos) of MDVs containing mtDNA (both MDV and mtDNA should be labeled) formed from mitochondria after MAPL expression; 2) the process (images and videos) of MDVs containing mtDNA contacting and fusing with lysosome, then mtDNA entering into lysosome; 3) the process (images and videos) of mtDNA within lysosome releasing into the cytoplasm.
2. There has been a lot of reports about the types and mechanism of mtDNA release upon some stresses. Does MAPL expression-induced mtDNA release depend on Bax/Bak pore or VDACS pore? Although the author showed that MAPL-induced cell death is not dependent on Bax/Bak pore or VDACS pore, the role of Bax/Bak pore or VDACS pore in MAPL-induced mtDNA release should be tested.
3. MDVs of control and MAPL expressed cells could be purified to test the content of mtDNA within MDVs.
4. In almost all figures related to MDVs, MDVs were not labeled. The specific marker for MDVs should be provided in the images. For example, in Figure 3, MDVs should be labeled (TOM20?) to confirm the released mtDNA locating in MDVs.
5. MDV membrane should be labeled with membrane protein (such as TOM20), and MDV membrane would be not colocalized with lysosome after mtDNA is delivered into lysosome. The related experiments need to be provided.
6. Since the primary function of VPS35 and MIROs is to transport selected cargo proteins, other key factors in MDV formation need to be examined to confirm that MDV is essential for mtDNA release.
7. The colocalization of mtDNA and lysosome does not confirm that mtDNA is delivered to the lysosome, 3D reconstruction of confocal images were required to support the statement.
8. The data about MAPL regulating inflammatory responses are insufficient. At least, the data about mtDNA-activated STING-cGAS pathway should be provided.

Minor comments

1. PicoGreen staining for mtDNA may also stain other DNA fragments. Therefore, the authors can use TFAM-GFP puncta to further indicate mtDNA and confirm mtDNA is released after MAPL expression.
2. In Figure 2F, qRT-PCR assay for cytosolic mtDNA is required.
3. In Figure 3A, MDVs should be labeled.
4. line 328-329, "We observed mtDNA release (Fig. 7A-B) at 6 hours of treatment, and subsequent delivery to lysosomes (Sup Fig. 3)." "supp Fig. 3" should be "Fig.2"; and supp Fig.2 just showed mtDNA colocalizing with lysosome, the statement "subsequent delivery to lysosomes" is not accurate.
5. In Figure 6A, the images related to MAPL- Δ RING should be provided.
6. "G" is missing in Figure 5G.
7. line 304, "Fig.6H" should be "Fig.6G"
8. There are too many references (105) in the manuscript.

READABILITY OF MANUSCRIPTS – Nature Cell Biology is read by cell biologists from diverse backgrounds, many of whom are not

native English speakers. Authors should aim to communicate their findings clearly, explaining technical jargon that might be unfamiliar to non-specialists, and avoiding non-standard abbreviations. Titles and abstracts should concisely communicate the main findings of the study, and the background, rationale, results and conclusions should be clearly explained in the manuscript in a manner accessible to a broad cell biology audience. Nature Cell Biology uses British spelling.

Methods should be written concisely, but should contain all elements necessary to allow interpretation and replication of the results. As a guideline, Methods sections typically do not exceed 3,000 words. The Methods should be divided into subsections listing reagents and techniques. When citing previous methods, accurate references should be provided and any alterations should be noted. Information must be provided about: antibody dilutions, company names, catalogue numbers and clone numbers for monoclonal antibodies; sequences of RNAi and cDNA probes/primers or company names and catalogue numbers if reagents are commercial; cell line names, sources and information on cell line identity and authentication. Animal studies and experiments involving human subjects must be reported in detail, identifying the committees approving the protocols. For studies involving human subjects/samples, a statement must be included confirming that informed consent was obtained. Statistical analyses and information on the reproducibility of experimental results should be provided in a section titled "Statistics and Reproducibility".

All Nature Cell Biology manuscripts submitted on or after March 21 2016 must include a Data availability statement at the end of the Methods section. For Springer Nature policies on data availability see <http://www.nature.com/authors/policies/availability.html>; for more information on this particular policy see <http://www.nature.com/authors/policies/data/data-availability-statements-data-citations.pdf>. The Data availability statement should include:

- Accession codes for primary datasets (generated during the study under consideration and designated as "primary accessions") and secondary datasets (published datasets reanalysed during the study under consideration, designated as "referenced accessions"). For primary accessions data should be made public to coincide with publication of the manuscript. A list of data types for which submission to community-endorsed public repositories is mandated (including sequence, structure, microarray, deep sequencing data) can be found here <http://www.nature.com/authors/policies/availability.html#data>.
- Unique identifiers (accession codes, DOIs or other unique persistent identifier) and hyperlinks for datasets deposited in an approved repository, but for which data deposition is not mandated (see here for details <http://www.nature.com/sdata/data-policies/repositories>).
- At a minimum, please include a statement confirming that all relevant data are available from the authors, and/or are included with the manuscript (e.g. as source data or supplementary information), listing which data are included (e.g. by figure panels and data types) and mentioning any restrictions on availability.
- If a dataset has a Digital Object Identifier (DOI) as its unique identifier, we strongly encourage including this in the Reference list and citing the dataset in the Methods.

We recommend that you upload the step-by-step protocols used in this manuscript to the Protocol Exchange. More details can be found at www.nature.com/protocolexchange/about.

All imaging data should be accompanied by scale bars, which should be defined in the legend.

Cropped images of gels/blots are acceptable, but need to be accompanied by size markers, and to retain visible background signal within the linear range (i.e. should not be saturated). The boundaries of panels with low background have to be demarked with black lines. Splicing of panels should only be considered if unavoidable, and must be clearly marked on the figure, and noted in the legend with a statement on whether the samples were obtained and processed simultaneously. Quantitative comparisons between samples on different gels/blots are discouraged; if this is unavoidable, it should only be performed for samples derived from the same experiment with gels/blots were processed in parallel, which needs to be stated in the legend.

- Some programs can generate Postscript by 'printing to file' (found in the Print dialogue). If using an application not listed above, save the file in Postscript format or email our Art Editor, Allen Beattie for advice (a.beattie@nature.com).

Unprocessed scans of all key data generated through electrophoretic separation techniques need to be presented in a supplementary

figure that should be labelled and numbered as the final supplementary figure, and should be mentioned in every relevant figure legend. This figure does not count towards the total number of figures and is the only figure that can be displayed over multiple pages, but should be provided as a single file, in PDF or TIFF format. Data in this figure can be displayed in a relatively informal style, but size markers and the figures panels corresponding to the presented data must be indicated.

The total number of Supplementary Figures (not including the "unprocessed scans" Supplementary Figure) should not exceed the number of main display items (figures and/or tables (see our Guide to Authors and March 2012 editorial <http://www.nature.com/ncb/authors/submit/index.html#supinfo>; <http://www.nature.com/ncb/journal/v14/n3/index.html#ed>). No restrictions apply to Supplementary Tables or Videos, but we advise authors to be selective in including supplemental data.

GUIDELINES FOR EXPERIMENTAL AND STATISTICAL REPORTING

REPORTING REQUIREMENTS – To improve the quality of methods and statistics reporting in our papers we have recently revised the reporting checklist we introduced in 2013. We are now asking all life sciences authors to complete two items: an Editorial Policy Checklist (found here <https://www.nature.com/authors/policies/Policy.pdf>) that verifies compliance with all required editorial policies and a reporting summary (found here <https://www.nature.com/authors/policies/ReportingSummary.pdf>) that collects information on experimental design and reagents. These documents are available to referees to aid the evaluation of the manuscript. Please note that these forms are dynamic 'smart pdfs' and must therefore be downloaded and completed in Adobe Reader. We will then flatten them for ease of use by the reviewers. If you would like to reference the guidance text as you complete the template, please access these flattened versions at <http://www.nature.com/authors/policies/availability.html>.

Version 1:

Decision Letter:

Dear Heidi,

Thank you again for submitting your revised manuscript "Mitochondrial ligase MAPL is an inflammatory rheostat that regulates immune signalling and cell death", which has now been seen by the original referees, whose comments are pasted below. In light of their advice, we regret that we cannot offer to publish the study in Nature Cell Biology.

As you will see, although the reviewers find that the work improved in revision, and while Reviewer #1 offered positive comments, the other reviewers raised persisting concerns that question the strength of the data and of the novel conclusions that can be drawn at this stage.

As you know, we strive to limit all manuscripts to a single round of major experimental revision to limit the overall time spent in peer review.

The points of greatest concern to us in the remarks below were the need to clarify and strengthen existing data (Rev#3 points #1, 4, 6) and to bolster the mechanistic analyses of cell death and innate immunity (Rev#2 points #1-2-3-4). More experimental work would be needed to address them thoroughly. Given our editorial policy on multiple rounds of major experimental work, we felt we have to return the manuscript to you at this stage, so that you can submit it elsewhere in the interest of time if you wish to fast-track publication.

Given interest in the study, if you decide to invest more time and resources in this study, we would be open to the possibility of considering a revised manuscript that would fully address the referee concerns emphasized above. However, any decision to re-review

such a revised study would depend on the strength of the revisions and the published literature at the time of resubmission.

We are very sorry that we could not be more positive on this occasion, but we thank you for the opportunity to consider this work. We also hope you find the reviews below useful as you determine the next steps for the manuscript.

With kind regards,
Melina

Melina Casadio, PhD
Senior Editor, Nature Cell Biology
Consulting Editor, Nature Structural & Molecular Biology
ORCID ID: <https://orcid.org/0000-0003-2389-2243>

Reviewers' comments:

Reviewer #1 (Remarks to the Author):

The authors have comprehensively addressed all the points I raised during initial review.

Reviewer #2 (Remarks to the Author):

In their revised manuscript, Nguyen et al. greatly improve the quality and quantification of their imaging data. Moreover, the mechanisms defining how VPS34 and MIROs modulate the generation of MAPL-dependent mtDNA containing MDVs has been strengthened and is convincing. However, an exciting and significant portion of the paper focuses on how MAPL dependent mtDNA release activates the innate immune system leading to NLRP3 dependent pyroptosis and cGAS dependent ISG expression. Although the authors added some new experiments to document the role of NLRP3 and cGAS, the data do not go far enough and leave many important aspects unclear. Without additional experiments to clarify and solidify the proposed model, the paper falls short of warranting publication in NCB.

1. We acknowledge the effort made to measure Nlrp3 and Il1b transcripts in Figure 1, but as described in point 1 of our review, this is not sufficient to conclude pyroptosis is happening. At a minimum, evidence including western blotting and ELISA evidence or use of inhibitors in these non-immune cell lines is required. The need for this is highlighted by the data showing GSDME cleavage and GSDMD knockdown in this figure. Robust scientific literature has shown that GSDME is not strongly cleaved by activation of NLRP3. However, the authors show that GSDME is cleaved at much higher levels than GSDMD in their MAPL-OE models. Additionally, GSDMD knock down does not provide a complete rescue, and therefore it is still unclear if NLRP3-dependent pyroptosis is the main form of cell death being initiated in the cell line experiments. Caspase-1-induced apoptosis involves the Bid-caspase-9-caspase-3 axis, which can be followed by GSDME-dependent secondary necrosis or pyroptosis. The authors should clarify the differential activation of GSDMD and GSDME and not lump them together in the manuscript and model.

2. Much of the paper relies on a MAPL overexpression system and is conducted in cells that do not have a canonical and fully functional inflammasome system. Therefore, the experiments in BMDMs are extremely important in solidifying the mechanisms and increasing the biological/immunological relevance of the findings. As I am sure the authors are aware, the NLRP3 inflammasome is activated in a two step process, but here, the authors argue for a direct activation of NLRP3 absent of an obvious signal 2. New data suggest that mtDNA release is not required for NLRP3 activation and caspase-1-dependent GSDMD cleavage. So, what is the mechanism by which NLRP3 gets activated? Inhibitors used in 7A show that NLRP3, Caspase-1, cGAS, and STING are all required for cell death in BMDMs after MAPL overexpression. Does MAPL overexpression oligomerize NLRP3 CARD domains or modify mitochondria-localized NLRP3? This could be assessed in native gels. Also, is ASC oligomerization required for MAPL overexpression-induced NLRP3 activity (assess by native gels)? Finally, can the authors see traditional NLRP3 activation hallmarks in the lysates from exp 7A? That is, Caspase-1 cleavage to the p20 form, GSDMD cleavage, and mature IL-1 β secretion? In addition, can the authors blot lysates to examine NF- κ B activation and IRF/STAT activation in the MAPL overexpressing BMDMs? I applaud the team for developing this system, but the absence of more definitive data here is disappointing.

3. LPS/IFN γ experiments show that MAPL-KO BMDMs have less SYTOX green + cells. Are the WT cells undergoing pyroptosis? Can the authors blot lysates as described above to check? Also, does the NLRP3i, etc. block SYTOX+ cells? The last round of review asked the authors to more closely examine how MAPL-KO BMDMs responded to canonical inflammasome stimuli. For example, LPS priming + nigericin and/or ATP, then assessing NLRP3 levels, caspase-1 cleavage, GSDMD cleavage. These experiments were not provided but are critical controls. Moreover, requested experiments to document pyroptosis in the LRRK2 KO cells were not provided upon revision.

4. Do the downregulated interferon genes in 7F and S8C reach significance in MAPL-KOs vs WT BMDMs? This is unclear, but critically important. Changes at the RNA level are not convincing if they do not translate to measurable decreases at the protein level, so blots would be informative.

Reviewer #3 (Remarks to the Author):

The quality of the revised manuscript has improved, and some of my concerns have been addressed and resolved. However, there are still several questions and issues that remain unresolved in the revised version.

Major comments

1. MAPL overexpression led to the formation of electron-dense double-membrane protrusions from mitochondria (Figure 3F). However, Figures 3C and 3E indicate that MDVs are mtDNA+/CI+/TOM40-, suggesting that MDVs should consist of a single membrane derived from the inner mitochondrial membrane. This presents an inconsistency in the data.

2. The data in Figure 3 primarily demonstrate mtDNA release through MDVs, with information on MIROs limited to Figure 3B. Therefore, the title of the paragraph, 'MIROs facilitate mtDNA removal in mitochondrial-derived vesicles,' should be revised. Additionally, since the

role of MIROs in MDV formation has already been reported (Cargo-selected MDVs are formed by the lateral tubulation of mitochondrial membranes along microtubules via the Rho GTPases MIRO1 and MIRO2 before DRP1-mediated scission 40), and the data related to MIROs in this manuscript are not closely aligned with the main focus and could therefore be omitted.

3. line 160-163, "silencing Gasdermin C, D and E together had no effect on the appearance of cytosolic mtDNA foci in MAPL-expressing cells (Fig. 2J). These data indicate that mtDNA is unlikely to be released to the cytosol through mitochondrial membrane pores". How are MDVs released into the cytosol upon MAPL expression? Is it through Bax/Bak or VDAC1 pores? No supporting data have been provided to address this.

4. Supplementary video 1 is not convincing. Firstly, the quality is poor with low resolution. Additionally, MitoTracker is not ideal for this experiment, as it depends on mitochondrial membrane potential and cannot demonstrate mtDNA release in a MDV-dependent manner. An outer membrane marker (such as TOM20) and an inner membrane marker (such as a complex I subunit) should be used instead.

5. The authors demonstrate that "mtDNA is trafficked to endolysosomal subcompartments" and "These data identify lysosomes as the target organelle for mtDNA-containing MDVs". Then, what happens to mtDNA after entering the endolysosomal subcompartments? Is the mtDNA degraded? The data in Figure 6 show that mtDNA can escape into the cytosol. However, how much mtDNA actually escapes?

6. The authors report that mtDNA is released into the cytosol from damaged lysosomes to activate cGAS/STING signaling, a process dependent on gasdermin pores in the lysosome. However, does gasdermin knockdown inhibit MAPL-induced activation of cGAS/STING signaling, considering that gasdermin knockdown does not block mtDNA release from mitochondria?

**For Nature Portfolio general information and news for authors, see <http://npg.nature.com/authors>.

Version 2:

Decision Letter:

Our ref: NCB-A53659B-Z

14th July 2025

Dear Dr. McBride,

Dear Heidi,

I am very excited the reviewers have gotten back to us so quickly and very happy to share with you their comments.

Thank you for submitting your revised manuscript "MAPL orchestrates gasdermin-mediated release of mtDNA from lysosomes that drives pyroptotic cell death" (NCB-A53659B-Z). It has now been seen by the original referees and their comments are below. The reviewers find that the paper has improved in revision, and therefore we'll be happy in principle to publish it in Nature Cell Biology, pending minor revisions to satisfy the referees' final requests and to comply with our editorial and formatting guidelines.

Thank you again for your interest in Nature Cell Biology. Please do not hesitate to contact me if you have any questions.

Sincerely,

Angela R Parrish, PhD
Locum Senior Editor
Nature Cell Biology

Reviewer #2 (Remarks to the Author):

The authors have added significant new data to the manuscript, clarifying the cell death mechanisms significantly. New results on GSDME being upstream of GSDMD are highly informative. Although there are some remaining questions, answering them will require additional work that is beyond the scope of this paper. I believe the paper provides several important advances and am supportive of publication. I thank the authors for their efforts to strengthen the story.

Reviewer #3 (Remarks to the Author):

The quality of the revised manuscript has improved, and some of my concerns have been addressed and resolved.

Version 3:

Decision Letter:

Dear Dr McBride,

I am pleased to inform you that your manuscript, "MAPL regulates gasdermin-mediated release of mtDNA from lysosomes to drive pyroptotic cell death", has now been accepted for publication in *Nature Cell Biology*.

Over the next few weeks, your paper will be copyedited to ensure that it conforms to *Nature Cell Biology* style. Once your paper is typeset, you will receive an email with a link to choose the appropriate publishing options for your paper and our Author Services team will be in touch regarding any additional information that may be required.

Publication is conditional on the manuscript not being published elsewhere and on there being no announcement of this work to any media outlet until the online publication date in *Nature Cell Biology*.

Please note that *Nature Cell Biology* is a Transformative Journal (TJ). Authors may publish their research with us through the traditional subscription access route or make their paper immediately open access through payment of an article-processing charge (APC). Authors will not be required to make a final decision about access to their article until it has been accepted. [Find out more about Transformative Journals](https://www.springernature.com/gp/open-research/transformative-journals)

Authors may need to take specific actions to achieve compliance with funder and institutional open access mandates. If your research is supported by a funder that requires immediate open access (e.g. according to [Plan S principles](https://www.springernature.com/gp/open-science/plan-s-compliance) or the [NIH public access policy](https://www.springernature.com/gp/open-science/us-federal-agency-compliance)) then you should select the gold OA route, and we will direct you to the compliant route where possible. Because authors warrant under our subscription licensing terms that they haven't committed to licensing any version of their article under a licence inconsistent with the terms of our agreement – including the applicable embargo period – publication under the subscription model isn't suitable for authors whose funders require no embargo.

If you have not already done so, we strongly recommend that you upload the step-by-step protocols used in this manuscript to protocols.io (<https://protocols.io>), an open online resource that allows researchers to share their detailed experimental know-how. All uploaded protocols are made freely available and are assigned DOIs for ease of citation. Protocols and Nature Portfolio journal papers in which they are used can be linked to one another, and this link is clearly and prominently visible in the online versions of both. Authors who performed the specific experiments can act as primary authors for the Protocol as they will be best placed to share the methodology details, but the Corresponding Author of the present research paper should be included as one of the authors. By uploading your Protocols onto protocols.io, you are enabling researchers to more readily reproduce or adapt the methodology you use, as well as

increasing the visibility of your protocols and papers. You can also establish a dedicated workspace to collect your Lab Protocols. Further information can be found at <https://www.protocols.io/help/publish-articles>.

Nature Cell Biology encourages authors presenting evidence for cell, biological, molecular, and genetic interactions to consider communicating these findings using Biofactoid (<https://biofactoid.org/>). This tool helps users share a searchable representation of interactions (e.g. binding, gene expression, post-translational modification) between genes, gene products, or chemicals. Information added to Biofactoid, with author attribution, is shared on social media and public databases, such as Pathway Commons, where it can be discovered and analyzed in the context of a large and growing corpus of knowledge.

With kind regards,

Angela R Parrish, PhD
Locum Senior Editor
Nature Cell Biology

** Visit the Springer Nature Editorial and Publishing website at http://editorial-jobs.springernature.com?utm_source=ejp_NCB_email&utm_medium=ejp_NCB_email&utm_campaign=ejp_NCB for more information about our career opportunities. If you have any questions please click [here](mailto:editorial.publishing.jobs@springernature.com).

NCB-M44823, Nguyen, Collier et al.

AUTHORS RESPONSE TO EDITOR AND REVIEWERS COMMENTS/CONCERNS:

We would like to thank the reviewers for taking their time to give us in-depth comments and suggestions. They have significantly improved the scope and quality of this manuscript. Please find below our responses to each suggestion.

The editors highlighted major changes needed for consideration which we summarize first here:

1- The reviewers asked for further analyses confirming that the form of cell death is pyroptosis and for more detailed analyses of the pathway:

Rev#1 “Figure 1..”; “Key experiments are shown in Figure 7..”; “additional aspects that need to be investigated..” paragraphs

Rev#2 points #1, #2, #4

We performed key new experiments to address these points. We utilized the chemical inhibitors against NLRP3, cGAS, STING and caspase 1 within the BMDM model system ectopically expressing MAPL showing clear inhibition of cell death quantified by Sytox Green uptake (**Fig 7A**). We added a series of western blots and qRT-PCR experiments to follow the impact of ectopic MAPL expression on cytokine and other immune responses in different cultured and primary cell lines (human fibroblast, BMDM and hepatocytes, **Figs 1H,I and Fig 7G**). We also completed a full RNA-Seq dataset from MAPL KO or WT BMDM treated with LPS/IFN γ to examine the requirement for MAPL on the signaling and transcriptional responses (**Fig. 7F and Supplemental table 2, Supp Fig 8**). These data show some requirement for MAPL in the induction of Type 1 interferon related genes (which may be cGAS/STING target genes), but the signaling pathways leading to expression of pyroptotic genes are not dependent on MAPL. This is further shown by efficient expression of Nlrp3, Pro-Caspase 1 and Pro-IL1 β protein measured by immunoblotting/ELISA in Figure 7. Overall, our data highlight the most critical role for endogenous MAPL acting upstream of cGAS/STING as a driver of pyroptotic cell death directly from mitochondria. Responses to each point are outlined below, corresponding to specific points raised.

2- The reviewers found the order of events unclear between MAPL, mtDNA exit from mitochondria, Gasdermin activation. They asked whether STING is involved. Further analyses are needed to clarify the mechanism:

Rev#1 “Figure 2..” “Does suppression of STING”.. “The authors’ model proposes..” paragraphs

Rev#2 point #5

Rev#3 points #1, #2, #8; point #7 if possible

We used inhibitors of cGAS and STING to confirm that mtDNA release is key to MAPL-induced cell death in BMDMs (**Fig 7A**). To directly test the role of mtDNA in the pathway we used Rho0 cells and showed that the MAPL-induced cleavage of gasdermins still occurred (**Fig 2E**). Coupled with the dependence of the pathway on caspase 1 (again using inhibitors suggested by reviewers), it shows that the initial activation of NLRP3 does NOT depend on mtDNA-mediated activation cGAS or STING. On the other hand, the requirement for MAPL in LPS/IFN γ induced death within BMDM

was restored upon the addition of 2'3'cGAMP (**Fig 7H**). This further highlights the critical role of MAPL-induced mtDNA release and activation of cGAS in the process of pyroptotic cell death. Responses to each point are outlined in detail below.

3- The reviewers additionally felt that more work is needed to implicate MDVs and clarify the model:

Rev#2 point #3

Rev#3 points #4, #5, #6

We now identified a protein cargo carried with mtDNA within MAPL-induced MDVs (**Fig 3C-E**) and provided additional video, EM and confocal evidence supporting the role of MDVs in mtDNA transport to the lysosomes (**Figs 3-6**). We additionally identified an even using time-lapse microscopy showing the exit of DNA from GAL3-positive foci (**Fig 6G**), indicating release of mtDNA from damaged lysosomes. Responses to each point are outlined in detail below.

ADDITIONAL NEW DATA IN REVISED MANUSCRIPT:

Fig. 1: New **Fig. 1B,C** expand the cell types we examined, revealing that MAPL induces cell death in U2OS, human fibroblasts, HUH-7 and baby mouse kidney (Fig. 1C) cells. New **Fig. 1I** shows similar induction of IL6, NLRP3 and IL1B in human fibroblast cells expressing MAPL (but not Δ RING), further suggesting that the action of MAPL is applicable to a range of cell types.

Fig. 2: New qRT-PCR of type 1 IFN gene expression (which can be activated downstream of cGAS/STING signalling) [IFNA4 and IFN β 1] are shown in **Fig. 2C**. We expanded our analysis of phenotypes in Rho0 cells with imaging and cell death protection in **Fig. 2F,G**. Biochemical analysis reveals that while Rho0 cells were fully protected against MAPL-induced cell death, they still show efficient cleavage of Gasdermins D and E (**Fig. 2E**). Cytosol fractionations and qPCR of mtDNA genes upon ectopic MAPL expression are now moved to **Fig. 6** where we describe the GAL3 recruitment and pore formation to release mtDNA.

Fig. 3: New analysis of mtDNA⁺ foci also contain a component of complex I (inner membrane) but not outer membrane marker TOM40 (**Fig. 3C** with quantification in D). We also identified vesicular tubulation events by confocal microscopy (**Fig. 3E**) containing mtDNA, complex I, but not TOM40. We expanded the EM panels documenting MDV profiles within MAPL expressing cells, including addition of control images from rtTA-expressing cells. We removed the EM images of siVPS35 based on the very astute comments of the reviewer (further explained in detail below). The quantification of mtDNA release in siVPS35 conditions is now moved into **Fig. 5E** where we focus on the role of the lysosomal machineries in MAPL induced death.

Fig. 4: Imaging data in **Fig. 4A** showing delivery of MDVs to lysosomes now include the Δ RING controls. We improved the clarity of the figure showing video frames of mtDNA⁺ MDV delivery to lysosomes in **Fig. 4C** by including the split channels at a few frames and enlarged the images. Here, we aimed to highlight the clear overlap of the PicoGreen signal and lysotracker in a subset of lysosomes. **Fig. 4D**, showing an additional event of selective mtDNA trafficking to lysosomes in a more direct manner, was also made clearer. We moved the data showing that Parkin was not recruited to mitochondria upon MAPL expression into the supplement.

Fig. 5: We now show individual data points for **Fig. 5D** including three independent repeats. We have added analysis of DNA release in cells after MAPL expression and siVPS35 or siLRRK2 in **Fig. 5E and F**.

Fig. 6: This figure still focuses on the breach at lysosomes indicated by GAL3-RFP recruitment. We have added Δ RING images to **Fig. 6A** and increased the clarity of images shown in **Fig. 6D** to clearly highlight the mtDNA within these lysosomes. In addition, we have included live video evidence of PicoGreen-positive stained “puffs” emerging from Gal3-positive lysosomes (**new Fig. 6G, and Supplementary Video 5**). We also performed fractionations of cytosol for qRT-PCR and include the data in **Fig 6H**. Lyso-IP showing the incorporation of the N-terminus of Gasdermin E was repeated with better controls included as we now optimized the purification to obtain much lower mitochondrial contamination (**new Fig. 6L**).

Fig. 7: This figure is mostly new and focuses only on the MAPL-induced killing of primary BMDM, and LPS/IFN γ treatments of MAPL^{-/-} BMDM. **Fig. 7A** now demonstrates the strong inhibition of MAPL-induced pyroptotic cell death (SYTOX Green uptake) by chemical inhibitors of NLRP3, cGAS, STING and caspase 1. New **Fig. 7H** shows that the block in LPS/IFN γ induced pyroptotic cell death is rescued upon addition of 2'3'cGAMP to MAPL KO BMDM. New **Fig. 7G** shows Western blots revealing no changes in the induction of NLRP3, Pro-caspase 1, MAPL, and ELISA of Pro-IL1 β release in MAPL^{-/-} BMDM, confirming MAPL is not required for main immune signalling in primary BMDMs in response to inflammation. **Fig. 7F** shows main changes in gene expression from RNAseq analysis of primary BMDM wild type or MAPL KO after 4.5h treatment with LPS/IFN γ .

Fig. 8: Since findings in Fig. 7 are consistent with our MAPL overexpression model, **Fig. 8** returns to examining the role of VPS35 and LRRK2 in LPS/IFN γ -induced cell death (data previously in Fig 7). We have also included images to highlight that enlarged lysosomes are observed in LRRK2 KO BMDMs after LPS/IFN γ treatment, and that these lysosomes contain DNA (**Fig. 8I**). Our new model (**Fig 8J**) now indicates the two separate arms of MAPL activity – one to activate NLRP3, and the other to induce mtDNA⁺ MDVs, transport to lysosomes and release into cytosol through gasdermin pores.

Point by point response to review

Reviewer 1

Reviewer #1 (R1): Nguyen and colleagues investigate how the mitochondrial E3 ligase MAPL induces cell death largely defining the mechanistic basis through MAPL overexpression before investigating (and demonstrating) a role for endogenous MAPL in pyroptosis induced in BMDMs. From their data, the authors propose a model that activated MAPL leads to mtDNA release from mitochondria, via lysosomes that are permeabilized by Gasdermin D. The mtDNA then, by binding cGAS activates NLRP3, leading to inflammasome activation and pyroptosis. The study is undoubtedly interesting and timely and the authors' data support some aspects of the model they propose. That said, I think some key questions remain outstanding, moreover some aspects require more rigorous demonstration. These points are outlined below:

1. Figure 1, the authors convincingly show that GSDMD contributes to MAPL induced killing however additional data is required to demonstrate that pyroptosis is occurring (since caspase-8 can also induce GSDMD cleavage/activation under non-pyroptotic conditions), key to address whether NLRP3 and caspase-1 are also required for MAPL induced cell killing, this would give more reassurance that cell death is happening via pyroptosis

HM: We thank the reviewer for encouraging us to validate these pathways further. The CRISPR screen identified NLRP3, ASC and Caspase 1 as requisite for MAPL-induced cell death, which initially prompted us to look at gasdermins and pyroptosis as the core death mechanism. We had provided evidence that the gene expression of NLRP3 and IL6 was induced upon MAPL expression (**Fig. 1**). We expanded these findings in human fibroblasts, which demonstrated increased expression of NLRP3, IL6 and IL1B (**New Fig 1I**). In our revised version we employed established NLRP3 (MCC950) and caspase 1 specific (YVAD) inhibitors and now show a dose-dependent block in MAPL-induced cell death, quantified by Sytox green uptake (**New Fig. 7A**, done in BMDM cells to also address R1's 5th point below). These data confirm both the induction and requirement for NLRP3 in MAPL-induced cell death, further consistent with this as a pyroptotic death pathway.

R1: 2. Figure 2. The data linking cGAS activation to inflammasome activity is intriguing but one question is how the authors propose that cGAS is regulating NLRP3 activity? Is cGAS (via STING-dependent IFN signaling) potentially involved in priming cells in response to mtDNA sensing (upon MAPL activation) facilitating NLRP3 expression.

HM: We appreciate that our work had not unmasked the precise relationship between cGAS/STING and NLRP3. We now provide new evidence that MAPL expression is activating NLRP3 fully independent of mtDNA release. While MAPL-induced cell death was entirely dependent on mtDNA (using Rho0 cells **Fig. 2D**), we still observe gasdermin D and E cleavage in these cells (**New Fig. 2E**) and the gasdermin-dependent recruitment of GAL3 to the lysosomes remains unchanged in Rho0 cells (**Fig. 6J**). This indicates that mtDNA and cGAS acted *downstream* of NLRP3, Caspase 1

and Gasdermin cleavage. As previously reported, once the mtDNA is released into the cytosol it may then bind and activate NLRP3 further to amplify the pathway (Xian et al., 2022), which can explain the increase in gasdermin cleavage in cells containing mtDNA (**Fig 2D**). Nevertheless, inhibiting cGAS or STING prevented MAPL-induced cell death in BMDMs (**New Fig. 7A**), suggesting cGAS plays an essential role in this pathway. Therefore, MAPL expression activates NLRP3 either directly or indirectly. The screen identified a series of kinases and adaptors linked to the immune signalling pathways including STK26/MST4 (Jiao et al., 2015), the linker for activation of T cells (LAT), GPR146 (Nakayama et al., 2020), TRIM44 (Yang et al., 2013) and others, any of which may be targets of MAPL. While we are working hard to define the specific SUMO or ubiquitin targets of MAPL that lead to NLRP3 activation, resolving this will take us significantly more time. We sincerely hope the reviewer might appreciate this point. We have included a paragraph to the discussion outlining the limitations of our study and essential next steps and highlight that the relationship between cGAS/STING and NLRP3 is only just emerging. It will take more time to fully delineate these mechanisms.

RI: 3. Does suppression of STING (RNAi or pharmacological inhibition phenocopy cGAS loss) think important to define whether cGAS is regulating NLRP3 activity in a STING dependent manner.

HM: As in our response to point 2 above, the fact that Rho0 cells lacking mtDNA retain significant caspase activity and gasdermin cleavage suggests cGAS and STING act downstream of NLRP3. We used chemical inhibitors against cGAS (G140) and STING (H151) applied to BMDM cells ectopically expressing MAPL. Both inhibitors blocked cell death visualized by Sytox Green uptake (**New Fig 7A**). Therefore we propose that the release of mtDNA from lysosomes and activation of cGAS then amplifies NLRP3 signalling, and leads to pyroptotic cell death.

In a second approach to test the contribution of cGAS/STING in cell death we now show that the inhibition seen in MAPL KO BMDM treated with LPS/IFN γ can be reversed upon the direct addition of cGAMP, the product of cGAS that binds to STING (**Fig 7H**). This experiment demonstrates that the most critical function of MAPL in this death pathway is the activation of the cell biology pathways resulting in mtDNA release through the lysosome.

RI: 4. The authors model proposes that MAPL lies upstream of mtDNA release leading to GSDMD activation, yet they convincingly show that GSDMD is required for mtDNA release from lysosomes, this is at odds with the model they present and requires clarification.

HM: We apologize for our lack of clarity in the text and we have worked to improve this in the current version with more data and (hopefully) better writing. Our data show very clearly that mtDNA is *not required* for gasdermin cleavage since Rho0 cells expressing MAPL still see gasdermin cleavage (**New Fig 2E**) but our data is also consistent with the idea that mtDNA can amplify the inflammasome response (**Fig. 2E**). We also found that GAL3 recruitment to lysosomes was not blocked in cells lacking mtDNA (**Fig. 6J**), yet it was blocked in cells lacking GSDMs (**Fig. 6K**). Therefore, our subsequent model proposes that MAPL expression leads to two distinguishable events: (1) the generation of mtDNA-containing MDVs that traffic to lysosomes, and in an independent arm, (2) the activation of NLRP3, caspase-1 and gasdermin cleavage. The two things together lead to mtDNA release from lysosomes and pyroptotic cell death. When gasdermins are lost,

the MDV mediated mtDNA delivery to lysosomes is unaffected but no pores can form to release mtDNA from the lysosomal compartments. When mtDNA is lost (Rho0), we still see gasdermin cleavage, but there's no mtDNA to be released from these lysosomes and cell death is blocked. We tried to draw these two arms in the model cartoon (**New Fig 8J**) we presented but hope these points are made more clearly in our revised version.

R1: 5. Key expts. are shown in Figure 7, looking at the role of endogenous MAPL in LPS induced pyroptosis in primary BMDMs. LPS directly activates caspase-11, thus potentially a different mechanism (for MAPL involvement) is at play here relative to what is described earlier in the paper (NLRP3 dependent – though this needs tested). Is LPS activating pyroptosis in this model dependent on NLRP3 or direct activation of caspase-11?

HM: In the context of LPS/IFN γ stimulation, we see robust induction of NLRP3 protein by 4.5 hours, and we see release of IL1 β into the media at 24 hours (**New Fig. 7G**). While this confirms that LPS is activating the NLRP3 inflammasome which can cleave gasdermins, we cannot exclude a contribution of caspase 11. A function of MAPL in regulating caspase 11 would need extensive further work to model. Given the data we have obtained thus far, we propose that MAPL-dependent release of mtDNA in MDVs, delivery to the lysosome and generation of gasdermin pores at the lysosomes – whether activated by caspase (1 or 11) - is what drives cell death.

Reviewer 2

Reviewer #2 (R2): In this manuscript, Nguyen et al. report a novel role for the outer mitochondrial membrane SUMO ligase MAPL in promoting activation of innate immune sensors cGAS and NLRP3, which execute inflammatory cell death upon MAPL expression. Through extensive CRISPR screens they identify transcriptional upregulation of inflammatory pathways in the presence of MAPL overexpression, which require MAPL SUMO ligase function and are linked to mtDNA release. Cells expressing MAPL have increased formation of mtDNA-containing MDVs that merge with lysosomes, which become leaky and allow mtDNA to access to the cytosol, where it is sensed by cGAS and NLRP3. Overall, this is an interesting and impactful study that adds to our understanding of mechanisms of mtDNA release and activation of cell-intrinsic innate immunity. The findings may help to advance knowledge on how mtDNA release simultaneously engages cGAS and NLRP3 to drive type I interferon, inflammation, and cell death. However, as presented, the paper has several mechanistic gaps and some additional experiments are required to solidify the story for NCB. Major points:

1. MAPL overexpression drives cell death, which is clear. What is less clear is whether this is truly pyroptosis driven by canonical NLRP3 activation, IFN-dependent necroptosis, or panoptosis involving aspects of both. Prior work has shown that MAPL overexpression activates NF- κ B, and therefore may bypass the priming step usually driven by a PAMP or DAMP. NLRP3 RNA increases after MAPL overexpression consistent with priming, but it is unclear if other inflammasome components are upregulated in MAPL overexpressing cells. The authors should immunoblot for NLRP3, ASC, caspase-1, and pro-IL-1 β in the presence and absence of both RING mutant and FL MAPL to determine if MAPL SUMO ligase activity and signaling drives inflammasome priming. ELISAs measuring IL-1 β or IL-18 release would also add support for a pyroptotic cell death mechanism. Additionally, a pan caspase inhibitor, ZVAD, was used; however, use of a specific caspase-1 inhibitor such as Ac-YVAD-oph or an NLRP3 inhibitor such as MCC950 would more strongly support pyroptotic mechanisms. These experiments are needed in Figures 1, 5, and 7 to definitively show that pyroptotic cell death is occurring in MAPL overexpression cells (less in MAPL Kos) and is altered with the knock down of VPS35 and LRRK2.

HM: We appreciate that Sytox green positivity reflects multiple types of pore-mediated cell death pathways. We have now added new data demonstrating that the specific inhibitors suggested by the reviewer block Sytox green uptake into BMDM cells ectopically expressing MAPL in a dose-dependent manner (**New Fig. 7A**). This was done for both NLRP3 (MCC950) and caspase 1 (YVAD), as requested by the reviewer, along with inhibitors of STING and cGAS. We repeated mRNA analysis in a human fibroblast line and also see induction of NLRP3 and IL1 β in a RING-dependent manner (**New Fig. 1I**). Therefore, in multiple cell types we observe MAPL induction of distinct players specific to the pyroptotic pathway. We had shown the dependence of LPS/IFN γ induced pyroptotic cell death on MAPL in primary BMDM, but now added data that MAPL was not required for the induced expression of NLRP3 protein or the release of IL1 β (by ELISA) (**New Fig 7G,G'**). Full RNAseq details all immune related genes, where only a subset were reduced in the absence of MAPL (**New Fig 7F and Supp Fig 8**). With this we have expanded our analysis of the

immune response upon MAPL expression, and in LPS/IFN γ conditions lacking MAPL. The loss of LRRK2 or VPS35 block LPS/IFN γ induced Sytox green uptake (cell death) in BMDM cells (**new Fig 8D,I**), so the pathway is indeed altered.

R2: 2. Figure 2 suggests that knocking down cGAS or depleting DNA alleviates the elevated cell death in MAPL overexpressing cells. In line with the above points, does cGAS knockdown or EtBr lower IL-6 or NLRP3 transcripts? MAPL overexpressing cells also have less mtDNA by PCR, but the IF images don't clearly show this. Can the authors quantitate both mitochondria-localized mtDNA nucleoids and cytosolic nucleoids after MAPL overexpression?

HM: Data in the Rho0 cells indicate that NLRP3 driven caspase activation and cleavage of gasdermins remain in the *absence* of mtDNA (**New Fig 2E**). Therefore, we place mtDNA downstream of the initial signaling through NLRP3/caspase/gasdermins. Previous work in the Shadel (West et al., 2015) and Prudent labs (Zecchini et al., 2023) have shown that loss of cGAS led to a partial ~40% reduction of cytokine or interferon responsive genes with different stimuli. So mtDNA release is not essential for the initial transcriptional side of the pathway, particularly once canonical signal transduction cascades are in play. Our new RNAseq analysis of primary BMDM treated with LPS/IFN γ at 4.5 and 24 hours in wild type or MAPL KO cells shows that while MAPL is essential for pyroptotic cell death, the major transcriptional responses are unaltered, with the exception of a reduction in Type I IFN related genes (**New Fig 7F**). We further show now that cell death can resume in MAPL KO BMDM upon addition of the cGAS product 2'3'cGAMP, arguing that the activation of these pathways downstream of mtDNA release is a key driver of death, outside of the overall transcriptional response to LPS/IFN γ . Our data indicates that cytosolic DNA sensors amplify (feeds forward) these signals, and are critical for pyroptotic cell death.

In the second part of this question the reviewer was concerned that our initial quantitative analysis of total cellular mtDNA levels showing significant reduction did not really match the Picogreen images we present. We therefore repeated the total qPCR in whole cells on many more biological replicates, and the data now show the average reduction at 24 hours was minimal (which is when most of the images are taken). We have now removed the qPCR analysis in whole cells and replaced it with our more robust datasets from fractionated cytosols showing that MAPL expression leads to the release of mtDNA (**new Fig. 6H**). We have also included new videos taken of more enlarged lysosomes carrying mtDNA after MAPL expression that show "puffs" of picogreen signal emerging from the galectin 3⁺ lysosome (**Fig. 6G**). We consider this further evidence of DNA being directly released from breached lysosomes using live cell imaging.

R2: 3. The electron micrographs displayed in Figure 3 clearly show MDV formation in MAPL overexpressing cells; however, this figure lacks the rtTA control +/- VPS35 siRNA. Also, apart from having MDVs, the mitochondria in the siNT+MAPL overexpression cells have very distorted cristae as compared to the siVPS35 cells. MAPL expression seems to drive many mitochondrial morphological alterations, and it is unclear how knockdown of VPS35 and MIROs, which should only limit MDV formation, rescues these phenotypes. Can the authors clarify what is going on here?

HM: We have included rtTA controls and generated a new **Fig 3D**. We know that MAPL expression drives mitochondrial fragmentation, as we have indeed reported on that before when we studied

DRP1 SUMOylation (Braschi et al., 2009) (Prudent et al., 2015). The reviewer, however, raises an important point that loss of VPS35 had additional impacts on the MAPL fragmented mitochondria beyond the blocking of mtDNA containing MDVs. We do not know the mechanism and have been looking into this in more detail. In a preliminary dataset (shown to the reviewer below) you see the quantification of mitochondrial morphology using confocal fluorescent imaging. Using a number of mitochondrial morphology parameters we see that MAPL expression generates smaller, fragmented mitochondria, which is indeed rescued upon VPS35 silencing (as the reviewer noticed in the EM). As we hope the reviewer might appreciate, we have not enough time to detail the mechanisms underlying this in the context of a revision, and we submit that this would distract from the core message of our paper. We are very interested in these observations indeed. For this resubmission, we have therefore removed the siVPS35 electron micrographs from the figure, but added a representative rtTA control image (**New Fig 3F**). Should the reviewer object, we are happy to put them back and could make a point that this silencing also improves mitochondrial morphology in the text (with quantification like seen below in supplement). We worry this might be a bit distracting.

R2: 4. The use of MAPL KO primary BMDMs in Figure 7 significantly adds to this study. It would greatly strengthen the previous findings to add MAPL/MAPL ring mut. overexpression to WT BMDMs through a doxycycline inducible retroviral transduction system. THP-1 monocyte/macrophages could also be used if BMDM transduction is technically challenging. In line with this, does LPS/IFN γ upregulate MAPL in BMDMs to initiate mtDNA release, or does it change its activity on mitochondria? Figure 7 should also include classical inflammasome activation protocols (LPS+ATP; LPS+nigericin; LPS+DNA transfection) to determine whether loss of MAPL minimizes inflammasome activation in BMDMs.

HM: We thank the reviewer for their appreciation of our use of primary BMDM from the KO mice. We have significantly expanded on the experiments shown in the previous version of Figure 7. We utilized our adenoviral system to express MAPL in primary BMDM and again showed the induction of pyroptotic cell death (**New Fig 7A**). We used this system to show that small molecule inhibition of NLRP3, cGAS, STING and caspase 1 all blocked MAPL-induced cell death (**New FIG 7A**). Therefore we didn't have a need to develop inducible or other expression systems.

We appreciate the request to expand our analysis of MAPL requirement in the induction of NLRP3. Our western blot analysis of NLRP3 and Pro-IL1 β expression and secretion (by ELISA) in LPS/IFN γ treated MAPL KO BMDM showed no change (**New Fig 7G,G'**), while cell death was blocked (**FIG 7B,C**). In response to the reviewers comments we performed complete RNA-Seq

datasets in BMDM +/- LPS/IFN γ at 4.5 and 24 hours after treatment (**Supplementary Table 2 and Supp Fig 8A,B**). We observed mild induction of MAPL mRNA at 4.5 hours after LPS/IFN γ treatment, as the reviewer predicted, which is in the **New Supplement Fig.8B**. However, this wasn't seen at the protein level (**New Fig 8G**). It's more likely that MAPL RING activity is activated upon LPS/IFN γ treatment, and we are working to identify potential kinases or modifications that may explain this. In general, however, MAPL knockout cells reveal that MAPL is not essential for robust inflammasome activation and signal transduction downstream of the TLR4 pathways. However, upon careful analysis of RNAseq data we detected a reduction in the mRNAs encoding a series of Type 1 interferon related genes (**New Fig. 7F**). Given that our data indicated MAPL is not essential in inflammasome activation, we didn't go much further in this revision exploring how MAPL may contribute to the activate the inflammasome in other pathways (LPS/nigericin/ATP/DNA). Our data instead demonstrate the strict requirement for MAPL in pyroptotic cell death that occurs 24-48 hours after treatment. We will continue to seek the mechanistic links between MAPL and NLRP3 activation *versus* MDV formation in future work.

R2: 5. The terminal steps of the mechanism are unclear in the sense that in order to permeabilize lysosomal membranes, gasdermins need to first be cleaved. However, NLRP3 activation in the model is downstream of the GSDM pore formation that allows mtDNA to leak and be sensed. It is well known that alterations in mitochondrial membrane dynamics allow for direct mtDNA release from the mitochondria. Due to MAPL's role in altering membrane dynamics, is it possible that some mtDNA may leak upstream of MDV release resulting in cGAS sensing and subsequent upregulation of interferons, NF-kB, and galectins that may prime and promote the downstream leak of mtDNA from the lysosome? Perhaps the DNA leaving directly from mitochondria is modified or localized to uniquely activate cGAS, whereas lysosomal mtDNA entering the cytosol is a better ligand for NLRP3? If this is true, knockdown of VPS35 or NLRP3 should rescue cell death, but the NF-kB and IFN-I signatures, including upregulation of inflammasome components, should remain intact. This is a critical aspect of the story that is unclear, but if expanded on, would greatly add to our understanding of how MAPL acts as a key factor integrating mtDNA release with both cGAS and NLRP3 activation.

HM: We appreciate these points questioning how NLRP3 activation would require lysosomal breach and mtDNA release. In this revised manuscript we returned to the Rho0 cells where we still observed the recruitment of Gal3 to lysosomes upon MAPL expression (**Fig. 6J**). Using western blot approaches we now show that gasdermin D and E cleavage is still observed in MAPL expressing Rho0 cells, showing more directly that MAPL expression activates NLRP3 without the need for cGAS activation (**New Fig. 2E**). The levels of gasdermin cleavage are further enhanced in cells containing mtDNA, indicating that cytosolic DNA can amplify NLRP3 activity. Therefore, we conclude that MAPL activation of NLRP3 is *upstream* of the caspase-1 mediated gasdermin cleavage and fully independent of mtDNA. The generation of mtDNA containing MDVs is a second consequence of MAPL expression that is essential to load this subpopulation of lysosomes with mtDNA, from which it escapes in a gasdermin-dependent mechanism.

The reviewer raises an important point from the literature that gasdermin pores can form directly in mitochondria in certain conditions. While we can never be entirely sure that there is zero release of

mtDNA directly, a number of experiments strongly suggest this is not occurring in our conditions. First, the loss of gasdermins did not alter the release of mtDNA from the mitochondria (**Fig. 2J**). Instead, loss of gasdermins blocked the recruitment of Gal3 to lysosomes (**Fig. 6K**), providing strong evidence that the action of gasdermins occurs at lysosomes. We repeated and optimized the biochemical isolation of lysosomes and revealed the presence of the N-terminal fragments of gasdermin in this organelle, a biochemical fraction depleted of mitochondria (**Fig. 6L**). Second, the overexpression of galectin 3 rescued MAPL-induced cell death, providing further evidence that the functional repair of lysosomes, not mitochondria, promoted survival (**Fig. 6C**). Furthermore, should mtDNA release from mitochondria occur through Bax/Bak pores, our use of Bax/Bak double KO cell lines shows that MAPL-induced cell death occurred unabated in this system (**Fig. 1C**). Lastly, the loss of mtDNA in Rho0 cells did not block gasdermin cleavage (**New Fig. 2E**) or Gal3 recruitment to lysosomes (**Fig. 6H**), suggesting MAPL expression activates NLRP3 pathways in a manner that does not require mtDNA at all, whether from mitochondria or lysosomal pores. Not until lysosomal breach occurs does cytosolic mtDNA lead to the amplification of these signaling pathways drive pyroptotic cell death. It is an interesting question whether mtDNA may be somehow modified within lysosomes, making it a better activator of cGAS. This could be possible through oxidation or, perhaps more likely, through the action of lysosomal DNA hydrolases to fragment the mtDNA. We currently don't have evidence either way but it is an intriguing idea.

The reviewer asked whether VPS35 or NLRP3 loss would block immune signaling in MAPL overexpression. While we agree that these are interesting research avenues, we spent additional time working in primary BMDMs to define these pathways in response to physiological inflammation. This work confirmed that VPS35 is required for mtDNA release in response to LPS/IFN γ , and its loss inhibits inflammatory cell death. This was consistent with our MAPL overexpression findings, and we hope that these new experiments help to clarify the roles of MAPL and VPS35 in the pyroptotic pathway. Further delineation of the relationship between MAPL, VPS35, cGAS, and NLRP3 in the future will be critical to add additional mechanistic precision to our discovery of this pathway.

R2: Minor concerns:

1. While the data are largely convincing, there are some concerns with rigor based on the figures and methods provided. These concerns are addressable with more information and quantitation.
 - a. While many of the graphs contain points indicating repetition, (apart from Figure 1) the figure legends lack information detailing the number and type of replicates.

HM: We have now added precise numbers for independent repeats (i.e. on different experimental days) and technical repeats (i.e. total number of cells analysed for each condition) in the legends. All individual data points are now shown for every experiment, with the vast majority completed in triplicate.

R2: b. All microscopy, including Figure 1K & 3D, should be quantified. This quantification should include more than n=2 as in Figure 5.

HM: All experiments have now been completed at least three times. Where possible, all microscopy has been quantified. This was unfortunately not possible for electron microscopy given technical considerations.

R2: c. Details on how the quantitation of the microscopy was completed should be expanded. This is most unclear in graphs labeled “(% cells with enlarged lysosomes).” Does this mean these cells have 1 large lysosome or are all the lysosomes enlarged? What is the threshold for large lysosomes?

HM: We have expanded upon our quantification within figure legends and methods sections. We repeated and expanded our analysis of lysosome size upon MAPL expression. We now present two sets of analysis, first we show that expression of MAPL leads to an increase in the average lysosome size in all cells, from $\sim 0.35 \text{ } \mu\text{m}^2$ to $0.4 \text{ } \mu\text{m}^2$ (**New Supplemental Figure 6**). Upon MAPL expression in siLRRK2 cells we observe that almost 50% of cells show 1-4 very large lysosomes over a micron in diameter, which is shown in **Fig 5G,I, and New Fig. 6G**. In LRRK2KO BMDM treated with LPS/IFN γ we also observe the formation of very enlarged, micron sized lysosomes shown in **New Fig. 8G**. We hope this has clarified the lysosomal changes in our conditions.

R2: d. Some of the graphs lack statistics to match the statements provided (i.e. Figure 1D, Figure 2B/E/F/G/H, Figure 5D, Figure 6C). Additionally, a description of what statistics were performed are missing from Figures 5 & 7.

HM: Most experiments were completed in triplicate, so assessment of patterns across each repeat and reproducibility is a strong indicator of biological trends and our conclusions. All *P* values are stated exactly in graphs where appropriate.

R2:

2. Improvements should be made to the included supplementary figures:

- a. Supplement 1B-E are not referred to in the text.
- b. Supplement B-C are not referred to in the figure legend.
- c. There is a figure labeled supplement 2, but the provided legend is labeled supplement

3. This is referred to as supplement 3 in the text.

d. The figure labeled supplement 2 with the supplement 3 legend does not provide enough information. Is there a difference between the two panels in this figure?

We have corrected these issues.

Reviewer 3

Reviewer #3 (R3): In this manuscript titled “The mitochondrial ligase MAPL is an inflammatory rheostat that regulates immune signalling and cell death”, Mai Nguyen et al. report that MAPL induces mtDNA trafficking in MDVs to lysosome, followed by the release of mtDNA into the cytosol due to lysosomal permeabilization, the released mtDNA then leads to inflammation that induces pyroptosis. The findings shown in this manuscript are potentially interesting to the readers. However, the data in the manuscript are preliminary and do not support the conclusions well. Consequently, a substantial addition of direct evidences is required to solidify these findings.

Major comments

1. The major events (processes) of mtDNA release upon MAPL expression require direct evidences (images and videos): 1) the process (images and videos) of MDVs containing mtDNA (both MDV and mtDNA should be labeled) formed from mitochondria after MAPL expression; 2) the process (images and videos) of MDVs containing mtDNA contacting and fusing with lysosome, then mtDNA entering into lysosome; 3) the process (images and videos) of mtDNA within lysosome releasing into the cytoplasm.

We have now ensured that we have attached the supplementary videos (and referenced them appropriately in text) to ensure that the reviewer can view events in real-time in addition to our still images in the main figures.

HM: Our data address these concerns directly: (1) We show images of MDVs (complex I-positive, TOM40-negative) containing mtDNA (**New Fig. 3C-E, quantified in 3D**), as well as vesicular tubulation events where tubules contain complex I and mtDNA, but not TOM40, further highlighting the selectivity of MDV production (**New Fig. 3E**). A video is also included in Fig. 3A showing the dynamic release of mtDNA from mitochondria (**Fig. 3A and supplementary video 1**). (2) We show captured two types of events using time-lapse imaging: **Fig. 4E** shows cytosolic mtDNA fusing with a lysosome that already contains DNA and **Fig. 4E** shows lysosome recruitment to the site of mtDNA exit, enabling the rapid uptake of mtDNA. These are shown in **supplementary videos 2 and 3**. (3) We were able to capture an instance where mtDNA is released from an enlarged GAL3-RFP-positive lysosome, indicating mtDNA release into the cytosol from a damaged lysosome (**Fig. 6G and supplementary video 4**). While release of mtDNA from mitochondria in **Fig. 3A** shows a dynamic and rapid event, with directionality, the events observed in **Fig. 6G** are less active, more like “puffs” of PicoGreen staining that rapidly diffuses. This suggests it is a more passive ‘leakage’ of DNA. This leakage event was seen in one of the larger GAL3-positive foci, so it may not be possible to observe leakage of mtDNA from smaller lysosomes, where less DNA may be released which would be beyond the sensitivity of our approaches. We therefore fractionated cytosol at 24 h after MAPL expression where smaller GAL3-positive lysosomes were more common, and confirmed there was still release of mtDNA into cytosol by qPCR at this earlier time point (**New Fig 6H**).

R3: 2. There has been a lot of reports about the types and mechanism of mtDNA release upon some stresses. Does MAPL expression-induced mtDNA release depend on Bax/Bak pore or VDACs pore? Although the author showed that MAPL-induced cell death is not dependent on Bax/Bak pore or

VDACs pore, the role of Bax/Bak pore or VDACs pore in MAPL-induced mtDNA release should be tested.

HM: We have demonstrated the requirement for at least two distinct protein machineries essential for the release of mtDNA into cytosol, MIRO1/2 (**Fig. 3B**), and VPS35 (**Fig 5.E,F**). We now provide evidence that mtDNA is released along with protein components of complex I (**New Fig. 3C-E**), and we provide ultrastructural evidence of vesicular profiles emerging from mitochondria (**New Fig 3F**) and subsequent fluorescent and video evidence that these are ultimately delivered to lysosomes (**Fig. 4**). The process of MAPL-induced cell death occurs entirely independent from Bax/Bak (**Fig. 1C**) yet relies on release of mtDNA (**Fig. 2D**), and the unbiased CRISPR screen did not indicate a role for these proteins in MAPL induced pyroptotic cell death. We also show that the induction of lysosomal repair upon expression of Gal3 rescued MAPL-induced cell death (**Fig. 6C**), demonstrating the requirement of damaged lysosomes in this death pathway, further arguing against the direct release from mitochondria. We cannot rule out that Bax/bak pores contribute to some mtDNA release but given that cell death is dependent on mtDNA but not Bax/bak, suggests this is not a major mechanism.

R3: 3. MDVs of control and MAPL expressed cells could be purified to test the content of mtDNA within MDVs.

HM: We agree that it would be fantastic if we could selectively purify mtDNA-containing MDVs but, unfortunately, we have not yet identified a specific outer membrane marker to pull them out with (as we did in Konig et al NCB 2021). We know there is mitochondrial outer membranes from the clear EM images showing the double membrane MDVs emerging from mitochondria, and due to the dependence on outer membrane MIRO1/2 for their generation. This is a major focus of our future work.

R3: 4. In almost all figures related to MDVs, MDVs were not labeled. The specific marker for MDVs should be provided in the images. For example, in Figure 3, MDVs should be labeled (TOM20?) to confirm the released mtDNA locating in MDVs.

HM: **Figure 3C-E** now demonstrates that some DNA foci contain Complex I but not TOM40, demonstrating their selective nature and location in MDVs. The mtDNA containing MDVs are a new class for us since they are negative for many other proteins we commonly follow into the lysosome. We thank the reviewer for pushing us to expand our panel of antibodies we tested to identify the subunit of Complex I as a component.

R3: 5. MDV membrane should be labeled with membrane protein (such as TOM20), and MDV membrane would be not colocalized with lysosome after mtDNA is delivered into lysosome. The related experiments need to be provided.

HM: DNA⁺ MDVs are now labelled with Complex I, demonstrating the presence of mitochondrial membranes. The experiment required three distinct labels to confirm cargo specificity within MDVs (anti-DNA, anti-TOM40 and anti-Complex I), which did not leave us a channel to include

lysosomes. We suspect the reviewer is correct that the complex I cargo protein would then be degraded explaining for the loss of signal, as we have seen for TOM20 in our previous work (Soubannier et al., 2012). Our video analysis suggests that these MDVs can be rapidly taken up by lysosomes (**Fig. 4D and supplementary video 3**), so it may be difficult to capture significant number of MDVs prior to this event. For the moment we have visualized the generation of a Complex1⁺/mtDNA⁺/TOM40⁻ MDV clearly emerging from the mitochondria in tubular structure after MAPL overexpression, where the constrictions are seen as the site of MDV formation (**Fig. 3C-E**).

R3: 6. Since the primary function of VPS35 and MIROs is to transport selected cargo proteins, other key factors in MDV formation need to be examined to confirm that MDV is essential for mtDNA release.

HM: To our knowledge there are currently no regulators of MDV formation that do not have secondary functions. Snx9, VPS35, MIRO1/2, PINK1/Parkin, DRP1, MFF/MIDS, the generation of PA, and others will all face the same criticism the reviewer presents us with here (see our review documenting these mechanisms Konig Mol Cell 2023, Konig NCB 2021). Our work over the last many years has shown that there are distinct classes of MDVs induced with different triggers that require the action of selected machineries. The mechanisms of cargo selection and signaling pathways that regulate MDV formation are the major focus of our lab. In this case, we did show that Parkin is not recruited in inflammatory (MAPL expressing) conditions (**Supplementary Fig. 4**). We respectfully submit that we provide two distinct MDV-specific machineries as requisite for their formation, and provide a series of confocal, EM and video evidence of MDV formation and transport. With the help of this reviewer who pushed us to identify additional cargoes, we have done as much as we can at this point to demonstrate the release of mtDNA is indeed, in MDVs.

R3: 7. The colocalization of mtDNA and lysosome does not confirm that mtDNA is delivered to the lysosome, 3D reconstruction of confocal images were required to support the statement.

HM: We now provide a 3D datasets obtained from stacks of the spinning disc confocal imaging and clearly show the inclusion of the DNA signal well within the lysosomes (**New Supplementary Fig. 5**). Our new video analysis of the picogreen signal releasing from a larger lysosome labelled with Gal3 further demonstrates the absolute inclusion of mtDNA within lysosomes (**New Fig. 6G**).

R3: 8. The data about MAPL regulating inflammatory responses are insufficient. At least, the data about mtDNA-activated STING-cGAS pathway should be provided.

HM: We have done a significant amount of new work to outline the immune response to ectopic MAPL expression, and the impact of MAPL loss in primary MAPL KO BMDM cells treated with LPS/IFN γ , including a full RNAseq dataset.

1. The use of Rho0 cells showed that mtDNA was not required for MAPL-induced cleavage of gasdermins (**New Fig. 2E**), indicating that the arm of MAPL that activates NLRP3 and caspase signaling occurs independent from mtDNA. Importantly, the mtDNA pathway is critical for

pyroptotic cell death as Rho0 cells are fully protected, and cGAS/STING are essential. To this end we now provide further evidence confirming that MAPL drives pyroptotic cell death (rather than another form of pore-mediated, SYTOX green positive death) through the use of selective inhibitors of cGAS, STING, NLRP3 and caspase 1 – all of which inhibited MAPL induced cell death in primary BMDM cells (**New Fig. 7A**).

Further evidence showing mtDNA release as a critical step in MAPL-dependent pyroptotic cell death is now shown in **New Fig. 7H** where the addition of 2'3'cGAMP (cGAS product) to MAPL KO BMDM rescued pyroptotic death phenotype driven upon LPS/IFN γ treatment. These data argue that the primary role of MAPL in driving LPS/IFN γ driven pyroptosis is the activation of mtDNA transport and release from lysosomes.

2. To determine how MAPL may activate, or be requisite for, upstream immune signaling cascades we expanded our analysis of MAPL-induced expression of IL6, NLRP3, IL1 β , IFNA4 and IFNB1 in both U2OS and human fibroblast cells (**New in Fig. 1I and 2C**). Therefore, MAPL expression is fully capable of activating NLRP3 pathways in the absence of any other critical damaging agent (like Nigericin). We appreciate that our data do not yet map the precise mechanisms of this activation, but we speculate in the discussion that the SUMOylation of NLRP3 may play a role. We hope the reviewer may understand that the full mechanistic dissection of these events will take significantly more time to complete.

While MAPL expression can induce immune signaling and pyroptotic death, our new data show that MAPL is not essential for the primary immune response to LPS/IFN γ , since we still observe robust protein expression of NLRP3 and secretion of IL1 β in MAPL KO BMDM (**New Fig 7G,G'**). However, MAPL is required for pyroptotic cell death in this system (**Fig 7C**).

To better understand the requirement for MAPL in innate immune signaling and transcriptional pathways at a global level, we performed RNAseq on wild type or MAPL KO BMDM following LPS/IFN γ treatment at time 0, 4.5 hrs and 24 hours (full datasets in **New Supplementary Figure 8** and **Supplementary Table 2**). The gene expression of IL6 and many other cytokines was unaltered in MAPL KO BMDM, however there was a ~30% reduction in gene expression of a host of interferon related genes (IFITs and others, **New Fig. 7F**). This indicates that the signaling pathways downstream of TLR4 receptors to NLRP3 do not require MAPL, where MAPL plays a role in the full activation of candidate cGAS/STING targets (Type 1 interferon related genes). We have also shown that the loss of either VPS35 or LRRK2 in primary BMDM protect against pyroptotic cell death, but due to blocks in two distinct steps (**new Figure 8**). We hope the reviewer appreciates that we have done a significant amount of new work to better map the role of MAPL in immune signaling and cell death. The data highlight the role of this mitochondria-to-lysosome axis of mtDNA transport as a critical driver of pyroptotic cell death downstream of the core signaling pathways. We think this is reminiscent of apoptotic and other cell death programs where mitochondria act as the executioners after the signals have been sent.

R3: Minor comments

1. PicoGreen staining for mtDNA may also stain other DNA fragments. Therefore, the authors can

use TFAM-GFP puncta to further indicate mtDNA and confirm mtDNA is released after MAPL expression.

HM: To address this concern, we isolated cytosol by biochemical fractions and measured mtDNA, specifically, using qPCR, demonstrating an increase in cytosolic mtDNA upon MAPL expression. These data are shown in **New Fig. 6H**. We considered the best control to confirm that any anti-DNA staining we follow is derived from mitochondria was our use of Rho0 cells, where that signal was lost entirely (**shown now in Fig 2F**).

R3: 2. In Figure 2F, qRT-PCR assay for cytosolic mtDNA is required.

HM: These data is now available in **Figure 6H**.

R3: 3. In Figure 3A, MDVs should be labeled.

R3: MDVs are now labelled with Complex I in **Figure 3C and E**.

R3: 4. line 328-329, “We observed mtDNA release (Fig. 7A-B) at 6 hours of treatment, and subsequent delivery to lysosomes (Sup Fig. 3).” “supp Fig. 3” should be “Fig.2”; and supp Fig.2 just showed mtDNA colocalizing with lysosome, the statement “subsequent delivery to lysosomes” is not accurate.

HM: This statement has been removed.

R3: 5. In Figure 6A, the images related to MAPL- Δ RING should be provided.

HM: These images are now included throughout rather than just showing the quantification of the datasets.

R3: 6. “G” is missing in Figure 5G.

HM: This has been corrected.

R3: 7. line 304, “Fig.6H” should be “Fig.6G”

HM: This has been corrected.

R3: 8. There are too many references (105) in the manuscript.

HM: The format of Nature Cell Biology does not limit the citation number to 30 as seen in Nature. We feel it is incredibly important to cite the relevant work upon which our study is based. A feature of our study is the intersection of multiple fields from mitochondrial cell biology and MDV formation to innate immune signaling pathways, Parkinsons related cell biology, lysosomal dynamics and stability, and cell death pathways. For these reasons we have cited numerous studies that are directly relevant to the experiments, tools, molecules and pathways we have uncovered.

Cited References in response to review:

- Braschi, E., R. Zunino, and H.M. McBride. 2009. MAPL is a new mitochondrial SUMO E3 ligase that regulates mitochondrial fission. *EMBO Rep.* 10:748-754.
- Jiao, S., Z. Zhang, C. Li, M. Huang, Z. Shi, Y. Wang, X. Song, H. Liu, C. Li, M. Chen, W. Wang, Y. Zhao, Z. Jiang, H. Wang, C.C. Wong, C. Wang, and Z. Zhou. 2015. The kinase MST4 limits inflammatory responses through direct phosphorylation of the adaptor TRAF6. *Nat Immunol.* 16:246-257.
- Nakayama, A., J. Albarran-Juarez, G. Liang, K.A. Roquid, A. Iring, S. Tonack, M. Chen, O.J. Muller, L.S. Weinstein, and S. Offermanns. 2020. Disturbed flow-induced Gs-mediated signaling protects against endothelial inflammation and atherosclerosis. *JCI Insight.* 5.
- Prudent, J., R. Zunino, A. Sugiura, S. Mattie, G.C. Shore, and H.M. McBride. 2015. MAPL SUMOylation of Drp1 Stabilizes an ER/Mitochondrial Platform Required for Cell Death. *Mol Cell.* 59:941-955.
- Soubannier, V., G.-L. McLelland, R. Zunino, E. Braschi, P. Rippstein, E.A. Fon, and H.M. McBride. 2012. A Vesicular Transport Pathway Shuttles Cargo from Mitochondria to Lysosomes. *Current Biology.* 22:135-141.
- West, A.P., W. Khoury-Hanold, M. Staron, M.C. Tal, C.M. Pineda, S.M. Lang, M. Bestwick, B.A. Duguay, N. Raimundo, D.A. MacDuff, S.M. Kaech, J.R. Smiley, R.E. Means, A. Iwasaki, and G.S. Shadel. 2015. Mitochondrial DNA stress primes the antiviral innate immune response. *Nature.* 520:553-557.
- Xian, H., K. Watari, E. Sanchez-Lopez, J. Offenberger, J. Onyuru, H. Sampath, W. Ying, H.M. Hoffman, G.S. Shadel, and M. Karin. 2022. Oxidized DNA fragments exit mitochondria via mPTP- and VDAC-dependent channels to activate NLRP3 inflammasome and interferon signaling. *Immunity.* 55:1370-1385 e1378.
- Yang, B., J. Wang, Y. Wang, H. Zhou, X. Wu, Z. Tian, and B. Sun. 2013. Novel function of Trim44 promotes an antiviral response by stabilizing VISA. *J Immunol.* 190:3613-3619.
- Zecchini, V., V. Paupe, I. Herranz-Montoya, J. Janssen, I.M.N. Wortel, J.L. Morris, A. Ferguson, S.R. Chowdury, M. Segarra-Mondejar, A.S.H. Costa, G.C. Pereira, L. Tronci, T. Young, E. Nikitopoulou, M. Yang, D. Bihary, F. Caicci, S. Nagashima, A. Speed, K. Bokea, Z. Baig, S. Samarajiwa, M. Tran, T. Mitchell, M. Johnson, J. Prudent, and C. Frezza. 2023. Fumarate induces vesicular release of mtDNA to drive innate immunity. *Nature.* 615:499-506.

Nguyen, Collier et al., NCB-A53659A

We thank the editors for considering a second revision of our manuscript and to the Reviewers for their helpful and constructive comments. We believe that the new data generated to address these comments improved our manuscript and provided a clear presentation of the role MAPL plays in the regulation of immune processes and cell death. Throughout the “Response to reviewers”, we are including the data and stating the figures where they are presented within the new version of the manuscript.

Reviewer #1 (Remarks to the Author):

The authors have comprehensively addressed all the points I raised during initial review.

HM: We are thrilled that our efforts after the first revision addressed all the comments from Reviewer 1.

Reviewer #2 (Remarks to the Author):

R2: In their revised manuscript, Nguyen et al. greatly improve the quality and quantification of their imaging data. Moreover, the mechanisms defining how VPS34 and MIROs modulate the generation of MAPL-dependent mtDNA containing MDVs has been strengthened and is convincing. However, an exciting and significant portion of the paper focuses on how MAPL dependent mtDNA release activates the innate immune system leading to NLRP3 dependent pyroptosis and cGAS dependent ISG expression. Although the authors added some new experiments to document the role of NLRP3 and cGAS, the data do not go far enough and leave many important aspects unclear. Without additional experiments to clarify and solidify the proposed model, the paper falls short of warranting publication in NCB.

1. We acknowledge the effort made to measure Nlrp3 and Il1b transcripts in figure 1, but as described in point 1 of our review, this is not sufficient to conclude pyroptosis is happening. At a minimum, evidence including western blotting and ELISA evidence or use of inhibitors in these non-immune cell lines is required.

HM: Our data now demonstrate that expression of MAPL in non-immune U2OS cells is sufficient to activate key markers of pyroptosis, including the induction of NLRP3 expression, gasdermin cleavage (as we had already shown), as well as IL6 secretion (ELISA), now in **Fig.1I and J** (shown right).

R2: Figure 1 shows GSDMD and GSDME cleavage that is inhibited by caspase inhibitors. There is also SYTOX Green uptake in cells. The requirement for GSDMD and GSDME is highlighted by the data showing GSDME cleavage and GSDMD knockdown in this figure. Robust scientific literature has shown that GSDME is not strongly cleaved by activation of NLRP3. However, the authors show that GSDME is cleaved at much higher

levels than GSDMD in their MAPL-OE models. Additionally, GSDMD knock down does not provide a complete rescue, and therefore it is still unclear if NLRP3-dependent pyroptosis is the main form of cell death being initiated in the cell line experiments. Caspase-1-induced apoptosis involves the Bid-caspase-9-caspase-3 axis, which can be followed by GSDME-dependent secondary necrosis or pyroptosis. The authors should clarify the differential activation of GSDMD and GSDME and not lump them together in the manuscript and model.

HM: We thank the reviewer for pushing us to further dissect the relationship between GSDME and GSDMD in this study. We have explored this question in a number of ways and made significant progress that has greatly refined the model. Following from our initial finding that MAPL expression led to a BAX/BAK-independent cleavage of both caspase-3 and caspase-7, an observation that was placed within Supp. Fig.1 of our previous submissions. It was this observation that led us to perform the genome-wide screen to map the mechanism of MAPL-induced cell death. We now tested whether either of the two GSDMs was required for caspase-3/7 cleavage, and show that their loss had no impact on this early event (now in **Fig.2K**, and right). In separating the two gasdermins we also discovered that GSDME was required for the cleavage of GSDMD, placing it upstream in the pathway to MAPL-induced death.

To address the question of each gasdermin in the pathway we now provide evidence that silencing either gasdermin D or gasdermin E effectively protects against SYTOX green uptake (new **Fig.1L**, left). However, we observed a major distinction between the requirement for these two gasdermins when looking at the recruitment of GAL3 to lysosomes, indicative of their

breach. In this experiment the loss of GSDME led to a near-complete loss of GAL3 recruitment to lysosomes, while the loss of GSDMD had no effect (new **Fig.6L**, shown right). The accompanying western blots for this experiment again shows the dependency on GSDME for GSDMD cleavage (new **Fig.6L**, shown right).

Induced expression of NLRP3 remained independent of either of the two GSDM proteins, consistent with MAPL activating caspases3/7 upstream of GSDM cleavages. This strongly indicates that GSDME function at the lysosome is required for the release

of mtDNA, which would then amplify activation of the inflammasome and caspase-1 required to cleave GSDMD and drive the final steps of pyroptotic death. We have updated our model in **Fig.8J** (shown left) to reflect the new data resulting from the comments of Reviewer 2.

R2: 2. Much of the paper relies on a MAPL overexpression system and is conducted in cells that do not have a canonical and fully functional inflammasome system. Therefore, the experiments in BMDMs are extremely important in solidifying the mechanisms and increasing the biological/immunological relevance of the findings. As I am sure the authors are aware, the NLRP3 inflammasome is activated in a two step process, but here, the authors argue for a direct activation of NLRP3 absent of an obvious signal.

HM: We fully understand that canonical inflammasome activation requires a two-step

process. We were as surprised as the reviewer to learn that the simple expression of a mitochondrial ligase could activate such a profound death pathway, which we argue is an important and novel finding. Our dissection of the MAPL-mediated death pathway stemming from an unbiased CRISPR screen unmasked the host of inflammasome-related genes whose loss was protective against MAPL-induced death, which was what led us to suggest that MAPL expression leads to the activation of NLRP3 in the absence of an obvious signal. New experiments performed in response to this second review (below) further demonstrate how MAPL leads to the activation of NLRP3 in the absence of a canonical (obvious) signal. Our work in MAPL KO BMDM further demonstrate that MAPL is requisite for LPS/IFN γ -induced pyroptosis (**Fig.7C, D**) further emphasizing the significant contribution of MAPL in a physiological inflammation pathway.

R2: New data suggest that mtDNA release is not required for NLRP3 activation and caspase-1-dependent GSDMD cleavage. So, what is the mechanism by which NLRP3 gets activated?

Inhibitors used in 7A show that NLRP3, Caspase-1, cGAS, and STING are all required for cell death in BMDMs after MAPL overexpression. Does MAPL overexpression oligomerize NLRP3 CARD domains or modify mitochondria-localized NLRP3? This could be assessed in native gels. Also, is ASC oligomerization required for MAPL overexpression-induced NLRP3 activity (assess by native gels)? Finally, can the authors see traditional NLRP3 activation hallmarks in the lysates from exp 7A? That is, Caspase-1 cleavage to the p20 form, GSDMD cleavage, and mature IL-1 β secretion? In addition, can the authors blot lysates to examine NF- κ B activation and IRF/STAT activation in the MAPL overexpressing BMDMs? I applaud the team for developing this system, but the absence of more definitive data here is disappointing.

HM: To begin, we first demonstrate that MAPL expression in BMDM induces the expression of the suggested markers including ASC1, NLRP3, p-STAT3 and NF- κ B p65, along with cleavage of both GSDMD and GSDME (now **Fig.7A**, shown left). We also completed BN-PAGE gels to

analyse oligomerization of NLRP3 and ASC. Compared to the strong oligomerization induced by LPS/nigericin at 1 hour, MAPL expression over 24 hours does not lead to significant NLRP3 oligomerization. However, we do see an induction of ASC1 and NLRP3 primarily in the TX-100 insoluble fractions (suggesting phase-separated or aggregation), along with the other markers which are hallmark features of pyroptotic cell death (new **Supp. Fig.7A, B**, shown below).

Overall the data suggests a requirement for an amplification (or two-step) process for MAPL-induced death where the initial event is the activation of caspase-3/7, an event we do not yet understand and have discussed in the limitations section at the end of the manuscript. In the absence of mtDNA release (Rho0 experiments) this is not enough to kill the cells. Importantly, the SYTOX green uptake was rescued in MAPL KO BMDM upon addition of the cGAS agonist cGAMP (**Fig.7L**), which suggests that the appearance of mtDNA in cytosol and activation of cGAS/STING is essential for cell death – but that this effect is NOT through any transcriptional targets. We now discuss this in the manuscript and speculate that the arrival of STING at the lysosome may lead to proton leak that facilitates GSDME pore expansion/assembly and mtDNA release into cytosol, an amplification step that activates NLRP3 further resulting in GSDMD cleavage and death. We humbly submit that the details of the non-transcriptional, biochemical requirements of cGAS/STING are the subject of our next study.

Therefore we hypothesize that the initial trigger by MAPL at mitochondria launches a pathway to lysosomes which can only kill if sufficient mtDNA release promotes the amplification of NLRP3 activation and GSDMD cleavage through the (non-transcriptional) actions of cGAS/STING. We sincerely hope the reviewer can appreciate the dissecting these complex steps will require extensive work and is beyond the scope of this initial discovery.

R2: 3. LPS/IFN γ experiments show that MAPL-KO BMDMs have less SYTOX green + cells. Are the WT cells undergoing pyroptosis? Can the authors blot lysates as described above to check? Also, does the NLRP3i, etc. block SYTOX+ cells?

HM: We now provide evidence that our primary BMDM treated with LPS/IFN γ show a clear induction of NLRP3 oligomers by BN-PAGE, which is reduced with NLRP3i (Supp. Fig.7C,D, shown right). We also observe the induction of p-STAT3, NF- κ B activation and gasdermin cleavage, all three of which are effectively inhibited by the NLRP3 inhibitor (shown in panel D, right). This shows that our system is reflecting pyroptotic pathways.

The last round of review asked the authors to more closely examine how MAPL-KO BMDMs responded to canonical inflammasome stimuli. For example, LPS priming + nigericin and/or ATP, then assessing NLRP3 levels, caspase-1 cleavage, GSDMD cleavage. These experiments were not provided but are critical controls.

In this second revision we now provide evidence that nigericin/LPS treatment of BMDM leads to a very strong induction of inflammasome oligomerization and signaling (NLRP3 induction, p-STAT3, NF- κ B activation) at 1 hour of treatment. Loss of MAPL did not impact this, as seen in new Supp. Fig.7E,F, shown right. Based on the previous work of the Leiberman lab and others, it has been suggested that nigericin leads to a breach of mitochondria that facilitates GSDM-mediated mtDNA release. As a potassium ionophore that exchanges protons for potassium across all membranes, it disrupts membrane potential and induces swelling across the plasma membrane, but also mitochondria and lysosomes. It makes sense to us that a death trigger that acts to directly disrupt mitochondrial membranes would not require this more regulated pathway of MAPL-mediated vesicle transport to lysosomes, where mtDNA release then depends on the generation of selective lysosomal pores (in this case via GSDME). We have added these data to the manuscript and conclude that MAPL is not requisite for the signaling and death induced by nigericin.

R2: Moreover, requested experiments to document pyroptosis in the LRRK2 KO cells were not provided upon revision.

We did not provide this because LRRK2 KO cells do not die, which is a critical finding of this study. We show that the mtDNA still arrives in lysosomes, and that the lysosomes enlarge but death is blocked (all in Fig.8).

R2: 4. Do the downregulated interferon genes in 7F and S8C reach significance in MAPL-KOs vs WT BMDMs? This is unclear, but critically important. Changes at the RNA level are not convincing if they do not translate to measurable decreases at the protein level, so blots would be informative.

HM: We did highlight the reduction of a subset of genes by ~20-40% which were generally within the Type I IFN gene signatures known to be targets of cGAS/STING. The reviewer was correct that we did not confirm these at the protein level. In the second revision, we highlight one of these, IFIT2 (for which there is a strong antibody). This transcript of this gene was reduced by ~40%, but Western blot analysis showed almost no change in the induced expression in the absence of MAPL, data which is now included in the main **Fig. 7J, K** (shown right). Given this new information we have placed the heat map highlighting the 50 most reduced genes into supplementary data **Supp.Fig.8** (since the relevance of this is in question based on these revisions) and moved the heat map showing that the majority of genes induced by LPS/IFN γ are unaltered upon the loss of MAPL into **Fig.7**.

We thank the reviewer for encouraging us to further explore the mechanisms of our pathway and hope that our revised manuscript with new experiments will satisfy their concerns.

Reviewer #3 (Remarks to the Author):

The quality of the revised manuscript has improved, and some of my concerns have been addressed and resolved. However, there are still several questions and issues that remain unresolved in the revised version.

Major comments

R3: 1. MAPL overexpression led to the formation of electron-dense double-membrane protrusions from mitochondria (figure 3F). However, figures 3C and 3E indicate that MDVs are mtDNA+/CI+/TOM40-, suggesting that MDVs should consist of a single membrane derived from the inner mitochondrial membrane. This presents an inconsistency in the data.

HM: A key, universally accepted definition of all MDVs is the concept of strict cargo selection, which means that most outer membrane proteins will be EXCLUDED from double membrane-bound MDVs. Moreover, MDV formation does not occur by extrusion of membrane through outer membrane pores but through the process of membrane protrusion and ultimate scission of a cargo-selected vesicular structure that can be composed of either outer mitochondrial membrane alone, or contain both inner and outer membrane content. We emphasise within the manuscript that we have not yet identified an outer membrane marker of these MDVs, but provide ultrastructural evidence to report on the presence of both inner and outer membrane. The process of MDV formation is based upon strict cargo selection mechanisms, so the fact that TOM20 or other outer membrane markers are absent is part of the proof that they are cargo-selected MDVs. This is why we confirm the process of MDV formation by

silencing either the MIROS or VPS35, both of which block the formation of mtDNA⁺ MDVs. As the reviewer requested in the previous round, we did show the presence of CI to confirm the mtDNA foci are within cargo-selected MDVs that includes protein.

R3: 2. The data in figure 3 primarily demonstrate mtDNA release through MDVs, with information on MIROS limited to figure 3B. Therefore, the title of the paragraph, 'MIROS facilitate mtDNA removal in mitochondrial-derived vesicles,' should be revised. Additionally, since the role of MIROS in MDV formation has already been reported (Cargo-selected MDVs are formed by the lateral tubulation of mitochondrial membranes along microtubules via the Rho GTPases MIRO1 and MIRO2 before DRP1-mediated scission 40), and the data related to MIROS in this manuscript are not closely aligned with the main focus and could therefore be omitted.

HM: The reviewer is correct that we previously published evidence that the MIRO GTPases are central and essential drivers of MDV formation. Therefore the MIRO silencing experiments are a key piece of data confirming, in addition to our confocal and electron microscopy work, that DNA is released in MDVs. A key function of MIROS is the production of MDVs, and when they are depleted using siRNA, the amount of DNA released is reduced. Together our new data showing CI⁺/mtDNA⁺ emerging from mitochondria in tubules strongly implicate mtDNA release in MDVs.

R3: 3. line 160-163, "silencing Gasdermin C, D and E together had no effect on the appearance of cytosolic mtDNA foci in MAPL-expressing cells (Fig. 2J). These data indicate that mtDNA is unlikely to be released to the cytosol through mitochondrial membrane pores". How are MDVs released into the cytosol upon MAPL expression? Is it through Bax/Bak or VDAC1 pores? No supporting data have been provided to address this.

HM: We provided evidence that the process is independent of Bax/Bak. We assumed the reviewer was requesting that we experimentally test whether mtDNA release in our system may be occurring through the very recently published VDIM pathway, with extrusion of inner membrane through VDAC pores (PMID: 39169179). Those authors showed VDIMS are formed independent from MIRO, which places them in a different class from ours. We already demonstrated that mtDNA release in MDVs is dependent on 2 distinct proteins we have previously shown are required for canonical double-membraned MDV formation, MIROS and VPS35. We further provided ultrastructural analysis documenting the double membraned structures, and we showed extensive video and confocal data to document the delivery of these MDVs to lysosomes, where genetic and biochemical evidence demonstrate that gasdermin pores form to release mtDNA from lysosomes.

C

For this second revision we directly tested whether VDAC1 is required for mtDNA release from mitochondria upon MAPL expression. We silenced VDAC1 (central for VDIM) and saw no impact on the MAPL-induced release of mtDNA in U2OS cells silenced for VDAC1 (new **Fig.3C**, shown left). There was also no effect on the induction of SYTOX green cell death, or on MAPL induced NLRP3 expression and gasdermin cleavage events (new **Supp. Fig.3B,C**, shown above right).

R3: 4. Supplementary video 1 is not convincing. Firstly, the quality is poor with low resolution. Additionally, MitoTracker is not ideal for this experiment, as it depends on mitochondrial membrane potential and cannot demonstrate mtDNA release in a MDV-dependent manner. An outer membrane marker (such as TOM20) and an inner membrane marker (such as a complex I subunit) should be used instead.

HM: This is a key experiment that suggested mtDNA is released in MDVs, which we followed up with additional data highlighting the role of VPS35, MIROs, and confocal imaging showing selective cargo selection, in addition to electron microscopy showing clear double-membrane mitochondrial protrusions. The use of MitoTracker was intentional, as MDVs do not carry potential, therefore live video showing a mtDNA foci exiting a MitoTracker-positive mitochondria is how we initially report on cargo-selected MDVs. These live video examples are expanded and confirmed throughout the manuscript where we use fixed immunofluorescence for C1 and other cargo-excluded markers like TOM40.

R3: 5. The authors demonstrate that “mtDNA is trafficked to endolysosomal subcompartments” and “These data identify lysosomes as the target organelle for mtDNA-containing MDVs”. Then, what happens to mtDNA after entering the endolysosomal subcompartments? Is the mtDNA degraded? The data in figure 6 show that mtDNA can escape into the cytosol. However, how much mtDNA actually escapes?

HM: Our qPCR data reveal that a small subset of mtDNA escapes, as seen in the live imaging example. We understand that this will be a fraction of the lysosomal mtDNA, but it only needs to be enough to activate cGAS/STING. The bulk is likely degraded. We consider mtDNA transport to lysosomes is at least one pathway for its degradation, and only in the context of inflammatory signaling would it be released into the cytosol. We now provide further evidence demonstrating that GSDME is requisite for lysosomal breach (**Fig.6L**), and that this step is important for GSDMD cleavage (**Figs 2K and 6L**), placing the two gasdermins sequentially along the pathway.

R3: 6. The authors report that mtDNA is released into the cytosol from damaged lysosomes to activate cGAS/STING signaling, a process dependent on gasdermin pores in the lysosome. However, does gasdermin knockdown inhibit MAPL-induced activation of cGAS/STING signaling, considering that gasdermin knockdown does not block mtDNA release from mitochondria?

HM: We have shown through a series of experiments outlined above that the mtDNA release from mitochondria occurs within MDVs, which are targeted to lysosomes.

Therefore to get the mtDNA into the cytosol requires a breach at lysosomes, which we demonstrate directly through live confocal imaging (**Fig. 6J**) and through the recruitment of GAL3 to indicate which lysosomes are ruptured (**Fig. 6A-D**). We now show that loss of GSDME, but not GSDMD, completely blocked the recruitment of GAL3 to lysosomes (new **Fig 6L**), and fully protected against cell death (new **Fig 1L**).

We also show that the block in pyroptotic cell death in BMDM lacking MAPL (and the downstream pathway) can be rescued upon the addition of cGAMP directly. This tells us that an irreversible step of the pyroptotic pathway is the activation of cGAS. In our updated model (**Fig. 8J**), we proposed a two-step process for MAPL-induced death where the initial event is the activation of caspase-3/7. MAPL induced the release of mtDNA+ MDVs that are transported to the lysosome. We suggest that there is an amplification of NLRP3 activation and GSDMD cleavage through the actions of cGAS/STING.